# Modeling and simulation of neocortical micro- and mesocircuitry (Part II, Physiology and experimentation)

**James B Isbister[1][*][†], András Ecker[1][†], Christoph Pokorny[1][†], Sirio Bolaños-Puchet[1][†], Daniela Egas Santander[1][†], Alexis Arnaudon[1], Omar Awile[1], Barros-Zulaica Natali[1], Jorge Blanco Alonso[1], Elvis Boci[1], Giuseppe Chindemi[1,2], Jean-Denis Courcol[1], Tanguy Damart[1], Thomas Delemontex[1], Alexander Dietz[1], Gianluca Ficarelli[1], Mike Gevaert[1], Joni Herttuainen[1], Genrich Ivaska[1], Weina Ji[1], Daniel Keller[1], James King[1], Pramod Kumbhar[1], Samuel Lapere[1], Polina Litvak[1], Darshan Mandge[1], Eilif B Muller[1,3,4], Fernando Pereira[1], Judit Planas[1], Rajnish Ranjan[1], Maria Reva[1], Armando Romani[1], Christian Rössert[1], Felix Schürmann[1], Vishal Sood[1], Aleksandra Teska[1], Anil Tuncel[1], Werner Van Geit[1], Matthias Wolf[1], Henry Markram[1][‡], Srikanth Ramaswamy[1,5][‡], Michael W Reimann[1][*][‡]**

[1]Blue Brain Project, École polytechnique fédérale de Lausanne (EPFL), Campus Biotech, Geneva, Switzerland; [2]Department of Basic Neurosciences, University of Geneva, Geneva, Switzerland; [3]CHU Sainte-Justine Research Center, Montréal, Canada; [4]Department of Neurosciences, Faculty of Medicine, Université de Montréal, Montréal, Canada; [5]Neural Circuits Laboratory, Newcastle University, Newcastle, United Kingdom

***For correspondence:**
james.isbister@
openbraininstitute.org (JBI);
mwr@reimann.science (MWR)

[†]These authors contributed equally to this work
[‡]These authors also contributed equally to this work

## eLife Assessment

This study reports a model of eight somatosensory areas of the rat cortex consisting of 4.2 million morphologically and electrically detailed neurons. The authors carry out simulation experiments aimed at understanding how multiscale organization of the cortical network shapes neural activity. While the reviewers found the results to be **solid**, they note that they could have likely been obtained using a much smaller portion of the model. Nonetheless, the release of the modeling platform represents a significant contribution to the field by providing a **valuable** resource for the scientific community.

**Abstract** Cortical dynamics underlie many cognitive processes and emerge from complex multiscale interactions, which are challenging to study in vivo. Large-scale, biophysically detailed models offer a tool that can complement laboratory approaches. We present a model comprising eight somatosensory cortex subregions, 4.2 million morphological and electrically detailed neurons, and 13.2 billion local and mid-range synapses. In silico tools enabled reproduction and extension of complex laboratory experiments under a single parameterization, providing strong validation. The model reproduced millisecond-precise stimulus-responses, stimulus-encoding under targeted optogenetic activation, and selective propagation of stimulus-evoked activity to downstream areas. The model's direct correspondence with biology generated predictions about how multiscale organization shapes activity; for example, how cortical activity is shaped by high-dimensional connectivity motifs in local and mid-range connectivity, and spatial targeting rules by inhibitory subpopulations. The latter was facilitated using a rewired connectome that included specific targeting rules observed for different inhibitory neuron types in electron microscopy. The model also predicted the role of

inhibitory interneuron types and different layers in stimulus encoding. Simulation tools and a large subvolume of the model are made available to enable further community-driven improvement, validation, and investigation.

## Introduction

Neuroscience aims to characterize the dynamics of the brain and understand how they emerge as a product of anatomy and physiology. This is challenging, however, due to the complexity of the brain's multiscale organization and its study in situ. Large-scale, data-driven, biophysically detailed models (*Markram et al., 2015*; *Billeh et al., 2020*) offer a tool that can complement laboratory investigation. This 'bottom-up' approach combines detailed constituent models into a 3D model of a given brain volume, thus consolidating disparate, multiscale data sources. Such models aim to provide a general tool for multiscale investigation; access and manipulation of any model component enable predictions about how these components combine to shape emergent dynamics. With increasing biological data, model refinement and validation must be continuous. This necessitates (1) open sourcing of models with high-quality software tools for iterative, community-driven refinement and validation, and (2) rigorous methodology for model parameterization and validation, such that model iterations can be meaningfully compared. Here and in our companion paper (*Reimann et al., 2024*), we present our methodology for parameterizing the anatomy and physiology of a cortical model, and validating its emergent dynamics. This enabled predictions about how cortical activity is shaped by high-dimensional connectivity motifs in local and mid-range connectivity, and spatial targeting rules by inhibitory subpopulations. Elsewhere, this has allowed the presented model to be used to study the formation of cell assemblies (*Ecker et al., 2024b*), functional synaptic plasticity (*Ecker et al., 2024a*), propagation of activity between cortical areas (*Bolaños-Puchet and Reimann, 2024*), the role of non-random connectivity motifs on network activity (*Pokorny et al., 2025*) and reliability (*Egas Santander et al., 2025*), the composition of high-level electrical signals such as the EEG (*Tharayil et al., 2025*), and how spike sorting biases population codes (*Laquitaine et al., 2024*).

Specifically, we built and validated a model of the entire non-barrel primary somatosensory cortex (nbS1) comprising eight subregions. Whilst our previous data-driven, biophysically detailed model (*Markram et al., 2015*) provided insights at the scale of a single cortical column (*Reimann et al., 2013*; *Reimann et al., 2017*; *Reimann et al., 2022*; *Nolte et al., 2019*; *Nolte et al., 2020*; *Newton et al., 2021*), the new model is ~140 times larger and to our knowledge offers the first simulations of in vivo-like spontaneous and stimulus-evoked activity in a biophysically detailed cortical model with interregion connectivity. In the companion paper, we introduce the anatomical model (*Figure 1*, step 1), describing how neuron morphologies were placed within an atlas-based geometry and connected through local and mid-range synapses. Here, we describe our improved techniques to model and validate the electrical properties of neurons and synapses (*Figure 1*, steps 2 and 3; *Reva et al., 2023*; *Barros-Zulaica et al., 2019*), and to compensate for input from missing brain areas (*Figure 1*, step 4). These improvements enabled enhanced validation of emerging in vivo-like activity (*Figure 1*, step 5), including the reproduction and extension of five published studies in rodent sensory cortex under a single in vivo-like regime (*Figure 1*, step 6).

Validating that network activity emerges from the same interactions driving in vivo dynamics, requires (1) an approach to tackle the large parameter space without overfitting and (2) comparison of emerging dynamics with laboratory experiments. With respect to (1), we strongly prefer parameterization with directly measured quantities over fitting parameters to yield the correct emerging activity. Additionally, where fitting is applied, we adhere to a principle of *compartmentalization* of parameters. That is, once a parameter has been parameterized at one biological level, it is no longer a free parameter at a higher level. For example, the maximal conductance of a synapse is fit to biological amplitudes of postsynaptic potentials, but is then never updated in the process of reaching in vivo-like population activity. If a connection is valid on the single-cell level, its contribution at the population level should be equally valid. As a result, the emerging in vivo-like activity is the consequence of only 10 free parameters representing the strength of extrinsic input from other brain regions into 9 layer-specific excitatory and inhibitory populations, and a parameter controlling the noise structure of this extrinsic input. These parameters were fitted using a novel methodology presented here and were in

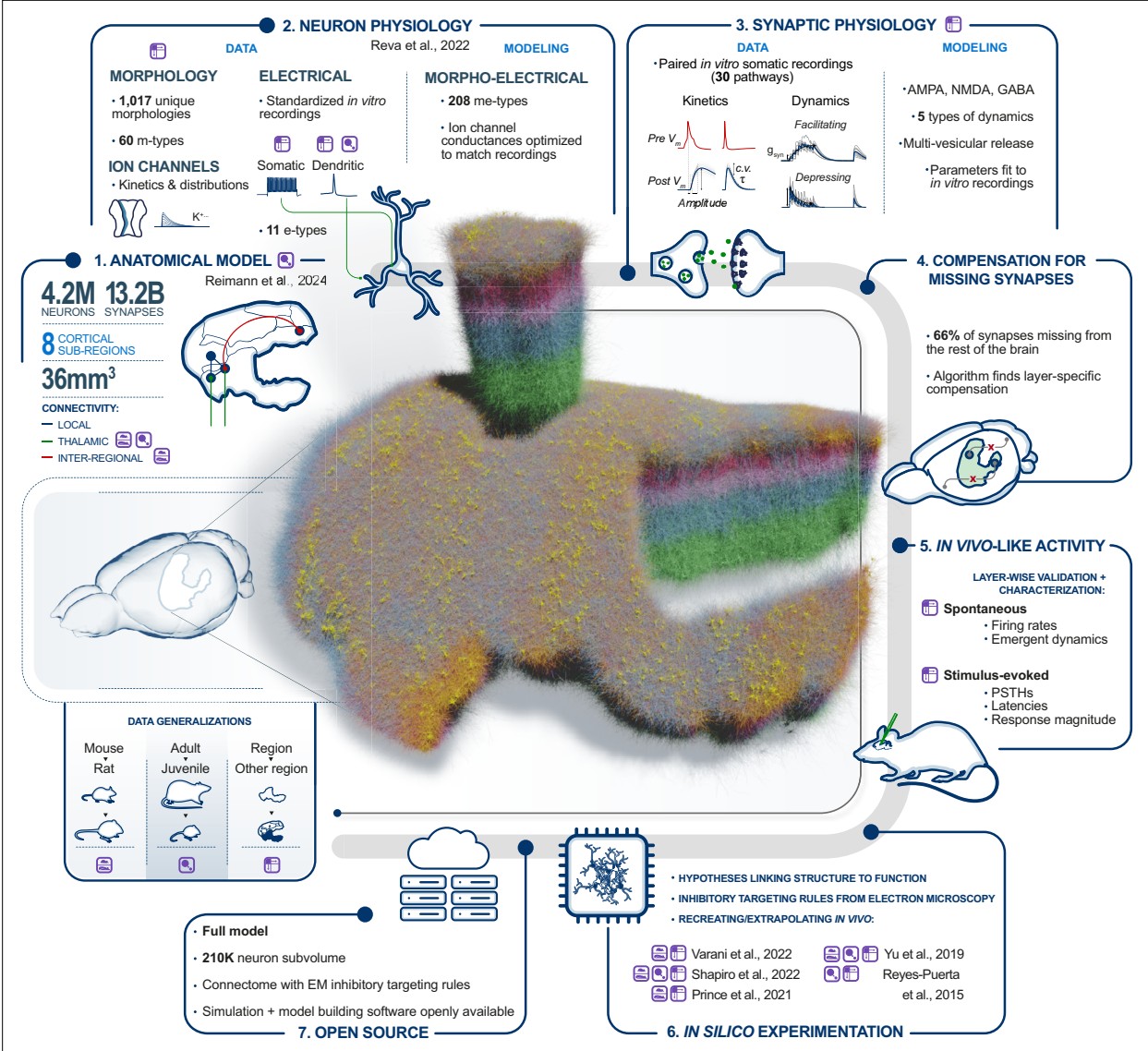

**Figure 1.** Overview of the physiology and simulation workflow. 1. Anatomical model: summary of the anatomical nbS1 model described in the companion paper. 2. Neuron physiology: neurons were modeled as multicompartment models with ion channel densities optimized using previously established methods and data from somatic and dendritic recordings of membrane potentials in vitro. 3. Synaptic physiology: Models of synapses were built using previously established methods and data from paired recordings in vitro. 4. Compensation for missing synapses: Excitatory synapses originating from outside nbS1 were compensated with noisy somatic conductance injection, parameterized by a novel algorithm. 5. In vivo-like activity: We calibrated an in silico activity regime compatible with in vivo spontaneous and stimulus-evoked activity. 6. In silico experimentation: Five laboratory experiments were recreated. Two were used for calibration and three of them were extended beyond their original scope. 7. Open Source: simulation software and a seven-column subvolume of the model are available on Zenodo (see 'Data availability statement'). Data generalizations: three data generalisation strategies were employed to obtain the required data. Left: mouse to rat; middle: adult to juvenile (P14) rat; right: hindlimb (S1HL) and barrel field (S1BF) subregions to the whole nbS1. Throughout the figure, the corresponding purple icons show where these strategies were used.

accordance with the mean number of missing synapses for each population (*Felleman and Van Essen, 1991*; *Harris et al., 2019*; *Gao et al., 2022*).

With respect to (2), we conducted the following validations: spontaneous activity reproduced layer-wise in vivo firing rates (or a specified proportion of in vivo firing rates to account for in vivo recording bias; *Wohrer et al., 2013*), varied along a spectrum of asynchronous to synchronous activity, exhibited spatially structured fluctuations, and produced long-tailed firing rate distributions with sub 1 Hz peaks as in vivo (*Wohrer et al., 2013*; *Buzsáki and Mizuseki, 2014*). Under the same parameterization, the model reproduced precise millisecond dynamics of layer-wise populations in response to

simple stimuli. These initial validations demonstrated that the model was in a more accurate regime compared to *Markram et al., 2015*—an essential step before testing more complex or larger-scale validations. For example, under the same parameterization we then observed selective propagation of stimulus-evoked activity to downstream areas, and reproduced and extended more complex experiments through accurate modeling of targeted optogenetic stimulation and lesions. Importantly, we highlight where emergent activity shows discrepancies with in vivo activity, to guide future data-driven model refinement.

The model generated a number of predictions (*Supplementary file 1*), including about the role of different layers in driving layer 2/3 stimulus-responses and how inhibitory interneuron types encode contrast, synchronous, and rate-coded information. Additionally, with access to the full structural connectome and tools for precisely editing it (*Pokorny et al., 2025*), we were able to make predictions about the relationship between structure and function. For example, we predict that an increase in the prevalence of non-random connectivity motifs towards deeper layers leads to a matched increase in spiking correlations, and that subregions more strongly innervated by mid-range connectivity have higher correlated activity locally. Additionally, we generated a new connectome that captured recently characterized spatially-specific targeting rules for different inhibitory neuron types (*Schneider-Mizell et al., 2025*) in the MICrONS electron microscopy dataset (*MICrONS Consortium, 2025*), such as increased perisomatic targeting by PV+ neurons, and increased targeting of inhibitory populations by VIP+ neurons. Comparing activity to the original connectome gave predictions about the role of these additional targeting rules. For example, inhibitory populations were more strongly inhibited (increasingly towards central layers) and required more non-local drive to reach the firing rates observed in vivo. Evoked responses increased and decreased in superficial and deeper excitatory populations respectively, suggesting layer-specific roles of the more specific inhibitory targeting.

To provide a framework for further studies and integration of experimental data, the full model is made available with simulation tools, as well as a smaller subvolume with the optional new connectome capturing inhibitory targeting rules from electron microscopy (*Figure 1*, step 7). The detailed modeling approach provides a one-to-one correspondence with most types of experimental data, allowing different datasets to be readily integrated. Due to the incredible speed of discovery in neuroscience, an integrative model will always be lagging behind the latest available data. We believe the solution is to provide a scientifically solid, validated model with clearly characterized strengths and weaknesses, along with the tools to advance or customize it for individual projects. We have therefore also made our tools for building and improving the model openly available: https://www.github.com/BlueBrain.

## Results

The companion paper describes the full anatomical nbS1 model containing 4.2M morphological neuron models, connected through 9.1B local synapses and 4.1B mid-range synapses (*Reimann et al., 2024*). Each neuron is modeled as a multicompartmental model belonging to one of 60 morphological types (m-types; *Figure 2A1*), and is either an instance or statistical variant of an exemplar in a pool of 1017 morphological reconstructions. Neurons were placed (orientated towards the surface) within an atlas-based geometry (*Figure 2A2*) based on estimated layer-wise density profiles of different morphological types. Local connectivity is based on axo-dendritic overlap with neighboring neurons, whilst mid-range connectivity combines data on interregion connectivity and laminar innervation profiles. Thalamocortical afferents were also modeled based on laminar innervation to the barrel cortex. To simulate emergent activity it is necessary to model and validate the electrical properties of these neurons and synapses (*Figure 1*, steps 2 and 3). These modeling steps are based on published methods and data sources (*Table 1*) and are summarized first. The method for compensating for missing synapses and the remaining results are then described (*Figure 1*, steps 4–6).

### Improved modeling and validation of neuron physiology

Similarly to *Markram et al., 2015*, electrical properties of single neurons were modeled by optimizing ion channel densities in specific compartment-types (soma, axon initial segment [AIS], basal dendrite, and apical dendrite) (*Figure 2B*) using an evolutionary algorithm (IBEA; *Van Geit et al., 2016*) so that each neuron recreates electrical features of its corresponding electrical type (e-type) under multiple

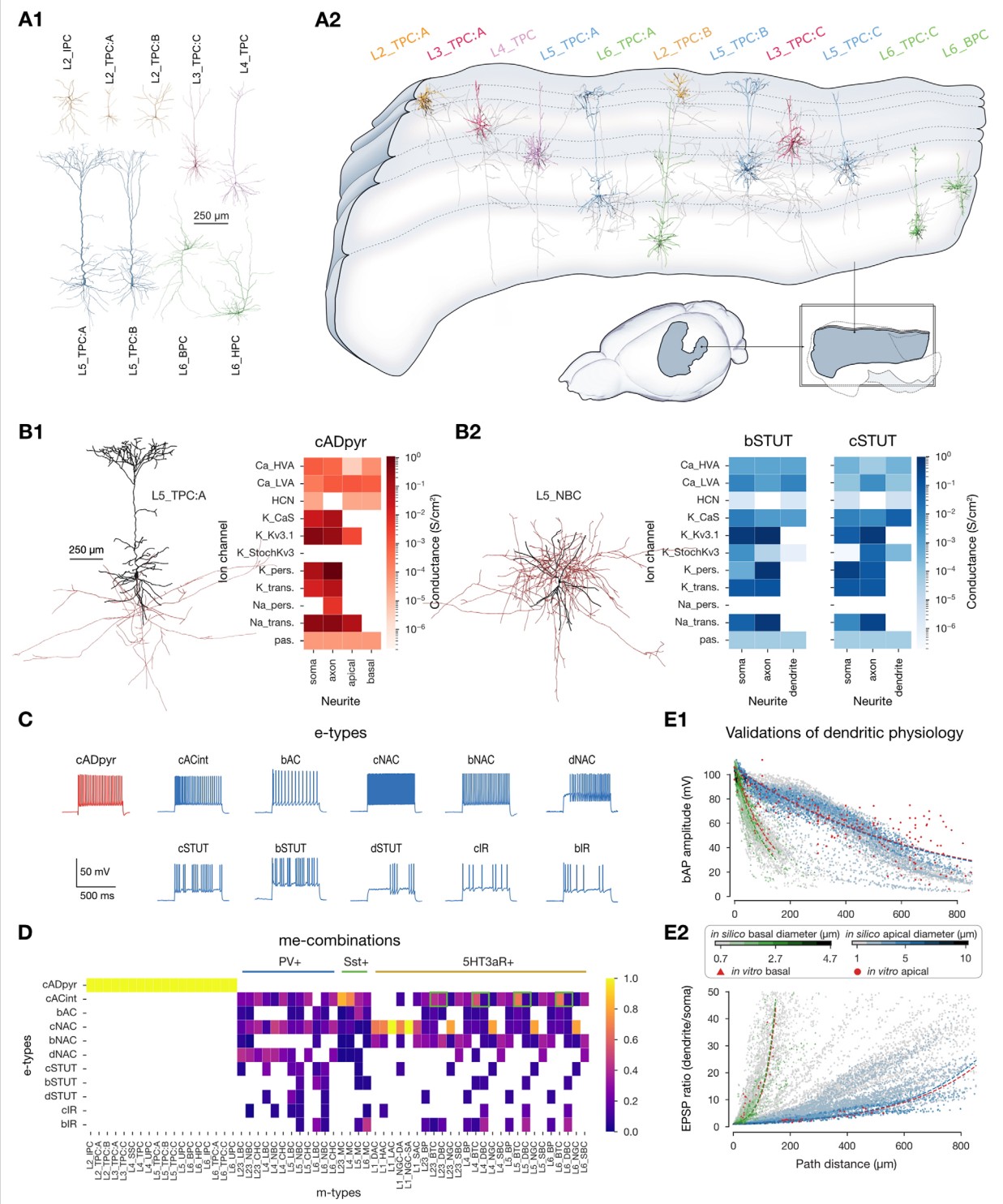

**Figure 2.** Modeling and validation of neuron physiology. (**A1**) Example excitatory neuron morphologies. See companion paper for exemplar morphologies of all morphological types. (**A2**) Example morphologies placed within the atlas-based volume. Axons shown in gray. (**B**) Optimized conductance densities for two exemplary e-types. (**B1**) cADpyr e-type on L5_TPC:A m-type; (**B2**) bSTUT and cSTUT e-types on L5_NBC m-type. (The L5_NBC m-type is combined with more e-types than the two shown, see panel **D**.) Morphologies were visualized with NeuroMorphoVis (**Abdellah et al., 2018**). Neurite diameters are enlarged (×3) for visibility. Soma and dendrites in black, axon in red. (**C**) e-types used in the model. As used in **Markram et al., 2015** and similar to the Petilla terminology **Ascoli et al., 2008**. (**D**) me-type composition. Heatmap shows the proportion of e-types for each m-type. Each me-type is assigned to one of the three I subpopulations. Assignments depending not only on m-type are highlighted by a green box. Particularly, for BTC and DBC m-types, the cACint e-type belongs to the Sst +subpopulation. (**E**) Validation of dendritic physiology of all L5 TTPCs. Panel

*Figure 2 continued on next page*

*Figure 2 continued*

reproduced from *Reva et al., 2023*. (**E1**) Validation of back-propagating action potential (bAP) amplitude (i.e., the dependence of bAP amplitude on distance from the soma) for basal (green) and apical (blue) dendrites. Reference data (in red) comes from *Stuart and Sakmann, 1994*; *Larkum et al., 2001* (apical) and *Nevian et al., 2007* (basal). Lines show exponential fits for the in silico (green and blue) and in vitro (red) data. Color bar indicates dendritic diameter. (**E2**) Validation of EPSP attenuation. Reference data (in red) comes from *Berger et al., 2001* (apical) and *Nevian et al., 2007* (basal). Lines and colorbar same as in (**E1**).

standardized protocols. Compared to *Markram et al., 2015*, electrical models were optimized and validated using (1) additional in vitro data, features, and protocols; (2) ion channel and electrophysiological data corrected for the liquid junction potential; and (3) stochastic channels (StochKv3) now including inactivation profiles. The methodology and resulting electrical models are described in *Reva et al., 2023* (see 'Methods'), and generated quantitatively more accurate electrical activity, including improved attenuation of excitatory postsynaptic potentials (EPSPs) and back-propagating action potentials. Electrical features included firing properties (e.g., spike frequency, inter-spike interval),

**Table 1.** Previously published modeling techniques and data to parameterize them that were combined in this work to model the physiology of nbS1 and conduct in silico experimentation.

**Data**

| Stage | Topic | Reference |
|---|---|---|
| Neuron physiology | E-type composition | *Markram et al., 2015*; *Markram et al., 2004* |
| | Electrophysiological recordings | *Markram et al., 2015* |
| | | *Larkum et al., 2001* |
| | | *Nevian et al., 2007* |
| | Ion-channel models | *Reva et al., 2023* |
| Synaptic physiology | Synaptic pathway physiology | *Markram et al., 2015* |
| | | *Barros-Zulaica et al., 2019* |
| | | *Gupta et al., 2000* |
| | | 13 additional sources in *Supplementary files 5–7* |
| In vivo-like activity | In vivo spont. firing rates | *Reyes-Puerta et al., 2015b* |
| | | *De Kock et al., 2007* |
| | In vivo evoked responses | *Reyes-Puerta et al., 2015b* |
| | | *Yu et al., 2019* |
| | In vivo thalamic spiking | *Diamond et al., 1992* |
| | | *Yu et al., 2019* |
| In silico experimentation | L4-L2/3 pathway | *Varani et al., 2022* |
| | Visual contrast | *Shapiro et al., 2022* |
| | Coding in inhibitory subpopulations | *Prince et al., 2021* |

**Modeling methods**

| Stage | Topic | Reference |
|---|---|---|
| Neuron physiology | E-model building | *Van Geit et al., 2016* |
| | | *Reva et al., 2023* |
| Synaptic physiology | Synapse model building | *Ecker et al., 2020* |
| | Model of multi-vesicular release | *Barros-Zulaica et al., 2019* |
| In vivo-like activity | Missing input compensation | *New, original methods based on |
| | | *Destexhe et al., 2001* |
| In silico experimentation | | *New, original methods |

action potential waveforms (e.g., fall and rise time, width), and passive properties (e.g., input resistance). The optimization was performed for a subset of neuron models. The resulting ion channel densities were generalized to other neuron models of the same e-type. For excitatory neurons such generalization was only made to neurons within the same layer. The resulting electrical activity of each neuron was validated against the corresponding electrical features, including the *characteristic* firing properties of the 11 e-types (*Figure 2C*). Model neurons with a mean zscore (over the electrical features) further than 2 standard deviations from the experimental mean were discarded. The new neuron models saw a fivefold improvement in generalizability compared to *Markram et al., 2015*; *Reva et al., 2023*.

For each of the 60 morphological types (m-types), the corresponding fractions of e-types were determined from experimental data as in *Markram et al., 2015*, resulting in 208 morpho-electrical types (me-types; *Figure 2D*). Correspondence with predominant expression of biological markers PV (parvalbumin), Sst (somatostatin), or 5HT3aR (serotonin receptor 3A) was determined based on me-type, as previously done in *Markram et al., 2015*; *Figure 2D*. Finally, for layer 5 thick-tufted pyramidal cell (L5 TTPC) morphologies, we found that dendritic electrical features, namely the attenuation of back propagating action potentials and EPSPs, reproduced experimental measurements (*Figure 2E*, *Reva et al., 2023*).

The required data to constrain region-specific me-type distributions was not available when building the model; to our knowledge, *Yao et al., 2023* is the first study that explored the question systematically (in mouse cortex). The atlas-based geometry does however impose heterogeneity in morphologies that e-types are paired with: only morphologies that are in line with the region-specific local curvature of the region are used throughout the volume. We demonstrate the effect of geometry on morphological composition in the companion paper.

## Improved modeling and validation of synaptic physiology

The biological realism of synaptic physiology was improved relative to *Markram et al., 2015* using additional data sources and by extending the stochastic version of the Tsodyks–Markram model (*Tsodyks and Markram, 1997*; *Markram et al., 1998*; *Fuhrmann et al., 2002*; *Loebel et al., 2009*) to feature multivesicular release, which in turn improved the accuracy of the coefficient of variations (CV; std/mean) of postsynaptic potentials (PSPs) as described in *Barros-Zulaica et al., 2019* and *Ecker et al., 2020*. The model assumes a pool of available vesicles that is utilized by a presynaptic action potential, with a release probability dependent on the extracellular calcium concentration ($[Ca^{2+}]_o$; *Ohana and Sakmann, 1998*; *Rozov et al., 2001*; *Borst, 2010*). Additionally, single vesicles spontaneously release as an additional source of variability with a low frequency (with improved calibration relative to *Markram et al., 2015*). The utilization of vesicles leads to a postsynaptic conductance with bi-exponential kinetics. Short-term plasticity (STP) dynamics in response to sustained presynaptic activation are either facilitating (E1/I1), depressing (E2/I2), or pseudo-linear (I3). E synaptic currents consist of both AMPA and NMDA components, whilst I currents consist of a single $GABA_A$ component, except for neurogliaform cells, whose synapses also feature a slow $GABA_B$ component. The NMDA component of E synaptic currents depends on the state of the $Mg^{2+}$ block (*Jahr and Stevens, 1990*), with the improved fitting of parameters to cortical recordings from *Vargas-Caballero and Robinson, 2003* by *Chindemi et al., 2022*.

The workflow for determining a dense parameter set for all synaptic pathways, starting with sparse data from the literature, is described for the use case of hippocampal CA1 in *Ecker et al., 2020* and briefly in the 'Methods'. We combined data sources used in *Markram et al., 2015*, with a large number of recent data sources (*Qi and Feldmeyer, 2016*; *Barros-Zulaica et al., 2019*; *Yang et al., 2020*; *Yang et al., 2022*). The resulting pathway-specific parameters are listed in the tables of *Supplementary files 2–4*, the most common short-term dynamics are depicted in *Figure 3A1–2*, and the assignment of STP profiles to different pathways are shown in *Figure 3A3*. PSP amplitudes and their CV closely matched their biological counterparts ($r = 0.99$, $n = 27$; *Figure 3B1*; table in *Supplementary file 5* and $r = 0.63$, $n = 10$; *Figure 3C1*; table in *Supplementary file 6*, respectively). The dense parameter set also allowed prediction of PSP amplitudes and CVs for all cortical pathways (*Figure 3B2, C2*). The frequency of miniature postsynaptic currents (mPSCs) were also in line with in vitro measurements ($r = 0.92$, $n = 5$; *Figure 3D*; table in *Supplementary file 7*).

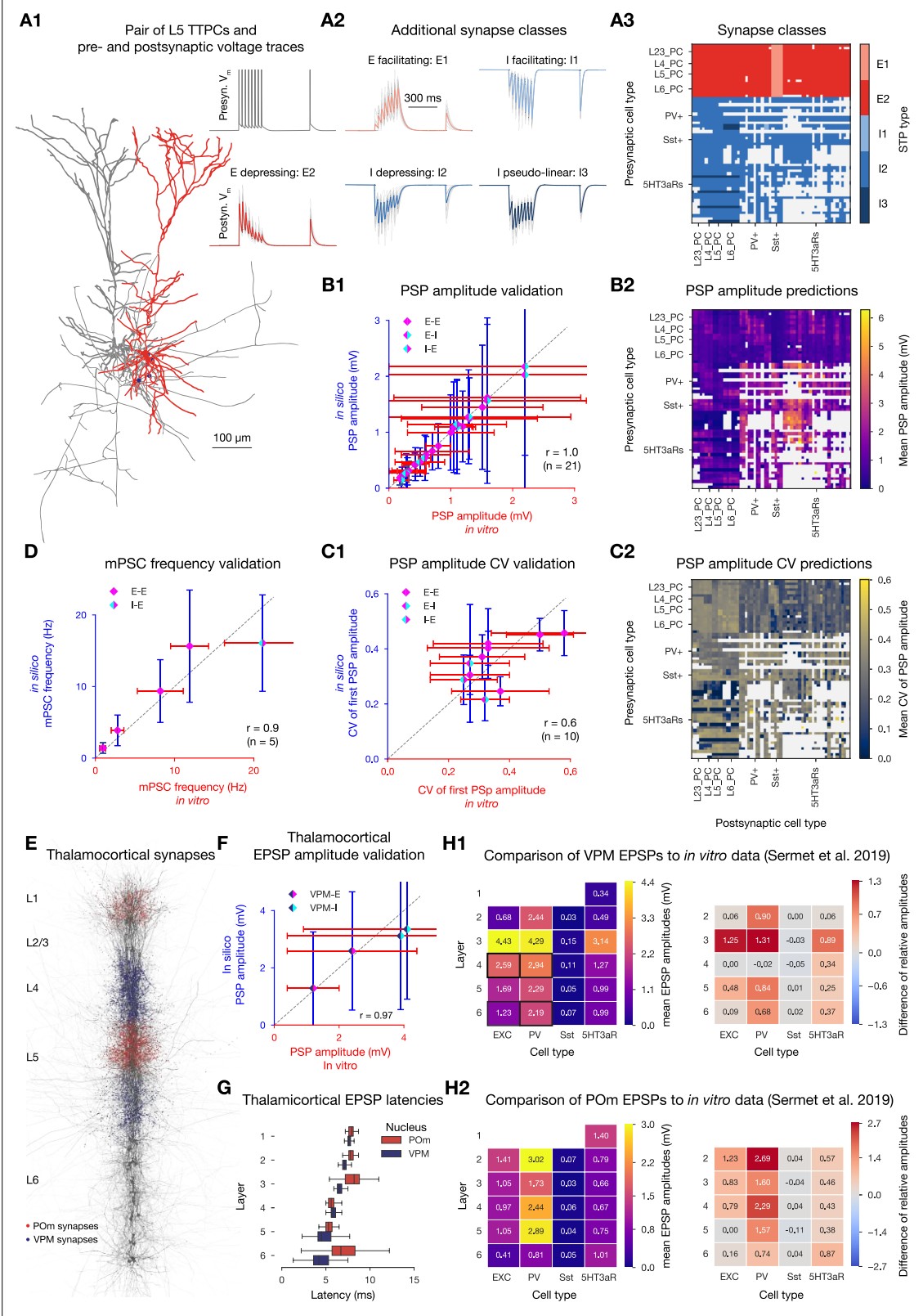

**Figure 3.** Modeling and validation of synaptic physiology. (**A1**) Exemplary pair of L5 TTPCs (visualized with NeuroMorphoVis; *Abdellah et al., 2018*). Presynaptic cell in gray, postsynaptic cell in red, synapses between them in purple. Neurite diameters are enlarged (x3) for visibility and axons were cut to fit into the figure. Pre- and postsynaptic voltage traces on the top right. (**A2**) Exemplary postsynaptic traces with different STP profiles. (**A3**) Assignment of STP profiles to viable pathways. (Pathways were considered viable if there were at least 10 connections in all eight subregions of the

*Figure 3 continued on next page*

*Figure 3 continued*

model.) (**B1**) Validation of first PSP amplitudes (see also table in *Supplementary file 5*). Dashed gray line represents perfect correlation between experimental and model values. Error bars show 1 standard deviation (also for **C1, D, F**). (**B2**) Predicted PSP amplitudes of all viable pathways in the circuit. Postsynaptic cells were held at –70 mV using an in silico voltage-clamp. Means were calculated over 100 pairs of neurons with 35 repetitions each. (**C1, C2**) same as (**B1**) and (**B2**), but showing the CV of the first PSP amplitude (corresponding table is in *Supplementary file 6*). (**D**) Validation of mPSC frequencies (see also table in *Supplementary file 7*). (**E**) Location of synapses from VPM fibers (purple) and POm fibers (red) on 38 neurons (dark gray) in a 5 µm radius column (visualized with `BioExplorer`). (**F**) Validation of thalamocortical EPSP amplitudes as in (**B1**). The four pathways used for the validation are marked with a black rectangle on H1 to its right. (**G**) EPSP latencies (time from presynaptic spike to the rise to 5% of peak EPSP amplitude in the postsynaptic trace). (**H1**) Left: mean VPM evoked EPSP amplitudes on postsynaptic cell types (over 50 pairs). Right: comparison of normalized in silico amplitudes (normalized by L4 E as in *Sermet et al., 2019*) to in vitro reference data from *Sermet et al., 2019*. Heatmap shows model minus reference values, thus positive values indicate a higher normalized EPSP amplitude in our model than in the reference experimental dataset. (**H2**) same as (**H1**) but for POm (normalized by L5 E as in *Sermet et al., 2019*).

## Improved modeling and validation of thalamocortical projections

The anatomical model includes fibers from the thalamus, based on fibers projecting to the barrel cortex from the ventral posteriomedial (VPM) and posteriormedial (POm) thalamic nuclei. These fibers make synaptic contacts within a radius of the fiber probabilistically based on laminar innervation profiles (see *Reimann et al., 2024*; *Figure 3E*). Compared to the previous model (*Markram et al., 2015*), POm projections are added and the physiology of synapses from VPM improved (see 'Methods'; *Figure 3E, F*). Latencies of layer-wise EPSPs increase with distance from the thalamus (*Figure 3G*). Additionally, thalamocortical EPSP amplitudes normalized relative to a single population were compared to normalized EPSPs in response to optogenetic stimulation targeting bundles of thalamic fibers (*Sermet et al., 2019*). This provided contrasting insights, however. For example, whilst VPM to L6 I EPSPs match the initial validation data (*Figure 3F*), VPM to L6 PV+ responses appear too strong relative to other populations (*Figure 3H1*). The results suggest that the model's POm to L5 E pathway is too weak, when compared to other POm to E and all POm to PV+ pathways (*Figure 3H2*, right).

## Defining subvolumes and populations

To enable smaller simulations and targeted analyses, we defined standardized partitions comprising all neurons (and their connections) contained in a *subvolume* of the full nbS1 model. We decomposed the model into full-depth hexagonal prisms of a certain diameter that we call *hexagonal subvolumes*. These are slightly curved, following the geometry of the cortex, and have an intact layer structure. When the diameter of the hexagons is 520 $\mu m$, comparable to the size of the single cortical column model from *Markram et al., 2015*, we call these subvolumes *columns*. Taken together, a *central column* and the six columns surrounding it define a *seven-column subvolume* (*Figure 4A*). Additionally, when we separately analyze the E and I neurons in different layers, we refer to them as neuron *populations* of the model or a subvolume. Groups of I neurons that predominantly express either PV, Sst, or 5HT3aR markers are referred to as *inhibitory subpopulations*.

## Compensating for non-modeled brain regions
### Algorithm efficiently finds spectrum of network regimes compatible with in vivo whilst accounting for unknown in vivo recording bias

Initial simulations produced no activity, and it was necessary to compensate for missing excitatory input from neurons external to the model. We estimated that the number of missing synapses for the full nbS1 and seven hexagon subvolume as approximately two and seven times the number of internal synapses, based on *Oh et al., 2014*. We model the effect of these missing synaptic inputs explicitly as time-varying and statistically independent somatic conductance injections, using Ornstein–Uhlenbeck (OU) processes that mimic aggregated random background synaptic inputs (*Destexhe et al., 2001*; *Figure 4B1*; see also 'Discussion'). The mean ($OU_\mu$) and standard deviation ($OU_\sigma$) are defined as a percentage of an individual cell's input conductance at its resting membrane potential. The choice of $OU_\mu$ and $OU_\sigma$ for different populations and $[Ca^{2+}]_o$ (*Figure 4B2*) determines the model's emergent dynamics and mean firing rates (*Figure 4B3*). Initial simulations showed that using the same value of $OU_\mu$ and $OU_\sigma$ for all neurons makes some populations highly active whilst

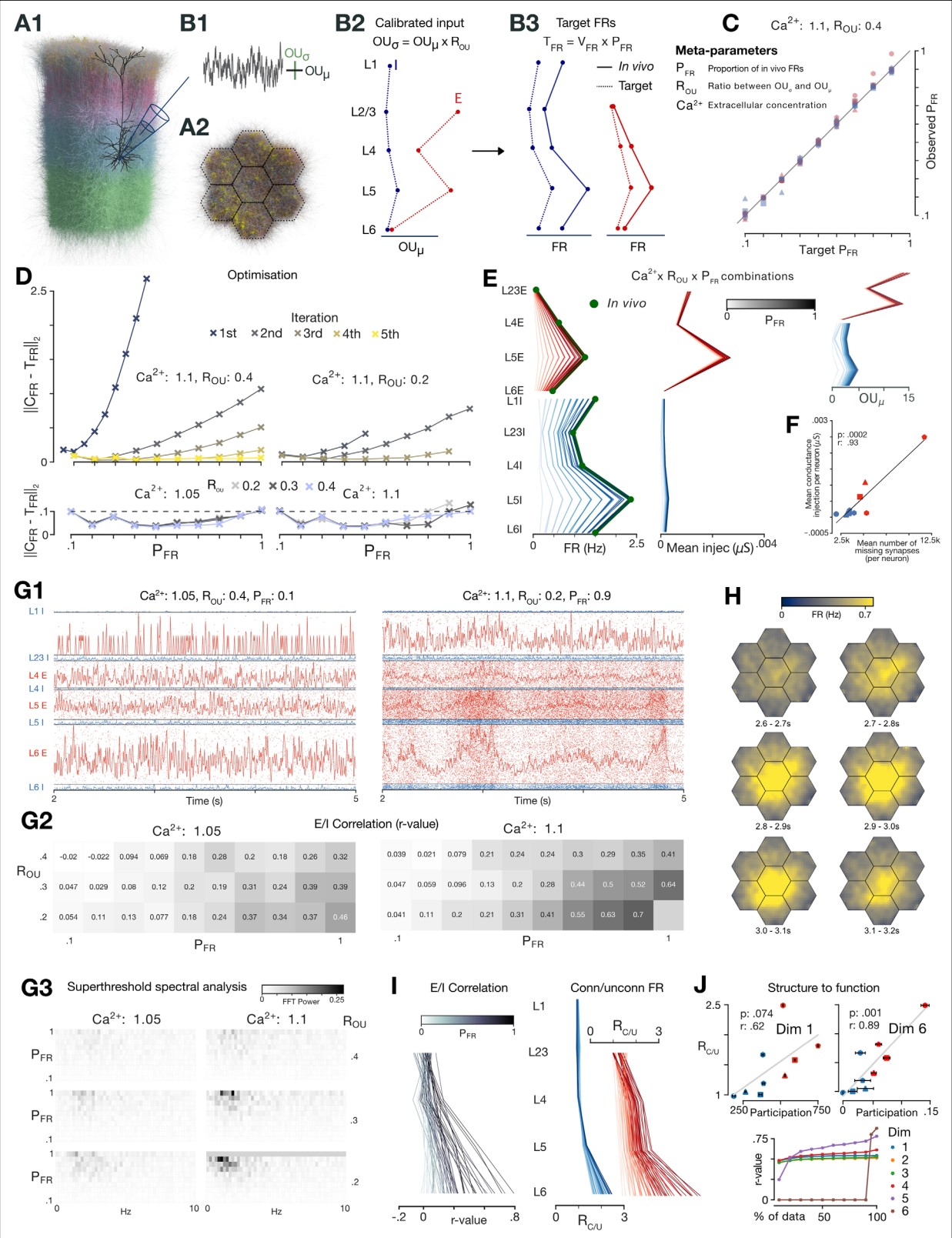

**Figure 4.** Spontaneous activity: calibration, in vivo-like dynamics, linking structure to function. (**A**) Seven-column subvolume. (**B**) $OU_\mu$ and $OU_\sigma$ of somatic conductance injection (**B1**) parameterized by population (**B2**) determine FRs (**B3**). Target FRs equal to in vivo firing rates multiplied by $P_{FR}$. Inset: table summarizing meta-parameters. (**C**) Target $P_{FR}$ (x-axis) plotted against the observed $P_{FR}$ (y-axis) for each E (red) and I (blue) population after calibration. The number of sides of the shape indicates the layer (triangle used for L2/3). (**D**) Euclidean distance between target and observed

*Figure 4 continued on next page*

*Figure 4 continued*

$P_{FR}$ values (over populations) decreases over iterations for two meta-parameter combinations (upper). Values after final iteration shown for all meta-parameter combinations (lower; dashed line shows termination condition). (**E**) (left) Firing rates, (center) mean conductance injection, and (right) $OU_\mu$ by population for the non-bursting simulations (one line per simulation; E and I values separated). (**F**) Estimated mean number of missing number synapses per neuron by population vs. the mean conductance injection, averaged over all combinations, for each population. Markers as in (**C**). (**G1**) Max normalized histograms for two meta-parameter combinations (5 ms bin size, 1σ Gaussian smoothing). (**G2**) Effect of meta-parameters on correlation between E and I histograms (5 ms bin size, 1σ Gaussian smoothing; central column). (**G3**) Fourier analysis of spontaneous activity for the 59 non-bursting simulations. (**H**) Activity in flat space over seven consecutive 100 ms windows. Activity smoothed (Gaussian kernel, $\sigma = 1 pixel$). (**I**) Left: correlation r-value between the histograms of layer-wise E and I populations for the 59 non-bursting simulations. Right: $R_{C/U}$ by population for the 59 non-bursting simulations. (**J**) Top: the mean *node participation* of E and I populations in each layer for dimensions 1 (left) and 6 (right) vs. $R_{C/U}$ for the parameter combination $[Ca^{2+}]_o = 1.05$ mM, $R_{OU} = 0.4$, and $P_{FR} = 0.3$. Markers as in (**C**). Line shows linear fit. Bottom: correlation r-values between mean node participation and $R_{C/U}$ when neurons with highest node participation are not included in the calculation of mean node participation (via a sliding threshold).

The online version of this article includes the following video and figure supplement(s) for figure 4:

**Figure supplement 1.** Spontaneous activity calibration: estimating $\chi$ and $\phi$.

**Figure supplement 2.** Mean dendritic length and missing synapses by population.

**Figure supplement 3.** Spontaneous activity firing rate distributions.

**Figure supplement 4.** Firing rate statistics.

**Figure supplement 5.** Over-/under-expression of neurons that maximize node participation.

**Figure 4—video 1.** Layer-wise E and I population rasters and max-normalized histograms of spontaneous activity for the central column of the seven-column subvolume for each of the 60 meta-parameter combinations.
https://elifesciences.org/articles/99693/figures#fig4video1

**Figure 4—video 2.** Visualization of spontaneous activity for the seven-column subvolume for the parameter combination $Ca^{2+} = 1.1$ mM, $P_{FR} = 0.9$, $R_{OU} = 0.2$ after collapsing activity to flatspace (activity binned and smoothed).
https://elifesciences.org/articles/99693/figures#fig4video2

others are silent or produces network-wide bursts. Finding population-specific $OU$ parameters which produce stable in vivo-like activity is challenging, however, due to the non-linearity and computational cost of simulations.

We developed an algorithm that efficiently calibrates population-specific $OU$ parameters for different values of $[Ca^{2+}]_o$, which affects the strength and reliability of synaptic transmission (see 'Synaptic physiology'). Additionally, it fixes the ratio $OU_\sigma/OU_\mu$, denoted $R_{OU}$, across all neurons to specific values representing the amount of noise in the extrinsic inputs (*Figure 4B2*). As extracellularly derived firing rates are known to be severely overestimated to an unknown degree (*Olshausen and Field, 2006*; *Wohrer et al., 2013*; *Buzsáki and Mizuseki, 2014*), the algorithm finds parameters which produce realistic inter-population firing rate ratios at different global levels of activity (*Figure 4B3*). Specifically, to account for uncertainty in the firing rate bias during spontaneous activity from extracellular spike sorted recordings it targets mean spontaneous firing rates of neuron populations that are a constant fraction $P_{FR}$ of in vivo reference values, for 10 $P_{FR}$ values between 0.1 and 1. We refer to $[Ca^{2+}]_o$, $R_{OU}$ and $P_{FR}$ as *meta-parameters*. The initial calibration was made for the seven hexagonal subvolume, and later generalized to the full nbS1 model.

The algorithm was first run for $[Ca^{2+}]_o = 1.1$ mM, $R_{OU} = 0.4$. Target firing rates for the 10 $P_{FR}$ values were reached (*Figure 4C*) after five iterations of 10 simulations of 6.5 seconds (*Figure 4D*, left upper). Only three iterations were needed when calibration began using previously calibrated parameters for different $R_{OU}$ and $[Ca^{2+}]_o$ (*Figure 4D*, right upper). This allowed us to parameterize extrinsic input for a range of combinations of the three *meta-parameters* (*Figure 4D*, lower and E, left) within biologically relevant ranges ($[Ca^{2+}]_o$: 1.05–1.1 mM; *Jones and Keep, 1988*; *Massimini and Amzica, 2001*; *Amzica et al., 2002*; *Gonzalez et al., 2022*; $P_{FR}$: 0.1–1.0; $R_{OU}$: 0.2–0.4; *Destexhe et al., 2001*). We found that the resulting mean conductance injections were highly population-specific (*Figure 4E*, center). The range of $OU_\mu$ values between populations was on average 5.2 times higher than the range of $OU_\mu$ values for a single population across meta-parameter combinations (*Figure 4E*, right). This was expected, as the missing mid-range innervation is known to have specific and heterogeneous laminar profiles (*Felleman and Van Essen, 1991*; *Harris et al., 2019*; *Gao et al., 2022*).

## Population-wise input compensation correlates with estimated number of missing synapses

We estimated the number of missing synapses per neuron assuming a total density of E synapses of 1.1 synapses/μm, based on mean spine densities (*Larkman, 1991*; *Datwani et al., 2002*; *Kawaguchi et al., 2006*), and subtracting the number of synapses present in the model (*Figure 4—figure supplement 2*). We confirmed a strong correlation between this measure and required conductance injection (*Figure 4F*). Heterogeneity in synaptic density within and across neuron classes and sections makes estimating the number of missing synapses challenging (*De Felipe and Fariñas, 1992*). Changing the assumed synaptic density value of 1.1 synapses/μm would only change the slope of the relationship, however. Estimates of mean number of existing and missing synapses per population were within reasonable ranges; even the larger estimate for L5 E (due to higher dendritic length; *Figure 4—figure supplement 2*) was within biological estimates of 13,000±3500 total afferent synapses (*De Felipe and Fariñas, 1992*).

## **In vivo-like spontaneous activity**

### $P_{FR}$, $R_{OU}$, and $[Ca^{2+}]_o$ determine the properties of correlative spontaneous activity

Rasters and histograms of spontaneous activity for two meta-parameter combinations are shown in *Figure 4G1* (and for all combinations in *Figure 4—video 1*). The first simulation uses a higher $R_{OU}$, lower $P_{FR}$, and lower synaptic transmission reliability ($[Ca^{2+}]_o$). In the second simulation, we found larger fluctuations, correlated between E and I populations across layers. Overall, increasing $[Ca^{2+}]_o$ and $P_{FR}$, and decreasing $R_{OU}$, increases the amplitude of correlated fluctuations (*Figure 4—video 1*, *Figure 4G2*) and increases power at lower frequencies (*Figure 4G3*). This supports the notion that decreasing $R_{OU}$ or increasing $[Ca^{2+}]_o$ shifts the model along a spectrum from externally (noise) driven to internally driven correlated activity. Note that the most correlated meta-parameter combination $[Ca^{2+}]_o$: 1.1 mM, $R_{OU}$: 0.2, $P_{FR}$: 1.0 produced network-wide 'bursting' activity, which we define as highly synchronous all or nothing events (*Figure 4—video 1*). Such activity, which may be characteristic of epileptic activity, can be studied with the model but is not the focus of this study.

We reemphasize that the $[Ca^{2+}]_o$, $R_{OU}$, and $P_{FR}$ meta-parameters account for uncertainty of in vivo extracellular calcium concentration, the nature of inputs from other brain regions and the bias of extracellularly recorded firing rates. Whilst estimates for $[Ca^{2+}]_o$ are between 1.0 and 1.1 mM (*Jones and Keep, 1988*; *Massimini and Amzica, 2001*; *Amzica et al., 2002*; *Gonzalez et al., 2022*) and estimates for $P_{FR}$ are in the range of 0.1–0.3 (*Olshausen and Field, 2006*), combinations of these parameters supporting in vivo-like stimulus-responses in later sections will offer a prediction for the true values of these parameters. Both these later results and our recent analysis of spike sorting bias using this model (*Laquitaine et al., 2024*) predict a spike sorting bias corresponding to $P_{FR} \sim 0.3$, confirming the prediction of *Olshausen and Field, 2006*.

### Long-tailed population firing rate distributions with means $\sim 1\,Hz$

To study the firing rate distributions of different subpopulations and m-types, we ran 50-second simulations for the meta-parameter combinations: $[Ca^{2+}]_o$: 1.05 mM, $R_{OU}$: 0.4, $P_{FR}$: 0.3, 0.7 (*Figure 4—figure supplement 3*). Different subpopulations showed different sparsity levels (proportion of neurons spiking at least once) ranging from 6.6 to 42.5%. *Wohrer et al., 2013* considered in detail the biases and challenges in obtaining ground truth firing rate distributions in vivo, and discuss the wide heterogeneity of reports in different modalities using different recording techniques. They conclude that most evidence points towards long-tailed distributions with peaks just below $1\,Hz$. We confirmed that spontaneous firing rate distributions were long-tailed (approximately lognormally distributed) with means on the order of $1\,Hz$ for most subpopulations. Importantly the layer-wise means were just below $1\,Hz$ in all layers for the $P_{FR} = 0.3$ meta-parameter combination. Moreover, our recent work applying spike sorting to extracellular activity using this meta-parameter combination found spike sorted firing rate distributions to be lognormally distributed and very similar to in vivo distributions obtained using the same probe geometry and spike sorter (*Laquitaine et al., 2024*).

## Spatially coordinated fluctuations and increase in correlative and recurrent activity from supra- to infragranular layers

In the horizontal dimensions, under higher correlation regimes, fluctuations are spatially coordinated and global within the central hexagon of the simulated subvolume (*Figure 4H*, *Figure 4—video 2*). In depth, the size of fluctuations and level of correlated activity increases from supragranular to infragranular layers (*Figure 4G1, I*, left, *Figure 4—video 1*), suggesting that activity in deeper layers of the model is more internally driven. We characterized the effect of recurrent connectivity by measuring the layer-specific decrease in firing rates when neurons in the model are disconnected from each other. As expected, we found that the effect (quantified by the ratio between connected and disconnected firing rates: $R_{C/U}$) increased from supra- to infragranular layers (*Figure 4I*, right).

## High-dimensional connectivity motifs predict layer-wise spontaneous activity

By analyzing the underlying network structure, we observed a corresponding gradient in the topology of intra-layer synaptic connectivity from supra- to infragranular layers. We calculated the mean *node participation* of neuron populations in various dimensions (see 'Methods'; *Figure 4J*, top). For dimension 1, this is simply the degree of a neuron; for dimensions above 1, this generalizes the notion to counting participation in dense, directed motifs of increasing size (*directed simplices*; *Reimann et al., 2017*). We found that correlations of this measure with the ratio of connected and disconnected firing rates increased with dimension (*Figure 4J*, bottom, 100% of data), indicating the importance of large directed simplices in shaping activity. Curiously, the correlations for higher dimensions were driven by a small number of neurons with very high node participation. This was indicated by a sharp drop in correlation when neurons above a given value were excluded (*Figure 4J*, bottom). Additionally, we studied the structural effect on the firing rate (here measured as the inverse of the inter-spike interval, ISI, which can be thought of as a proxy of non-zero firing rate). We found that for the connected circuit, the firing rate increases with simplex dimension; in contrast with the disconnected circuit, where this relationship remains flat (see *Figure 4—figure supplement 4*, red vs. blue curves and 'Methods').

This also demonstrates high variability between neurons, in line with biology, both structurally (*Towlson et al., 2013*; *Nigam et al., 2016*) and functionally (*Wohrer et al., 2013*; *Buzsáki and Mizuseki, 2014*). We next identified the cell types that are overexpressed in the group of neurons that have the 5% highest values of node participation across dimensions (*Figure 4—figure supplement 5*). This could inform theoretical point neuron models with cell-type specificity, for example. We found that while in dimension 1 (i.e., node degree) this consists mostly of inhibitory cells, in higher dimensions the cell types concentrate in layers 4–6, especially for TPC neurons. This is in line with our structural layer-wise findings in Figure 8B in *Reimann et al., 2024*.

## **Stimulus-responses reproduce in vivo dynamics with millisecond-scale precision**

### Recreating simple whisker deflection experiments

We compared stimulus-responses with in vivo barrel cortex responses to both *single whisker deflections* and *active whisker touches* under anaesthetized (*Reyes-Puerta et al., 2015b*) and awake states (*Yu et al., 2019*), respectively. While the model is of non-barrel somatosensory regions, this nevertheless provides validations of overall excitability and the laminar structure of responses reflecting general trends of cortical processing. We activated a percentage ($F_P$) of the VPM fibers projecting to the central column of the seven-column subvolume (*Figure 5A1*). For each selected thalamic fiber, spike times were drawn from VPM peristimulus time histogram (PSTH) recorded in vivo for the two stimulus types (*Figure 5A2*; 'Methods'). For both stimulus types, a single thalamic stimulus was presented 10 times at 1-second intervals at three intensities ($F_P : 5\%, 10\%, 15\%$).

### Validation of millisecond precise in vivo population responses and corresponding prediction of in vivo spontaneous firing rates

For the parameter combination $[Ca^{2+}]_o = 1.05$ mM, $R_{OU} = 0.4$, $P_{FR} = 0.3$ and $F_P = 10\%$, E and I populations in each layer show clear responses on each trial (*Figure 5B*; three trials shown), except for L1. *Figure 5—videos 1 and 2* show all simulated combinations of the four meta-parameters. Out of 72 parameter combinations, 21 passed an initial assessment of in vivo similarity to the in vivo data based

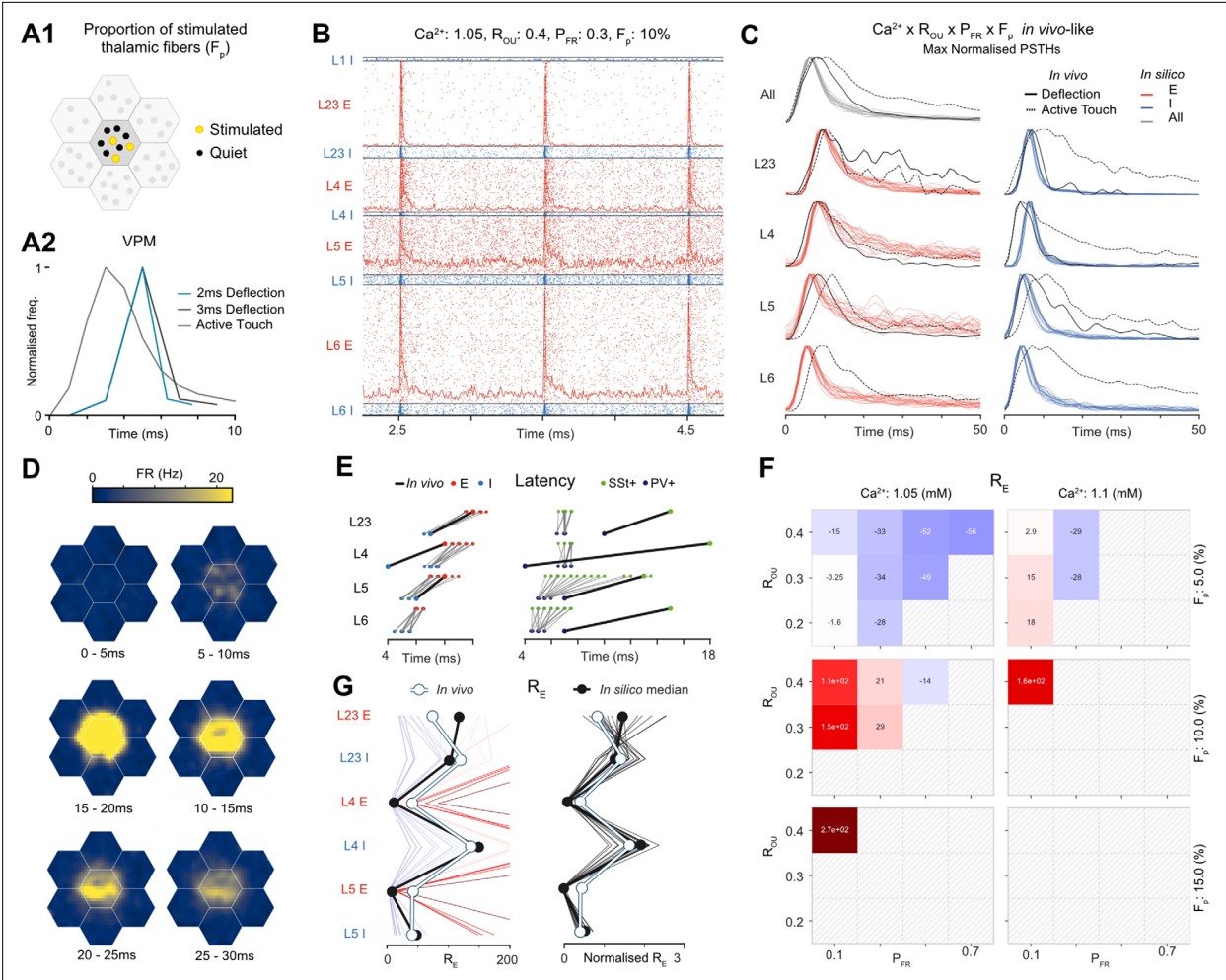

**Figure 5.** Layer-wise population responses to single whisker deflection and active whisker touch stimuli closely match in vivo millisecond dynamics and response amplitudes. (**A1**) Stimuli are simulated by activating a proportion ($F_P$) of fibers projecting to the central hexagon of the seven-column subvolume. (**A2**) Each fiber is assigned a spike time from a PSTH of VPM activity recorded in vivo for either a single whisker deflection or active whisker touch paradigm. (**B**) Spiking activity and histograms (baseline and max normalized) for each layer-wise E and I population for the parameter combination $[Ca^{2+}]_o = 1.05$ mM, $R_{OU} = 0.4$, $P_{FR} = 0.3$, and $F_P = 10\%$ for a 2.5-second section of the 10 whisker deflection test protocol. (**C**) Trial-averaged PSTHs (baseline and max normalized) in response to the single whisker deflection stimulus for all of the parameter combinations which passed the initial criteria, and the in vivo references. (**D**) Spatiotemporal evolution of the trial-averaged stimulus-response in flat space for the same parameter combination in (**B**). (**E**) Left: latencies of different layer-wise E and I populations for all criteria passing combinations in response to single whisker deflection stimuli. Right: latencies for inhibitory interneuron subpopulations in response to active whisker touch stimuli. Corresponding in vivo references are shown. (**F**) Heatmap showing the effect of the meta-parameters on the difference of $R_E$ for the entire central column from the corresponding in vivo reference. Parameter combinations which failed the criteria tests are masked. (**G**) Left: ratio $R_E$ between maximum evoked response and spontaneous baseline activity by population. One line per simulation. Simulation line color shows the difference of $R_E$ for the entire central column from the corresponding in vivo reference (color values shown in **G**). Thick black line shows median of in silico values. White shows in vivo reference. Right: normalized ratios of each simulation by dividing by $R_E$ for the entire central column.

The online version of this article includes the following video and figure supplement(s) for figure 5:

**Figure supplement 1.** Whisker deflection responses.

**Figure supplement 2.** Active whisker touch responses.

**Figure 5—video 1.** Layer-wise E and I population rasters and max-normalized histograms of evoked activity during the 10 single whisker deflection protocol for each of the simulated meta-parameter combinations.
https://elifesciences.org/articles/99693/figures#fig5video1

**Figure 5—video 2.** Trial-averaged max-normalized histograms of evoked activity during the 10 single whisker deflection protocol for each of the simulated meta-parameter combinations.
https://elifesciences.org/articles/99693/figures#fig5video2

*Figure 5 continued on next page*

*Figure 5 continued*

**Figure 5—video 3.** Visualization of trial-averaged response of the seven-column subvolume over the 10 single whisker deflection protocol for the parameter combination $Ca^{2+} = 1.1$ mM, $P_{FR} = 0.3$, $R_{OU} = 0.4$, $F_P = 20\%$ after collapsing activity to flatspace (activity binned and smoothed). https://elifesciences.org/articles/99693/figures#fig5video3

on latencies and decays (see 'Methods', *Figure 5C*, *Figure 5—figure supplement 1*). Responses remained localized to within 600 $\mu m$ of the stimulus location (*Figure 5D*, *Figure 5—video 3*). In vivo, layer-wise I populations precede the corresponding E populations and response latencies increase with distance from thalamus, except for the fast response of L4 I. The latencies and decays are matched by criteria passing simulations (*Figure 5C, E*, left), except for a longer L4 I latency, and shorter and longer sustained responses of L2/3 E and L4 E. PV+ and Sst+ subpopulations respond reliably in layers 2–6, but the 5HT3aR+ subpopulation responses are weaker and less reliable (*Figure 5—figure supplement 1*). Under the active whisker touch paradigm, PV+ latencies precede Sst+ latencies in vivo, and either precede or are simultaneous with Sst+ latencies in silico (*Figure 5E*, right).

The passing parameter combinations form a contiguous region in the parameter space (*Figure 5F*), characterized by combinations of low $P_{FR}$, $[Ca^{2+}]_o$ and $F_P$ with higher $R_{OU}$, which displayed weakly correlated spontaneous activity (compare *Figure 5F* with *Figure 4G2*). This offers a prediction that true in vivo firing rates are within the range of these lower FRs. Specifically, simulations with $P_{FR}$ from 0.1 to 0.5 robustly support realistic stimulus-responses, with the middle of this range (0.3) corresponding with estimates of in vivo recording bias; both the previous estimates of *Olshausen and Field, 2006* and from a spike sorting study using this model (*Laquitaine et al., 2024*). The remaining results apply to the criteria passing simulations. We characterized the magnitude of responses by $R_E$, the ratio between the peak of the evoked response and the pre-stimulus baseline activity level. Assuming that extracellular sampling biases are similar during spontaneous and evoked activity, $R_E$ provides a bias-free metric for comparing response magnitude with in vivo. Decreasing $R_{OU}$ and $P_{FR}$, and increasing $[Ca^{2+}]_o$ and $F_P$ led to a global increase of $R_E$, and results matched in vivo closely for a large part of the parameter space (*Figure 5F*).

Interestingly, the parameter combination closest to in vivo was central in the parameter space of criteria passing simulations (*Figure 5G*, left). Values of $R_E$, normalized by the mean $R_E$ across populations, were remarkably constant (*Figure 5G*, right), indicating that variation of the parameters provides a global scaling of response magnitude. Moreover, I populations matched the in vivo reference closely, while the responses of L2/3 E were slightly greater, and for L4 E and L5 E slightly weaker. The region of the parameter space where responses match in vivo for the active whisker touch stimulus was the same as for single whisker deflections (*Figure 5—figure supplement 2*). Latencies were also similar to in vivo, with the main discrepancy being slower responses in L4 (E and I). With respect to the meta-parameters, we found that the best match for the anesthetized state was characterized by lower firing rates ($P_{FR}$) and reduced external noise levels (lower $R_{OU}$) compared to the awake active whisker touch paradigm (white areas in *Figure 5F*, *Figure 5—figure supplement 2B*). These results fit expectation for anaesthetized and awake states respectively, and provide predictions of their spontaneous firing rates, noise levels and corresponding in silico regimes.

## Prediction of 2–9% population response sparsity

Response sparsity (i.e., the proportion of neurons spiking at least once following a stimulus) varied across populations and meta-parameter combinations, but was around the range of 10–20% reported in vivo in the barrel cortex (*Barth and Poulet, 2012*; *Figure 5—figure supplement 1D*). Interestingly, excitatory population responses were sparser than the corresponding inhibitory layer-wise populations across layers. The observed heterogeneous layer-wise population sparsities offer a prediction for in vivo. Moreover, the majority of responding cells spike only once to single whisker deflections (*Figure 5—figure supplement 1E*), as reported in vivo (*Isbister et al., 2021*).

## Validation and prediction through reproduction and extension of complex laboratory experiments using a single model parameterization

Building upon the in vivo-like activity states, we combined various simulation techniques to recreate laboratory experiments from the literature (see 'Methods'). For these experiments, we use the

parameter combination: $[Ca^{2+}]_o = 1.05$ mM, $R_{OU} = 0.4$, and $P_{FR} = 0.3$, as this combination supports realistic in vivo-like responses to the two stimulus types over a range of fiber percentages ($F_P$).

## Exploring the canonical model: Layer-wise contributions to L2/3 stimulus-responses

In the canonical model of the cortex (reviewed, e.g., in *Lübke and Feldmeyer, 2007*; *Feldmeyer, 2012*) information from the thalamus arrives to L4, then propagates to L2/3, from there to L5 (which serves as the main cortico-cortical output layer) and lastly to L6 (which projects back to the thalamus). The coordinated action of all layers of S1 is required for high-level behavioral tasks (*Park et al., 2020*). As the canonical model is based on the highest layer-wise density of axons, it cannot describe all interactions in the cortex. For example, VPM does not only innervate L4, but also the border of L5 and L6, as well as the bottom of L2/3 (*Meyer et al., 2010*; *Constantinople and Bruno, 2013*; *Sermet et al., 2019*).

## Recreating modulation of L2/3 PCs during optogenetic inactivation of L4 PCs during whisker deflection

To study how L4 contributes to the stimulus preference of L2/3 PCs, *Varani et al., 2022* used optogenetic inactivation of L4 PCs during whisker stimulation and quantified the changes in the subthreshold response of L2/3 PCs. They found that the early phase of the subthreshold response significantly differed from the control condition if the whisker was deflected in either the most or the least preferred direction (see the top and bottom rows of their Figure B,C). From this they concluded that both L4 and VPM contribute to the direction tuning of L2/3 PCs. After reproducing their experimental conditions in silico (*Figure 6A, B*, *Figure 6—figure supplement 2*; see 'Methods'), we confirmed that we can reproduce their results, that is, subthreshold responses of L2/3 PCs decreased, when L4 PCs were inhibited (*Figure 6C* for preferred direction whisker stimulation; see 'Methods').

## Prediction that L4 PCs enhance L2/3 PC responses directly, not through disynaptic inhibition

We then leveraged our in silico setup to study what *Varani et al., 2022* could not because of methodological limitations. In our reading, the authors aimed to test how direct E connections from L4 PCs to L2/3 PCs influence the stimulus representation in L2/3. This connection cannot specifically be blocked in vivo, instead (95% of) the L4 PC population is inhibited (as well as some lower L3 PCs). In our setup, we could selectively block the connection and found almost the same result (compare *Figure 6C and D*, left). This extends the conclusion of *Varani et al., 2022*: L4 PCs contribute to the stimulus preference of L2/3 PCs via direct E connections, and not via disynaptic inhibition.

## Prediction of the contribution of different layer-wise populations to L2/3 PC response

The authors also discussed studying L5 PCs' contribution to L2/3 responses (as a large fraction of L5 PC axons terminate in L2/3), but this is infeasible with current mouse lines. Leveraging our model, we found that L5 contributes much less to subthreshold L2/3 traces than L4 (*Figure 6D*, right). Extending to other presynaptic layers, we found that the contribution of L2/3 is similar to that of L4, whereas inputs from L6 are negligible (*Figure 6D*, right). Whilst mouse lines targeting L5 PCs might arrive soon (which could validate our prediction), blocking L2/3 connections between L2/3 cells without hyperpolarizing the same L2/3 population seems only achievable in silico.

## Recreation of contrast tuning modulation by optogenetic activation of PV+ interneurons

*Shapiro et al., 2022* compared the modulatory effects of optogenetic activation of interneuron subtypes on contrast tuning of neurons in V1. Whilst visual regions differ from somatosensory regions, the study provided an opportunity to explore the generality of interactions between E and I subpopulations. We therefore created a version of the study in silico, presenting spatio-temporally modulated patterns with various contrast levels through the VPM inputs (*Figure 6E, F*; see 'Methods).

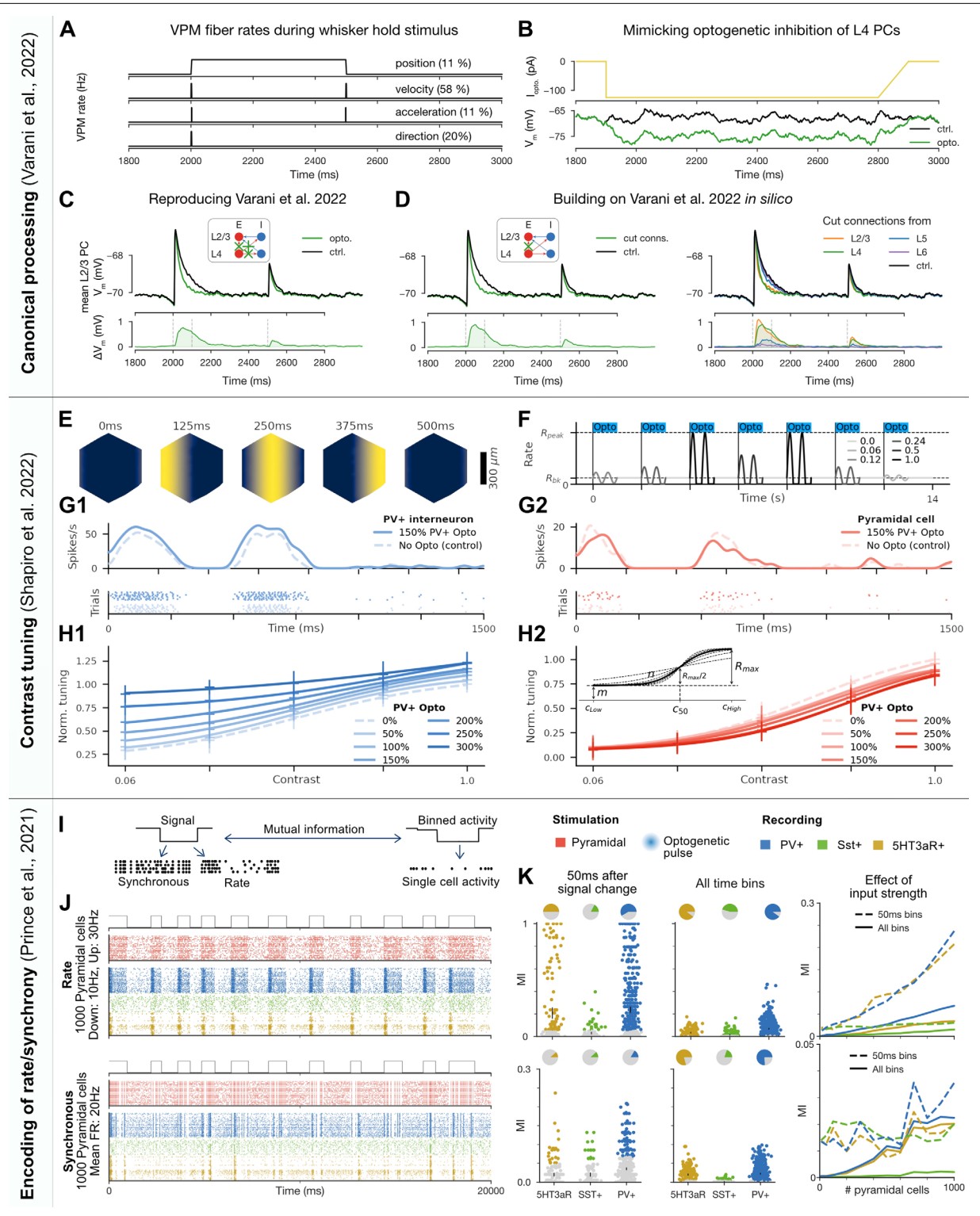

**Figure 6.** Reproducing and extending detailed experiments exploring the canonical model and encoding of contrast, rate-coded and synchronous information. First experiment: reproducing the effect of optogenetic inactivation of L4 pyramidal cells on L2/3 stimulus-responses and predicting the role of other pathways through circuit lesions. (**A**) Schematics of whisker kinetics and VPM fiber rates during 500-ms-long whisker hold stimulus. Fraction of VPM fibers coding for each kinetic feature are taken from *Petersen et al., 2008*. (**B**) Mimicking the effect of activation of the Halo inhibitory opsin in silico. Injected somatic hyperpolarizing current mimicking opsin activation (top), and the resulting somatic voltage trace from a combination of injected conductance, current, and synaptic PSPs from the network (bottom). (**C**) Comparison of average traces from selected L2/3 PCs (see 'Methods')

*Figure 6 continued on next page*

*Figure 6 continued*

in control (black) and optogenetically inhibited (green) conditions. (**D**) Same as (**C**), but instead of mimicking the optogenetic inhibition of L4 PCs, only the connections to L2/3 PCs are 'cut' (compare inset with the one in **C**). The right part depicts connections systematically cut from PCs in all layers, while the left shows L4 only for a better visual comparison with the conditions of *Varani et al., 2022* in (**C**). Second experiment: contrast tuning modulation by optogenetic activation of PV+ interneurons. (**E**) Spatial distribution of firing rates of VPM fibers projecting to a single simulated column at different points in time, corresponding to linear drifting gratings with a temporal frequency of $f_{temp} = 2$ Hz and a spatial frequency of $f_{spat} = 0.001$ cycles/extμm. (**F**) Firing rate signal of a single VPM fiber corresponding to a random sequence of drifting grating patterns with different contrast levels $C$, as indicated by the legend. All optogenetic stimuli targeting PV+ (or Sst+) interneurons were completely overlapping with the grating stimulus intervals, as indicated by the blue bars. (**G1**) Spiking activity and PSTH of an exemplary PV+ interneuron over 10 repetitions (trials) in response to a grating stimulus at maximum contrast ($C = 1.0$) without (control) and with medium strength (150%) optogenetic stimulation. (**H1**) Contrast tuning modulation of an exemplary PV+ interneuron by different levels of PV+ optogenetic stimulation, ranging from 0% (control) to 300%. The individual markers denote actual data points (mean ± SD over 10 repetitions, normalized to baseline response at maximum contrast) while curves illustrate sigmoidal fits to these data points. (**G2/H2**) Same as (**C1/D1**), but for exemplary PCs. (**H2**) inset: Illustration of sigmoidal parameters $m$, $n$, $R_{max}$, and $c_{50}$. Third experiment: encoding of synchronous and rate-coded signals by interneuron subtypes. (**I**) A binary signal is encoded either through changes in the rate or synchronicity of optogenetic pulses applied to a set of PCs. We define a direct correspondence between single optogenetic pulses and single spikes. Spiking activity is measured in PV+, Sst+, and 5HT3aR+ cells, and the mutual information between the binned activity of individual cells and the original binary signal is measured. (**J**) Visualization of the rate (upper) and synchronous (lower) coded stimulus experiments stimulating 1000 PCs, showing the binary signal (top), input neuron spike trains for 40 neurons (middle), and responses of the three L2/3 neuron types (bottom). (**K**) Upper: results for the rate coded stimulus experiment. Left: mutual information between spiking activity and the binary signal (one point for each cell that spiked at least once). Only activity in the 50 ms following the change of the binary signal is considered. Cells with mutual information not significantly different from that of a shuffled control are colored gray. Center: same as left but considering all time bins. Right: the effect of the number of stimulus neurons on the mean single cell mutual information for each subpopulation. Lower: the same as upper but for the synchronous stimulus type.

The online version of this article includes the following figure supplement(s) for figure 6:

**Figure supplement 1.** In silico experimental methods.

**Figure supplement 2.** Additional validations: reproducing the experiments of *Varani et al., 2022*.

**Figure supplement 3.** Calibration of the drifting grating stimulus based on peak firing rates.

**Figure supplement 4.** Contrast tuning by layer and paradoxical effect of optogenetic PV+ stimulation.

**Figure supplement 5.** Sigmoidal parameter fits to contrast tuning functions.

**Figure supplement 6.** Modeling the effects of optogenetic stimulation of interneurons.

Overall, we found within the single simulated column 228 PCs and 259 PV+ interneurons with robust contrast tuning of their responses (see 'Methods'). Within spike detection range of a vertically penetrating extracellular electrode (50 μm; *Henze et al., 2000*; *Buzsáki, 2004*), we found 15 PCs and 13 PV+ interneurons with robust tuning, which is roughly comparable to the original study (average of 8.2 tuned PCs and 3.0 PV +neurons per mouse *Shapiro et al., 2022*). Unlike the original study we found no Sst+ neurons with robust tuning.

Additionally, we found the following results, all in line with *Shapiro et al., 2022*: The firing rates of both PCs and PV+ neurons were affected by optogenetic activation of the PV+ subpopulation (*Figure 6G*), resulting in changes to the contrast tuning curves of these populations (*Figure 6H*, *Figure 6—figure supplement 4A*). The main effect of the optogenetic manipulation was an increase in contrast tuning for PV+ and a decrease for PCs. However, at high contrasts, we observed a paradoxical effect of the optogenetic stimulation on L6 PV+ neurons, reducing their activity with increasing stimulation strength (*Figure 6—figure supplement 4B*; *Mahrach et al., 2020*). This effect did not occur under gray screen conditions (i.e., at contrast 0.0) with a constant background firing rate of 0.2 Hz or 5 Hz, respectively (not shown). The individual tuning curves could be accurately fit by sigmoidal functions ($r^2_{PV+} = 0.995 \pm 0.008$), and consequently the effect of optogenetic activation could be quantified in terms of changes to the fitted parameters (*Figure 6H2* inset, *Figure 6—figure supplement 5*; 'Methods'). We found great variability between neurons with increasing trends for parameter $m$ in PV+ and $c_{50}$ in PCs, and decreasing trends for $R_{max}$ in PV+ and both $m$ and $R_{max}$ in PCs. Unlike in *Shapiro et al., 2022*, we found in addition an increasing trend of $c_{50}$ also in PV+ interneurons (*Figure 6—figure supplement 5*).

Comparing three abstract mathematical models (*Shapiro et al., 2022*; 'Methods'), we found that the saturation additive model best captured the effect of the optogenetic manipulation for PV+ neurons (*Figure 6—figure supplement 6A, B1*), as in the reference study. The relative contributions of saturation and addition varied strongly between individual neurons (*Figure 6—figure supplement*

*6C1*). At low intensities of the optogenetic manipulation, this distribution matched qualitatively the results of *Shapiro et al., 2022*, but at higher intensities we observed a shift towards stronger saturation. Combining the saturation additive model with a simplified, population-level model of PC firing (*Shapiro et al., 2022*; 'Methods') provided a good description of the observations (*Figure 6—figure supplement 6B2*) with relative contributions of saturation and addition that qualitatively match the reference (*Figure 6—figure supplement 6C2*).

## Encoding of synchronous and rate-coded signals by inhibitory subpopulations

*Prince et al., 2021* studied how different I subpopulations represent signals that are encoded as changes in the firing rate or synchronicity of PCs (*Figure 6I*). We repeated their study in silico (*Figure 6J*), but using in vivo-like conditions as the authors have suggested in their 'Discussion'. The original study pointed out the importance of membrane properties and short-term dynamics of synaptic inputs onto the subpopulations in shaping the results. Note that the lower synaptic reliability in vivo weakens synaptic depression (*Borst, 2010*), and the ongoing background input leads to higher membrane conductance (*Destexhe et al., 2001*); therefore, our results are expected to differ substantially and provide an independent prediction for a different dynamic state. While in the original in vitro study stimulation of 10 PCs was sufficient to find a decodable signal, we expected a larger number to be required under our in vivo-like conditions due to the lower $[Ca^{2+}]_o$, and the higher level of noise (see 'Methods).

Indeed, we found little discernible mutual information in I spiking for the activation of 10 PCs, but that mutual information increased with the number of activated neurons (*Figure 6K*, right). For 1000 activated neurons each of the three I subpopulations in L2/3 showed clear but qualitatively unique modulation for both encoding schemes (*Figure 6J*). As in *Prince et al., 2021*, we found differences in the encoding capabilities of the I subpopulations (*Figure 6K*). We found strongest encoding in PV+ neurons for both schemes, and overall low encoding strength in Sst+ neurons. Further, we found overall low amounts of mutual information for the synchronous stimulus, which is in line with *Prince et al., 2021*, as they linked encoding of that stimulus to depressing synaptic dynamics, which would be weakened in vivo.

## Exploring the effect of inhibitory targeting trends observed in electron microscopy (EM)

### Additional targeting rules recreate quantitative EM-connectivity trends

We next explored how certain trends in inhibitory targeting observed in a recent mouse electron microscopic connectome (*Schneider-Mizell et al., 2025*; *Figure 7A*) affect spontaneous and evoked activity. Consequently, an alternative connectome which quantitatively recreates these trends has been built (*Reimann et al., 2024*). We refer to this as the Schneider–Mizell compatible connectome (SM-connectome). It ensures that PV+ neurons target the perisomatic region, VIP+ neurons target other inhibitory neurons, and cells in L1 connect predominantly monosynaptically. Consequently, inhibition onto E neurons is sparser and closer to the soma (*Figure 7A2*). After validating the presence of these trends, we characterized how this affects the dynamics of the spontaneous and evoked states (*Figure 7A3*). Note that individual synapses were recalibrated to preserve the pathway specific reference PSP amplitudes. As expected, the largest change was for PV+ neurons where the average number of vesicles in the release-ready pool $N_{RRP}$ was increased by 2.4 times.

### EM targeting trends increase inhibition of inhibitory populations enabling increased long-range excitation

Inspite of this, the change in connectivity affected the activity sufficiently that recalibration of missing synaptic compensation was needed to attain the target firing rates for the original meta-parameter combinations (*Figure 7—figure supplement 1*). Mean conductances onto I neurons had to be increased, indicating the efficacy of the newly introduced I-I targeting by VIP+ neurons (*Figure 7B1*). On the other hand, conductances for E neurons remained roughly the same, except for L6. Curiously, we found that the firing rates of I populations were reduced when the internal connectivity was activated compared to when it was inactive, further highlighting the efficacy of the I-I targeting

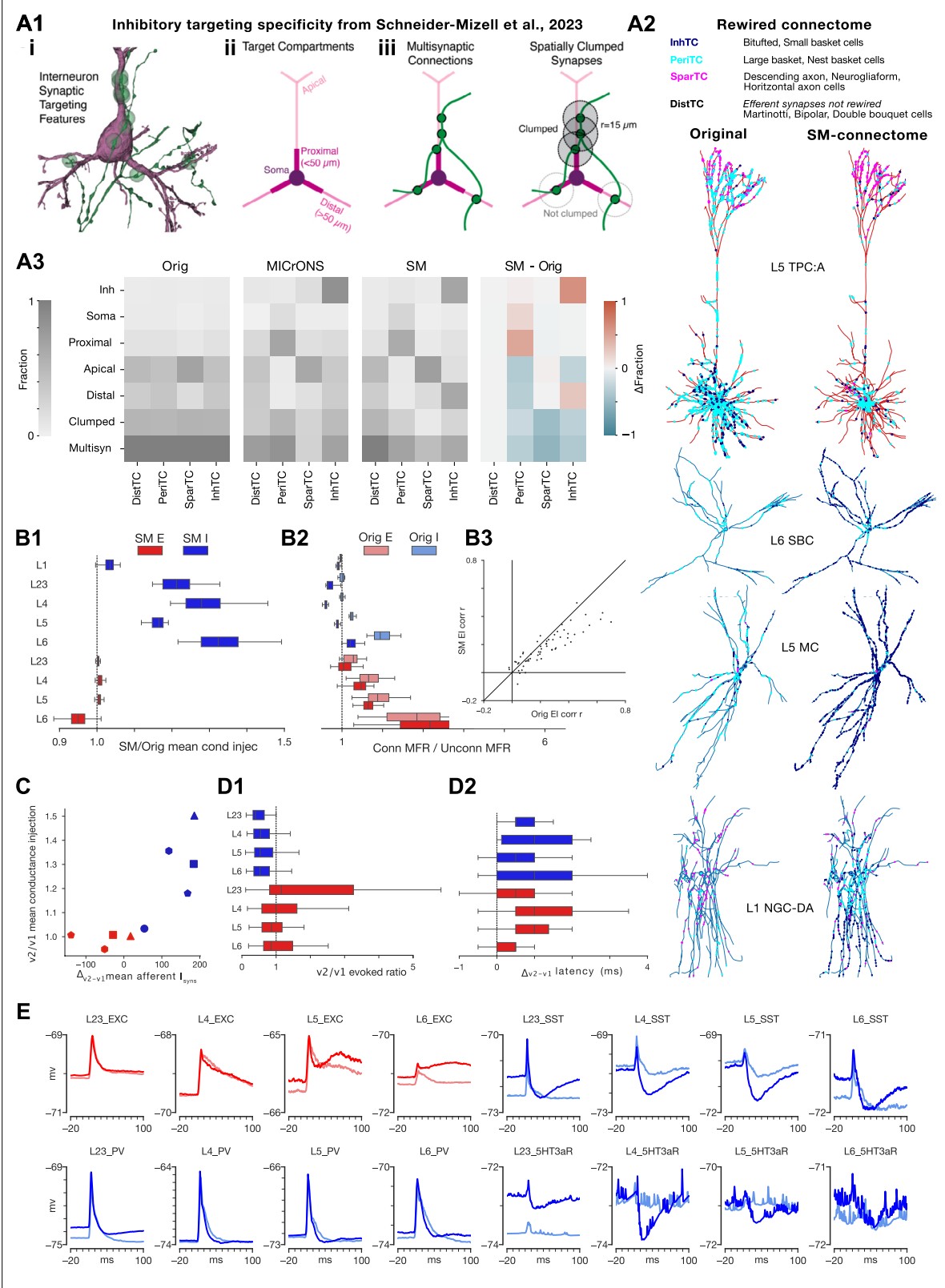

**Figure 7.** The effect of inhibitory targeting trends observed in electron microscopy on spontaneous and stimulus-evoked activity. (**A1**) Panel illustrating interneuron synaptic targeting features and caption taken from *Schneider-Mizell et al., 2025* (under CC-BY 4.0 license). (i) For determining inhibitory cell subclasses, connectivity properties were used such as an axon (green) making synapses (green dots) the perisomatic region of a target pyramidal cell (purple). (ii) Dendritic compartment definitions for excitatory neurons. (iii) Cartoon of a multisynaptic connection (left) and the synapses within

*Figure 7 continued on next page*

*Figure 7 continued*

the multisynaptic connection considered 'clumped' along the presynaptic axon (right). (**A2**) Exemplar neurons showing the synapse locations from different inhibitory m-type groupings in the original and SM connectome, respectively: distal targeting (DistTC: martinotti, bipolar, double bouquet cells), Perisomatic targeting (PeriTC: large basket, nest basket cells), Sparse targeting (SparTC: descending axon, neurogliaform, horizontal axon cells), Inhibitory targeting (InhTC: bitufted, small basket cells). (**A3**) Heatmaps showing the mean fraction of efferent synapses from the inhibitory m-type groupings onto inhibitory neurons, somas, proximal dendrites, apical dendrites, distal dendrites, or that are part of clumped or multisynaptic connections. Heatmaps show mean fractions (left to right) for the original, *Schneider-Mizell et al., 2025* characterization of the MICrONS dataset (*MICrONS Consortium, 2025*), SM-connectome, and the change between the original and SM-connectome. (**B1**) Proportional change in mean conductance injection between original and SM-connectome by population. (**B2**) Distribution of ratios between connected and disconnected firing rates ($R_{C/U}$) by population over non-bursting meta-parameter combinations for original and SM-connectome. (**B3**) Comparison of correlation (r-values) between E and I histograms (5 ms bin size, 1σ Gaussian smoothing; central column) between original and SM-connectome (each point represents a specific meta-parameter combination). (**C**) Scatter plot showing the change in the mean number of afferent inhibitory synapses onto each E (red) and I (blue) population after calibration versus the proportional change in mean conductance injection. The number of sides of the shape indicates the layer (triangle used for L2/3). (**D1**) Proportional change in the ratio between the peak of the evoked response and the pre-stimulus baseline activity level ($R_E$) between the original and SM-connectome. Plot shows distribution over meta-parameter combinations which passed the initial assessment of in vivo similarity to the in vivo data based on latencies and decays (see 'Methods'). (**D2**) Change in the latencies of populations peak responses between original and SM-connectome (distribution over same meta-parameter combinations shown in **D1**). (**E**) Mean membrane potential responses for layer-wise subpopulations for the meta-parameter combination $[Ca^{2+}]_o = 1.05$ mM, $R_{OU} = 0.4$, $P_{FR} = 0.3$, and $F_P = 10\%$. Responses averaged over 20 stimulus repetitions with an inter-trial-interval of 500 ms.

The online version of this article includes the following figure supplement(s) for figure 7:

**Figure supplement 1.** Re-calibration of SM-connectome.

(*Figure 7B2*, *Figure 7—figure supplement 1*). For E neurons in L2-5, activating internal connectivity increased firing rates, but less than in the original connectome. As the number of inhibitory synapses innervating them decreased (*Figure 7A2*), this indicates that the placement of PV +synapses closer to the soma increased the efficacy of inhibitory as expected. Taken together, this indicates that in the SM-connectome, I populations are more externally driven and more internally inhibited (as a function of proximity to L4), and L6 E more internally driven during spontaneous activity. Over the different meta-parameter combinations, correlation in the network decreased relative to the original circuit with some exceptions (*Figure 7B3*, *Figure 7—figure supplement 1*). This indicates that activity in the SM-connectome is shifted slightly towards a more asynchronous state.

By correlating the structural and functional changes we concluded that the increased conductance injection into the I populations can be explained by the increased inhibitory synapse count (*Figure 7C*). Conversely, the mostly decreased synapse counts for E populations did not lead to a decrease of conductance injection, further indicating that the more perisomatic positioning of PV+ synapses compensates for the reduced synapse counts.

## EM targeting trends affect population-wise stimulus-evoked response magnitudes and latencies

The lack of effect onto most of the E populations disappeared in the evoked state (*Figure 7D*). Overall, responses increase strongly in L23 and slightly decreased in L5 and L6 (*Figure 7D1*), while latencies of responses increased in every layer (*Figure 7D2*), suggesting different roles of the more specific inhibitory targeting in different layers.

## EM targeting trends hyperpolarize Sst+ and HT3aR+ late response, and disinhibit L5/6 E

Studying somatic membrane potentials for different subpopulations in response to whisker deflections shows that PV+, L23E, and L4E subpopulations are largely unaffected in the SM-connectome (*Figure 7E*). Interestingly, Sst+ and 5HT3aR+ subpopulations show a strong hyperpolarization in the late response that is not present in the original connectome. Interestingly, this corresponds with a stronger late response in L5/6 E populations, which could be caused by disinhibition due to the Sst+ and 5HT3aR+ hyperpolarization. This could be explored further in follow-up studies using our connectome manipulator tool (*Pokorny et al., 2025*).

## Mid-range connectivity determines independent functional units and selective propagation of stimulus-responses at larger spatial scales

### Stable spontaneous activity only emerges in nbS1 at predicted in vivo firing rates

After calibrating the model of extrinsic synaptic input for the seven-column subvolume, we tested to what degree the calibration generalizes to the entire nbS1. Notably, this included the addition of mid-range connectivity (*Reimann et al., 2024*). The total number of local and mid-range synapses in the model was 9138 billion and 4075 billion, that is, on average full model simulations increased the number of intrinsic synapses onto a neuron by 45%. Particularly, we ran simulations for $P_{FR} \in [0.1, 0.15, ..., 0.3]$ using the OU parameters calibrated for the seven-column subvolume for $[Ca^{2+}]_o = 1.05$ mM and $R_{OU} = 0.4$. Each of these full nbS1 simulations produced stable non-bursting activity (*Figure 8A*), except for the simulation for $P_{FR} = 0.3$, which produced network-wide bursting activity (*Figure 8—videos 1 and 2*). Activity levels in the simulations of spontaneous activity were heterogeneous (*Figure 8B*, *Figure 8—video 3*). In some areas, firing rates were equal to the target $P_{FR}$, whilst in others they increased above the target (*Figure 8C*). In the more active regions, mean firing rates (averaged over layers) were on the order of 30–35% of the in vivo references for the maximum non-bursting $P_{FR}$ simulation (target $P_{FR}$: 0.25). This range of firing rates again fits with the estimate of firing rate bias from our paper studying spike sorting bias (*Laquitaine et al., 2024*) and the meta-parameter range supporting realistic stimulus-responses in the seven-column subvolume. This also predicts that the nbS1 cannot sustain higher firing rates without entering a bursting regime.

### Independent functional subunits emerge during spontaneous activity with accurate layer and distance-dependence of noise correlations

The subregion with consistently higher firing rates was the upper-limb representation (S1ULp), separated from subregions of lower activity by the dysgranular zones (S1DZ, S1DZO). This demonstrates that the larger spatial scale of the model supports several independently acting functional units. Locations with higher firing rates were accompanied by higher spiking correlations of the local population (*Figure 8B2*, *Figure 8—video 3*), indicating activity that was more driven by intrinsic excitation. Unlike in the simulations of smaller volumes, the correlation emerged mostly from short, transient packets of activity in L6 (*Figure 8A*). Consequently, deviation from the target $P_{FR}$ in those cases was largest in L6 (*Figure 8C*). Taken together, it appears as if the additional synapses from mid-range connectivity moved the model along the spectrum from extrinsically vs. intrinsically driven, but further for some regions than for others. We confirmed a divergence into two populations: one with correlation slightly increased from the simulations of smaller volumes, and one with almost maximal correlations (*Figure 8D*). This distinction was determined mostly in L6 with increases of correlations in other layers remaining small.

The distance dependence of correlations followed a similar profile to that observed in a dataset characterizing spontaneous activity in the somatosensory cortex (*Reyes-Puerta et al., 2015a*; compare red line in *Figure 8I* with *Figure 8—figure supplement 1*). In the in vivo dataset spiking correlation was also low but highest in lower layers, with short 'up-states' in spiking activity constrained to L5 and 6 (see Figure 1E and F in *Reyes-Puerta et al., 2015a*). In the model, they are constrained to L6.

### Stable propagation of evoked activity through mid-range connectivity only emerges in nbS1 at predicted in vivo firing rates

We repeated the previous single whisker deflection evoked activity experiment in the full model, providing a synchronous thalamic input into the forelimb sub-region (S1FL; *Figure 8E*, *Figure 8—videos 4–6*). Responses in S1FL were remarkably similar to the ones in the seven-column subvolume, including the delays and decays of activity (*Figure 8F*). However, in addition to a localized primary response in S1FL within 350 $\mu m$ of the stimulus, we found several secondary responses at distal locations (*Figure 8E*, *Figure 8—video 6*), which was suggestive of selective propagation of the stimulus-evoked signal to downstream areas efferently connected by mid-range connectivity. The response of the main activated downstream area (visible in *Figure 8E*) was confined to L6 (*Figure 8G*). In a follow-up study using the model to explore the propagation of activity between cortical regions (*Bolaños-Puchet and Reimann, 2024*), it is described how the model contains both a feedforward

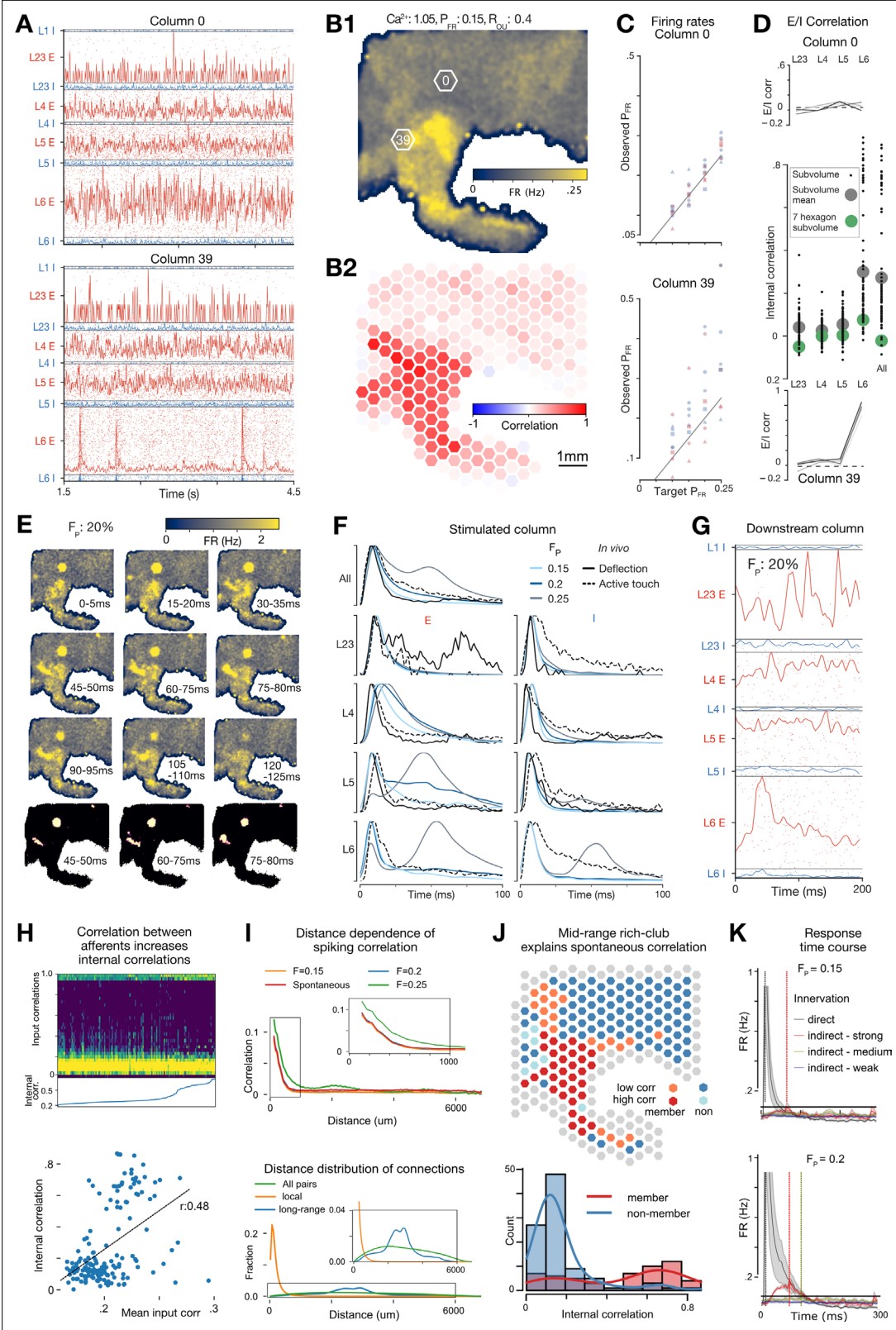

**Figure 8.** Full nbS1 simulations: independent functional units, millisecond-precise stimulus-responses, selective propagation of stimulus-evoked activity through mid-range connectivity, linking structure to function. (**A**) Spiking activity and max normalized histograms for layer-wise E and I populations for the meta-parameter combination $[Ca^{2+}]_o$: 1.05 mM, $R_{OU}$: 0.4, $P_{FR}$: 0.15, and the $0^{th}$ (upper) and $39^{th}$ (lower) hexagonal subvolumes (shown in **B**). (**B1**) Mean FRs across the model. Columns 0 and 39 (subsequently used) are highlighted. (**B2**) Spiking correlations of E/I populations in 240 hexagonal

*Figure 8 continued on next page*

*Figure 8 continued*

subvolumes (diameter: ≈460 $\mu m$). (**C**) Target vs. observed $P_{FR}$ values for the four non-bursting simulations. The number of sides of each marker corresponds to the layer (triangle represents L2/3). Red: E, blue: I. (**D**) Correlations between E and I populations by layer for the four non-bursting simulations for the two hexagons (top and bottom). Color of line from light to dark represents $P_{FR}$. Middle: distributions of spiking correlations of layer-specific populations in columns (520 $\mu m$) of the nbS1 model (black dots; mean values: gray). Compared to the central column when the seven-column subvolume is simulated in isolation (green). (**E**) Trial-averaged activity in flatspace following single whisker deflections. Lower: subset of time windows shown with only the top 2% of bins for each time window after baseline activity level subtracted. (**F**) Trial-averaged PSTHs (baseline and max normalized) for column 0. For $F_P = 15\%$ and $F_P = 25\%$, respectively, the evoked responses passed and failed the latency and decay-based criteria tests for similarity to in vivo. Response for $F_P = 20\%$ passed the tests for all populations except L5 E, which showed a more sustained response. (**G**) Trial-averaged response of column 39. (**H**) Correlations of mid-range inputs. Upper: distribution of spiking correlations of inputs into 240 subvolumes. Calculated at reduced spatial resolution, based on connection counts and correlations between the subvolumes (see 'Methods'). Subvolumes are sorted by the E/I correlation of neurons within them. Lower: mean of the correlations of inputs into 240 subvolumes vs. internal spiking correlation for all subvolumes during spontaneous activity. Black line: linear fit. calculated based on connection counts and correlations between 50 $\mu m$ hexagonal subvolumes (see 'Methods'). Data for evoked activity is shown in *Figure 8—figure supplement 1*. (**I**) Upper: spiking correlations of subvolumes against their distances in spontaneous (red) and evoked (orange: $F_p = 0.15\%$, blue: $F_p = 0.2\%$, green: $F_p = 0.25$) activity. For increased spatial resolution, smaller (58 $\mu m$) subvolumes were used, hence correlation values are not comparable to values in (**B**). Lower: distribution of soma distances of neuron pairs connected by local (orange) or mid-range (blue) connectivity. Green: distances of all pairs, independent of connectivity. (**J**) Upper: classification of subvolumes based on low or high internal correlation and membership in mid-range rich club Data for evoked activity shown in *Figure 8—figure supplement 1*. Lower: E/I correlation of subvolumes for members of the rich club (red) and non-members (blue). (**K**) FRs of subvolumes following a stimulus with $F_P = 0.15\%$ (top) and $F_P = 0.2\%$ (bottom). Mean (lines) and SEM (shaded area) over 10 repetitions shown for subvolumes directly innervated by the VPM stimulus (black), strongly innervated by directly innervated subvolumes (with over $10^6$ synapses, red), or by medium strength (over $10^5$ synapses, yellow) or weak (over $10^4$ synapses) indirect innervation. Dashed lines indicate locations of peaks.

The online version of this article includes the following video and figure supplement(s) for figure 8:

**Figure supplement 1.** Full nbS1—supplementary.

**Figure 8—video 1.** Layer-wise E and I population rasters and max-normalized histograms of spontaneous activity for hex0 for meta-parameter combinations: $Ca^{2+} = 1.05$ mM, $R_{OU} = 0.4$, and $P_{FR} = 0.05 - 0.3$.
https://elifesciences.org/articles/99693/figures#fig8video1

**Figure 8—video 2.** Layer-wise E and I population rasters and max-normalized histograms of spontaneous activity for hex39 for meta-parameter combinations: $Ca^{2+} = 1.05$ mM, $R_{OU} = 0.4$, and $P_{FR} = 0.05 - 0.3$.
https://elifesciences.org/articles/99693/figures#fig8video2

**Figure 8—video 3.** Visualization of spontaneous activity for the full nbS1 model for the parameter combination $Ca^{2+} = 1.05$ mM, $P_{FR} = 0.15$, $R_{OU} = 0.4$ after collapsing activity to flatspace (activity binned and smoothed).
https://elifesciences.org/articles/99693/figures#fig8video3

**Figure 8—video 4.** Layer-wise E and I population rasters and max-normalized histograms of stimulus evoked activity for hex0 for each meta-parameter combination $Ca^{2+} = 1.05$ mM, $R_{OU} = 0.4$, $P_{FR} = 0.15$, and $F_P = 0.1 - 0.25$.
https://elifesciences.org/articles/99693/figures#fig8video4

**Figure 8—video 5.** Layer-wise E and I population rasters and max-normalized histograms of stimulus evoked activity for hex39 for each meta-parameter combination $Ca^{2+} = 1.05$ mM, $R_{OU} = 0.4$, $P_{FR} = 0.15$, and $F_P = 0.1 - 0.25$.
https://elifesciences.org/articles/99693/figures#fig8video5

**Figure 8—video 6.** Visualization of trial-averaged response of the full nbS1 model over the 10 single whisker deflection protocol for the parameter combination $Ca^{2+} = 1.1$ mM, $P_{FR} = 0.15, R_{OU} = 0.4$, and $F_P = 20\%$ after collapsing activity to flatspace (activity binned and smoothed).
https://elifesciences.org/articles/99693/figures#fig8video6

projection pattern, which projects to principally to synapses in L1 and L23, and a feedback-type pattern, which principally projects to synapses in L1 and L6. On visualizing the innervation profile from the stimulated hexagon to the downstream hexagon, we can see that we have stimulated a feedback pathway (*Figure 8—figure supplement 1*).

## Membership in mid-range connectivity rich club predicts correlation of spontaneous activity

As the model is homogeneous across the different subregions in terms of neuronal morphologies and physiology, we expected the differences in spontaneous and evoked activity across the model to be explained by the topology of synaptic connectivity (*Reimann et al., 2024*). While the extrinsic input from the OU injections was uncorrelated by design, synaptic input from local connectivity would feature at least some degree of correlation. This led to asking, how correlated was the input from

mid-range connections? We estimated for each pair of mid-range connections innervating a 460 $\mu m$ hexagonal subvolume its spiking correlation and compared the resulting distribution to the correlation within the subvolume itself (see 'Methods'; *Figure 8H*, upper). We found low correlations for most pairs, but with a significant number of high-correlation outliers for subvolumes with high internal correlations. Moreover, when we considered the mean of the measure, we found that it explains most of the variability in internal correlations, for both spontaneous (*Figure 8H*, lower) and evoked states (*Figure 8—figure supplement 1*).

This led to the following tentative explanation: Local connectivity provides inputs from within 500 $\mu m$, which is virtually guaranteed to be correlated to some degree (*Figure 8I* upper vs. lower, orange). This is a result of connections being *locally dense*, as they are concentrated within a small neighborhood (*Figure 8* lower, green vs. orange). mid-range connectivity originates from further away, sampling from different functional units that may or may not be correlated, and being more diluted (*Figure 8I* lower, green vs. blue). Consequently, mid-range inputs are likely to act not much different from the uncorrelated OU process, unless this is overcome by a non-random topology of mid-range connectivity.

Specifically, we considered the mid-range connectivity at meso-scale in terms of the graph of synapse counts between 460 $\mu m$ subvolumes. When the graph was thresholded at 8 million synapses, most subvolumes lacked connections at that strength completely, but others formed a single connected component, indicating the formation of a *rich club* (*Colizza et al., 2006*; *Figure 8—figure supplement 1J*, upper, red and orange). We found that subvolumes with high internal correlation were part of the rich club, or directly adjacent to one of its members (*Figure 8J*). In the presence of strong thalamic inputs, this could be overcome, as the increased activity level spread along sufficiently strong mid-range connections (*Figure 8K*), albeit with substantial delays between 60 and 100 ms. This could induce high correlations in strongly connected parts of the model, outside the rich club as well (*Figure 8—figure supplement 1*), and shows that the model supports selective propagation of activity to areas efferently connected with mid-range connectivity.

## Discussion

We presented novel methods to build, simulate, and validate models of cortical tissue that correspond directly with 3D digitized brain atlases. As demonstrated, this enables laboratory experiments to have a simulatable in silico counterpart, and vice versa: predictions made by the model automatically have a precise correspondence in biology. By recreating and extending five laboratory experiments under a single model parameterization, we provided strong model validation and demonstrated the model's natural versatility. To our knowledge the simulations of the full nbS1 represent the first simulations of stable in vivo-like spontaneous and stimulus-evoked activity in a large-scale biophysically detailed model of multiple cortical subregions connected through local and mid-range connectivity. The model generated an initial set of predictions about the relationship between cortical structure and function, and provides a basis for future studies and predictions. The table in *Supplementary file 1* describes each quantitative prediction made in this article, and how it might be tested in vivo. Additionally, all assumptions are listed in the table in *Supplementary file 8*. We made the model and simulation tools available for use, enabling community-driven testing, exploration and refinement.

Excitation from non-modeled brain regions is modeled by injecting somatic conductances and is calibrated for specific populations. For the purpose of calibration, we presented a novel technique, which allows rapid and methodical characterization and validation of emergent model activity. The efficiency of the technique allows rapid recalibration after changes to the model such as adding new brain regions or changing anatomical details. Connected divided by unconnected firing rates measured during the process are also a prediction of the degree to which the activity of a population is determined by incoming connections from local vs. extrinsic sources. We found an increase of the importance of local connectivity from superficial to deeper layers, in line with the canonical view of the cortex (*Lübke and Feldmeyer, 2007*; *Feldmeyer, 2012*), which places layers 5 and 6 at the end of local information flow.

Although abstract relative to the model, the injection amplitudes could be predicted from the number of missing synapses, suggesting it provides a functional representation of their anatomical counterparts. The modular model of extrinsic connectivity can be refined or replaced in the future or when larger fractions of the brain are modeled. The calibration framework could optimize per

population parameters for other compensation methods, whilst still offering an interpretable spectrum of firing rate regimes at different levels of $P_{FR}$. For example, more realistic compensation schemes could be explored which introduce (a) correlations between the inputs received by different neurons and (b) compensation distributed across dendrites, as well as at the soma. We predict that such changes would make spontaneous activity more correlated at the lower spontaneous firing rates which supported in vivo-like responses ($P_{FR} : 0.1 - 0.5$), which would in turn make stimulus-responses more noise correlated.

Our approach to obtaining in vivo-like activity contrasts with a recent approach in a large-scale biophysical model of primary visual cortex of similar scale to our smaller subvolume (*Billeh et al., 2020*). There, extrinsic inputs were delivered through dendritic synapses instead of somatic injection, which is more anatomically realistic, and intrinsic recurrent weights were adjusted to match extracellularly recorded firing rates. They find a single activity state, instead of the in vivo-compatible spectrum here that allowed us to contrast anesthetized and awake states. The model in general differs from hybrid models, which jointly use biologically detailed models and point neuron models (*Billeh et al., 2020*; *Dura-Bernal et al., 2023*), and also from the work of *Egger et al., 2020*, who modeled activation of L5 pyramidal cells by constraining patterns of synaptic input based on receptive fields and predicted anatomical innervation.

Given the complexity of the model, how can we be confident that its activity is in vivo-like? First, we demonstrated correlated spontaneous dynamics in the form of global and local fluctuations and dynamic E-I balance, as found experimentally (*Renart et al., 2010*). Second, population firing rate distributions were long-tailed with sub 1 Hz peaks, and were similar for spontaneous and stimulus-evoked activity, as reported (*Wohrer et al., 2013*). Moreover, mean firing rates below 1 Hz are required by metabolic constraints (*Attwell and Laughlin, 2001*; *Lennie, 2003*). In response to simple whisker stimuli, response sparsity, spike counts, and the temporal profile and amplitudes of layer-wise populations were similar to in vivo under anesthetized and awake conditions, for different regions of the meta-parameter space. The effect of the spontaneous meta-parameters on evoked responses, also shows that the networks spontaneous state affects stimulus-responses, as observed in vivo (*Isbister et al., 2021*). Although from the barrel system, these stimuli offer some of the simplest stimulus-response paradigms, enabling principles of neural dynamics and information processing to be studied; in particular, the correspondence between atomic units of sensory information and neural representations. The model also predicts the number of thalamic fibers stimulated for whisker-flick stimuli. Such validations are crucial for complex models, as they provide context for more complex validations. Moreover, in a rich and complex nonlinear system, the source of discrepancies with in vivo activity are simpler to ascertain under simpler protocols.

For more complex validations, we reproduced protocols from three laboratory experiments. These experiments used a wide array of techniques, which could be accurately recreated due to the model's close correspondence with biological tissue. Additionally, we could go beyond the original experiments. For example, we predicted an increased number of neurons would be required to have a decodable effect on postsynaptic activity under in vivo conditions over the in vitro conditions of the original experiment. This can be explained by lower synaptic reliability under in vivo conditions. Also, we predicted a paradoxical effect of optogenetic stimulation on L6 PV+ interneurons, namely a decrease in firing with increased stimulus strength. This is reminiscent of the paradoxical responses found by *Mahrach et al., 2020* in the mouse anterior lateral motor cortex (in L5, but not in L2/3) and barrel cortex (no layer distinction), respectively. While *Mahrach et al., 2020* conducted their recordings in awake mice not engaged in any behavior, we observed this effect only when drifting grating patterns with high contrast were presented. Nevertheless, consistent with their findings, we found the effect only in deep but not in superficial layers, and only for PV+ interneurons but not for PCs. Our model could therefore be used to improve the understanding of this paradoxical effect in follow-up studies. These examples demonstrate that the approach of modeling entire brain regions can be used to further probe the topics of the original articles and cortical processing. This could be continued in the future using paradigms that take advantage of the multiple regions offered by the model to study processing in cortical hierarchies.

Responses to whisker flick stimuli were closer to biology for values of the $P_{FR}$ meta-parameter between 0.1 and 0.5, according with the known inflation of mean firing rates stemming from the bias of extracellular spike sorting techniques towards larger and more active neurons (*Olshausen and*

*Field, 2006*). Supporting our estimates, patch-clamp experiments show ubiquitous neuronal silence (*Crochet and Petersen, 2006*), and spontaneous and stimulus-evoked barrel cortex firing rates as much as 10 times lower than for extracellular recordings (0.05–0.15 Hz vs. 0.8–1.5 Hz; *Olshausen and Field, 2006*). This is also in line with our recent work using the model, which estimated a spike sorting bias corresponding to $P_{FR} = 0.3$ using virtual extracellular electrodes (*Laquitaine et al., 2024*). Each recording technique comes with challenges and biases (*Shoham et al., 2006*; *Olshausen and Field, 2006*; *Barth and Poulet, 2012*; *Wohrer et al., 2013*). Calcium imaging and patch-clamp experiments are less biased towards frequently spiking neurons (*Wohrer et al., 2013*), yet the former can only infer spiking activity and may be biased towards neurons that favor marker expression. Patch-clamp techniques can reduce firing rates, be biased towards larger or more active neurons (*Olshausen and Field, 2006*) and are limited to recording a few neurons simultaneously, preventing characterizations of single trial population dynamics. An earlier version of the model has been used to simulate cortical extracellular potentials (*Reimann et al., 2013*). We can now simulate modern extracellular electrodes and compare multi-unit activity and spike sorting results to the in vivo-like ground truth to gain further insights into the potential over-estimation of mean firing rates.

During simulations of the entire nbS1, we found emerging spatial inhomogeneities in network activity. Specifically, we observed sharp transitions in firing rates and pairwise correlations at disgranular zones. This highlights that the model supports several, independently acting functional units. Analysis of spatial correlations suggests a radius of approximately 400 $\mu m$ for a single unit. This property was not present in the previous intra-regional scale model (*Markram et al., 2015*), but is rather an emergent property of the interregional scale, demonstrating that we achieved our goal of building a model that can be used to study mid-range interactions between regions. Analyzing the source of the inhomogeneities suggested a prominent role of a non-random rich-club structure of mid-range connections at the meso-scale. Within functional units, we found that inhomogeneous increases of spiking correlations were confined to spontaneous packets of L6 activity.

In keeping with the philosophy of compartmentalization of parameters and continuous model refinement (see Introduction), it was essential to improve validity at the columnar scale (relative to *Markram et al., 2015*) as part of demonstrating validity of the full nbS1. Indeed, improved parameterization and validation at smaller scales was essential to parameterizing background input which generated robust nbS1 activity within realistic $[Ca^{2+}]_o$ and firing rate ranges. We view this as a major achievement, as it was unknown whether the model would achieve a stable and meaningful regime at the start of our investigation. Whilst we would have liked to go further, our primary goal was to publish a well characterized model as an open resource that others could use to undertake further in-depth studies. In this regard, we are pleased that the parameterization of the nbS1 model has already been used to study EEG signals (*Tharayil et al., 2025*), as well as propagation of activity between two subregions (*Bolaños-Puchet and Reimann, 2024*). Investigation, improvement and validation must be continued at all spatial scales in follow-up papers with detailed description, figures and analysis, which cannot be covered in this article. Each new study increases the scope and validity of future investigations. In this way, this model and paper act as a stepping stone towards more complex questions of interest to the community such as perception, task performance, predictive coding and dendritic processing. This was similar for *Markram et al., 2015* where the initial paper was followed by more detailed studies. Unlike the *Markram et al., 2015* model, the new model can also be exploited by the community and has already been used in a number of follow-up papers studying (*Ecker et al., 2024a*; *Bolaños-Puchet and Reimann, 2024*; *Pokorny et al., 2025*; *Egas Santander et al., 2025*; *Tharayil et al., 2025*; *Laquitaine et al., 2024*). We believe that the number of use cases for such a general model is vast and is made larger by the increased size of the model.

It is important to highlight where the model deviates from biological data. First, the model omits several known anatomical elements, such as glia and the presence of gap junctions between certain neuron-types, but the model's spatial context provides a natural scaffold for them. Our modeling techniques assume generality of rules and parameters, unless indicated by the data. Therefore, violations indicate that biological rules and data may be more specific. For example, model thalamic inputs innervate any dendrite placed in the targeted layers equally, but this did not reproduce the timing of inhibitory subpopulation responses. Stronger innervation of PV+ neurons than other inhibitory neurons may rectify this. However, it is unclear whether innervation should be anatomically or physiologically stronger, as the best available data mixes structural and physiological strength (*Sermet et al., 2019*).

In our previous work (*Roussel et al., 2023*), we linked mouse inhibitory me-models to transcriptomic types (t-types) in a whole mouse cortex transcriptomic dataset (*Gouwens et al., 2019*). This can provide a direct correspondence in future large-scale mouse models. As we model only a single electrical type for pyramidal cells, there is no one-to-one correspondence between our me-models and the 10 different pyramidal cell types identified there. We are not currently aware of any method which can recreate the electrical features of different types of pyramidal cells using only generic ion channel models. To achieve the firing pattern behavior of more specific electrical types, usually ion channel kinetics are tweaked, and this would violate the compartmentalization of parameters. In future we hope to build morpho-electric-transcriptomic type (met-type) models by selecting gene-specific ion channel models (*Ranjan et al., 2019*; *Ranjan et al., 2024*) based on the met-type's gene expression. Data specific to different neuron sections (i.e., soma, AIS, apical/basal dendrites) of different met-types, such as gene expression, distribution of ion channels, and voltage recordings under standard single cell protocols, would be particularly useful.

Finally, when required, we generalized data from different animals and brain regions to build our rat nbS1 model. This is the accepted state-of-the-art in computational neuroscience, for example, all 19 data-driven models of rodent microcircuitry listed in Figure 2 of the recent review of *Ramaswamy, 2024* conduct some sort of mixing, including the advanced mouse V1 model of the Allen Institute (*Billeh et al., 2020*). Whilst a first truly single species model would be a great advance, it is not required to be the basis of in silico research. Our model (and previous ones) can be used to study the many cortical mechanisms that are common to the closely related organisms rat and mouse. This paper and its companion paper serve to present a methodology for modeling micro- and mesoscale anatomy and physiology, which can be applied for other cortical regions and species. With the rapid increase in openly available data, efforts are already in progress to build models of mouse brain regions with reduced reliance on data mixing thanks to much larger quantities of available atlas-based data. This also includes data for the validation of emergent network-level activity. Here we chose to compare network-level activity to data mostly from the barrel cortex, as well as a single study from primary visual cortex. Whilst a lot of the data used to build the model was from the barrel cortex, the barrel cortex also represents a very well characterized model of cortical processing for simple and controlled sensory stimuli. The initial comparison of population-wise responses in response to accurate thalamic input for single whisker deflections was essential to demonstrating that the model was closer to in vivo, and we were unaware of similar data for non-barrel somatosensory regions. Moreover, our optogenetic and lesion study demonstrated the capacity to compare and extend studies of canonical cortical processing in the whisker system.

However, we cannot always predict the effect caused by mixing data sources. An interesting example of this is described in the companion paper (*Reimann et al., 2024*), stating that certain rules of inhibitory connectivity found in a mouse electron-microscopic (EM) connectome (*Schneider-Mizell et al., 2025*) cannot be explained by an non-selective pruning axo-dendritic appositions. We demonstrated the utility of the model to explore the effect of any connectivity rule, using the example of a mouse EM-compatible 'SM-connectome'. After rewiring and refitting synaptic, and conductance injection parameters, we found that the purely perisomatic targeting by PV+ neurons and the increased inhibitory targeting by VIP+ neurons shifted how much individual layers were driven by intrinsic and extrinsic excitation. This could not be explored with current techniques in vivo and highlights a strength of in silico experimentation. On the other hand, this data source contradicts some of our rat nbS1 parameters, for example, the number of synapses per connection from light microscopy (*Markram et al., 2015*) do not match with the ones obtained from mouse V1 EM. Thus we used the rat data sources for building our baseline model, but make both models available in a simulatable format (see 'Data availability').

# Methods

**Key resources table**

| Reagent type (species) or resource | Designation | Source or reference | Identifiers | Additional information |
|---|---|---|---|---|
| Software | Electrical model optimization | https://github.com/BlueBrain/singlecell-emodel-suite; *Reva et al., 2023*; *Van Geit et al., 2024* | | |
| Software | Simulation | This paper | | 10.5281/zenodo.8075202 |
| Software | Model loading and interaction | This paper | | 10.5281/zenodo.8026852 |
| Software | Model and simulation analysis | This paper | | 10.5281/zenodo.8016989 |
| Software | Specific simulation configurations | This paper | | 10.5281/zenodo.7930276 |
| Software | Connectome Manipulation | https://github.com/BlueBrain/connectome-manipulator; *Pokorny et al., 2025*; *Wolf et al., 2024* | | |

## Resource availability

### Materials availability

No materials were used in this computational work.

## Method details

### Optimization of ion channel conductance densities

Neuron models from *Reva et al., 2023* were made as follows. Ion channel conductance densities were optimized for a pool of reconstructed neurons, for which in vitro recordings had previously been made. Ion channel densities were then generalized to other neuron models of the same e-type. Resulting neuron models were only used in the construction of the model if the value of the cost function was within 5 standard deviations of the experimental mean.

### AIS adjustment

For some electrical models, further adjustments were made to AIS sections to handle the large input currents present under conditions of high network activity. In short, for certain pyramidal cell morphologies, the AIS was replaced by a cylinder with a length of 60 $\mu m$. Its diameter was manually calibrated such that the ratio of the AIS input resistance over the input resistance of somato-dendritic compartments was equal to 2. Afterwards, it was confirmed that the cost function of their electrical models remained below the threshold.

### Parameterization of synaptic physiology

The Tsodyks–Markram synaptic model (*Tsodyks and Markram, 1997*; *Markram et al., 1998*; *Fuhrmann et al., 2002*; *Loebel et al., 2009*), upgraded to feature multi-vesicular release (*Barros-Zulaica et al., 2019*; *Ecker et al., 2020*) comprises the following parameters: peak conductance $\hat{g}$, depression and facilitation time constants $D$, $F$, the decay time constant of the PSC $\tau_{decay}$, the release probability $U_{SE}$, the average number of vesicles in the release-ready pool $N_{RRP}$, the Hill coefficient of the nonlinear $[Ca^{2+}]_o$ dependent scaling of release probability $U_{Hill}$, and finally $\hat{g}_{ratio}$: the NMDA/AMPA ratio for excitatory synapses, and the $GABA_B/GABA_A$ ratio for inhibitory synapses. Where possible, pathway and synapse-type specific parameter values were taken from the literature, and after eventual corrections (for differences in the temperature and solutions used in the experiments), were put directly into the model using the technique described for the use case of hippocampal CA1 in *Ecker et al., 2020*. For example, parameters $U_{SE}$, $D$, $F$, and $U_{Hill}$ have previously been fitted for a number of pathways using data acquired from paired-recording experiments. Furthermore, $\hat{g}_{ratio}$ parameter has been previously determined from outside-out patch recording experiments.

The remaining parameters ($\hat{g}$ and $N_{RRP}$) were iteratively calibrated using an in silico setup reproducing the paired-recordings experiments detailed below. For this parameterization, all

paired-recording data sources from *Markram et al., 2015* were re-used. To further constrain the variance of EPSP amplitudes in L5 TTPCs, fitted parameters from *Barros-Zulaica et al., 2019* were used. The dataset was also enriched with recent recordings from L6 (*Qi and Feldmeyer, 2016*; *Yang et al., 2020*; *Yang et al., 2022*). Instead of generalizing L4 E peak synaptic conductances for all thalamic fibers (as in *Markram et al., 2015*), synaptic conductances were constrained using EPSP amplitude measurements from thalamocortical slices (*Beierlein and Connors, 2002*; *Beierlein et al., 2003*). Fibers originating from POm were not present in *Markram et al., 2015*. Due to the lack of in vitro measurements, physiological parameters from VPM synapses were generalized to describe POm synapses.

Not all synaptic peak conductances can be measured with paired recordings due to the space-clamp artifact (*Markram et al., 2015*; *Ecker et al., 2020*), and estimating the number of release-ready vesicles is also challenging experimentally (*Loebel et al., 2009*; *Barros-Zulaica et al., 2019*). Thus, for synaptic pathways for which there was in vitro reference data, these two parameters were determined in a simulation-driven iterative process. Firstly, 50 pairs of in silico neurons were randomly sampled. Second, the postsynaptic cell was current-clamped, and the PSP at the soma of the postsynaptic cell in response to a presynaptic action potential was measured. This was repeated 35 times for each pair as in *Ecker et al., 2020*. Thirdly, the mean and the CV of these amplitudes were extracted and compared with reference in vitro data. Finally, peak conductance ($\hat{g}$), and the average number of vesicles in the release-ready pool ($N_{RRP}$) were adjusted to match the amplitude and CV of the in vitro reference (*Ecker et al., 2020*). In some cases, this resulted in reaching the lower bound of univesicular release (*Barros-Zulaica et al., 2019*).

Where experimental data was not available, mean synaptic physiology parameters from similar pathways were used (*Markram et al., 2015*; *Ecker et al., 2020*). For the VPM to L5 E pathway, the maximum of VPM to L4 E and VPM to L6 E conductances (instead of the mean) were used, based on resulting evoked firing rates found in a later modeling step (*Figure 5—figure supplement 1C*). To calibrate the mPSC frequencies of different pathways, single-cell simulations with different values of the spontaneous release frequency for all synapses of a set of 1000 cells in a given pathway were run. In these simulations, in silico voltage-clamp recordings were performed to measure the resulting mPSC frequency at the soma. This data was then fitted with a logarithmic function and the value of the spontaneous release frequency matching the in vitro reference value for the mPSC frequency was interpolated. As in vitro paired recording data is sparse, all available sources to determine synaptic parameters were reused for validation.

## Datasets used for validation of in vivo-like dynamics

Activity in the model is compared with spike-sorted extracellular recordings from the barrel cortex of anesthetized rats (*Reyes-Puerta et al., 2015b*). The dataset contains single unit spike times for both spontaneous activity and responses to single whisker deflections, across six layer-wise neuron populations (L2/3 E, L2/3 I, L4 E, L4 I, L5 E, and L5 I). We extend the dataset using the average of L2/3 I, L4 I, and L5 I as a reference for L1 I and L6 I, and an alternative reference for L6 E (*De Kock et al., 2007*). The spontaneous firing rates of the nine neuron populations are collected in the vector of reference firing rates, $V_{FR}$.

As a secondary comparison, we use a dataset of in vivo juxtacellular and whole-cell recordings of barrel cortex responses in awake mice during an active whisker touch paradigm (*Yu et al., 2019*). While the animal model is different, using this data enables comparison to awake responses of layer-wise E populations and I subpopulations (VIP+ [vasoactive intestinal peptide, a part of our 5HT3aR+ subpopulation], Sst+, FS [fast spiking, corresponding to our PV+ subpopulation]). Moreover, this allows validating the model against properties of cortical responses across similar but different paradigms, reducing the risk of over fitting.

## Somatic conductance injection

Random background synaptic inputs, representing uncorrelated inputs from other brain regions not present in the model, are modeled as a conductance-based somatic injection (*Figure 4B1*). Specifically, we inject conductance signals described by an OU process (similar to *Destexhe et al., 2001*), given by the following equation:

$$\frac{dg(t)}{dt} = -\frac{1}{\tau}[g(t) - g_0] + \sqrt{D}\chi(t) \tag{1}$$

where $g(t)$ is the conductance signal, $g_0$ its mean value, $\tau$ is the decay time constant, $D$ is the diffusion coefficient, and $\chi(t)$ is a Gaussian white noise process. The diffusion coefficient can also be expressed in terms of the standard deviation $\sigma$ of the signal: $D = \frac{2\sigma^2}{\tau}$. Note that we only consider an excitatory conductance signal representing mid-range excitatory drive, whereas local inhibitory input is deemed sufficient within the circuit model. As we are modeling a nonspecific random background noise, we use a different, uncorrelated OU process for each cell.

Given the diversity of morphologies and electrical behaviors of cells in our circuit, we scale the mean and standard deviation of the injected signal based on individual cell properties. Specifically, we compute the input resistance $R_{in}$ of each cell at its resting potential, then take its reciprocal value $G_{in} = 1/R_{in}$ (units of conductance). The mean $g_0$ and standard deviation $\sigma$ of the injected conductance signal are then expressed as percentages $OU_\mu$ and $OU_\sigma$ of this input conductance $G_{in}$:

$$g_0 = \frac{OU_\mu}{100} * G_{in} \tag{2}$$

$$\sigma = \frac{OU_\sigma}{100} * G_{in} \tag{3}$$

The $OU_\mu$ and $OU_\sigma$ percentage values are used in the calibration technique described in the following sections. To reiterate, $OU_\mu$ and $OU_\sigma$ scale the mean of the injected conductance and the size its fluctuations respectively.

## Details of spontaneous activity calibration

$OU_\mu$ and $OU_\sigma$ are parameterized separately for each of nine populations (layer-wise E and I combinations; *Figure 4B2*), leading to 18 parameters, grouped into two 9-dimensional vectors $\boldsymbol{OU_\mu}$ and $\boldsymbol{OU_\sigma}$. Spontaneous firing rates across populations, denoted by the vector $\boldsymbol{C_{FR}}$, are then determined by the vectors $\boldsymbol{OU_\mu}$ and $\boldsymbol{OU_\sigma}$, and $[Ca^{2+}]_o$. The objective of the calibration procedure is to determine a mapping that outputs the $\boldsymbol{OU_\mu}$ and $\boldsymbol{OU_\sigma}$ required to achieve a particular set of firing rates $\boldsymbol{C_{FR}}$, for a given $[Ca^{2+}]_o$.

This requires two simplifications: First, we reduce the space of possible $\boldsymbol{OU_\mu}$ and $\boldsymbol{OU_\sigma}$ values by fixing $\boldsymbol{OU_\sigma} = \boldsymbol{OU_\mu} * R_{OU}$, where $R_{OU}$ is a prespecified constant that controls the level of noise assumed to be present in the external synaptic input. Second, we learn the mapping only for dynamical states where the firing rates of the nine populations are a proportion of reference in vivo firing rates, that is, $\boldsymbol{C_{FR}} = P_{FR} * \boldsymbol{V_{FR}}$, with $P_{FR} \in [0, 1]$ (*Figure 4B2*). This is based on the assumption that extracellular spike detection sampling biases (*Olshausen and Field, 2006*; *Wohrer et al., 2013*) affect all populations equally.

The mapping can be written as $\Psi_{Ca^{2+}, R_{OU}} : \boldsymbol{C_{FR}} \mapsto \boldsymbol{OU_\mu}$ and is simplified into two separate mappings which are determined separately and then combined (*Figure 4—figure supplement 1*). The first mapping finds values of the 9 output parameters $\boldsymbol{OU_\mu}$, for a given $R_{OU}$, which produce particular *unconnected firing rates* $\boldsymbol{U_{FR}}$ (i.e., where the neurons in the network are not connected to each other). The second then maps between $\boldsymbol{U_{FR}}$ and $\boldsymbol{C_{FR}}$.

The first mapping $\chi : (R_{OU}, \boldsymbol{U_{FR}}) \mapsto \boldsymbol{OU_\mu}$, is attained by running 1-second simulations over various combinations of $OU_\mu$ and $OU_\sigma$ (*Figure 4—figure supplement 1*). $R_{OU}$ constrains $OU_\mu$ and $OU_\sigma$ to a line of possible combinations in the 2D space. We generate an approximation of $\chi$ based on linear interpolation between the results of simulations along the line. Note, in particular, that $\chi$ is independent of $[Ca^{2+}]_o$.

Given the mapping $\chi$, the remaining problem is to determine the values of $\boldsymbol{U_{FR}}$ that achieve certain target firing rates given by $P_{FR} * \boldsymbol{V_{FR}}$, for $P_{FR} \in [0, P_{max}]$ and each combination of $[Ca^{2+}]_o$ and $R_{OU}$, that is, to determine the mapping $\phi_{Ca^{2+}, R_{OU}} : \boldsymbol{U_{FR}} \mapsto \boldsymbol{C_{FR}}$. This is done by measuring values of $\boldsymbol{C_{FR}}$ produced for given values of $\boldsymbol{U_{FR}}$ in a number of simulations. However, instead of exploring the whole parameter space of possible $\boldsymbol{U_{FR}}$ values, we only explore a small number of values of $\boldsymbol{U_{FR}}$ which we expect to produce the target $\boldsymbol{C_{FR}}$ values. This is done by iteratively learning the mapping $\phi_{Ca^{2+}, R_{OU}} : \boldsymbol{U_{FR}} \mapsto \boldsymbol{C_{FR}}$ and in each iteration using $\phi$ to predict $\boldsymbol{U_{FR}}$ values that bring the dynamics closer to the target firing rate, that is, by sampling values given by $\boldsymbol{U_{FR}} = \phi^{-1}(\boldsymbol{C_{FR}}) = \phi^{-1}(P_{FR} * \boldsymbol{V_{FR}})$ for 10 equidistant values

of $P_{FR} \in [0, P_{max}]$. The initial guess of $\phi$ is simply the identity, that is, assuming no effect of connecting the network, with $P_{max} = 0.5$. All other iterations use $P_{max} = 1.0$.

We update $\phi$ by fitting an exponential: $\boldsymbol{C_{FR}} = \alpha * e^{\beta \boldsymbol{U_{FR}}} + \kappa$ to the values of $\boldsymbol{U_{FR}}$, $\boldsymbol{C_{FR}}$ measured on the latest iteration (*Figure 4—figure supplement 1*). In a recurrently connected network such as our model, the connected firing rate of a population depends on the firing rates of all other populations in complex and nonlinear ways. This is addressed by our simplification to restrict the compatible dynamic states, where by definition the target firing rate of one population determines the target firing rates of all others.

The firing rates of the L2/3 I and L4 I and L6 I populations could not be sufficiently lowered for $P_{FR} < 0.2$ and $P_{FR} < 0.1$, respectively, as they spontaneously fired even at their resting potentials (*Figure 4—figure supplement 1D*).

## Generalization to other network states

We first approximated $\phi$ for $[Ca^{2+}]_o = 1.1$ mM and $R_{OU} = 0.4$. To reduce the number of iterations steps needed for other combinations of $[Ca^{2+}]_o$ and $R_{OU}$, we initialized the optimization process with the mapping $\phi$ obtained for this parameter pair instead of basing it on the identity (*Figure 4—figure supplement 1D*, lower; *Figure 4—figure supplement 1*). Only for $P_{FR}$ above 0.8 and $[Ca^{2+}]_o = 1.1$ mM did we have to slightly relax our acceptance criteria (*Figure 7D*, lower). Additionally, $[Ca^{2+}]_o = 1.1$ mM, $P_{FR} = 1.0$, and $R_{OU} = 0.2$ led to bursting activity uncharacteristic of in vivo activity and was excluded from further study.

## Evoked activity comparison and validation

The spike times of single whisker deflection stimuli were drawn from a PSTH of VPM neurons in response to 3 ms mechanical single whisker deflections made at 1 Hz in urethane anesthetized rats (*Diamond et al., 1992*; *Figure 5A2*). Since the in vivo data for cortical responses to single whisker deflections was from responses to 2 ms mechanical whisker deflections, we compressed the tail of the VPM PSTH to allow a direct comparison with in silico responses (*Figure 5A2*). A VPM PSTH for the active whisker touch stimulus was used from the same dataset as the corresponding cortical responses (*Yu et al., 2019*).

Firstly, in silico responses were compared to data from the C2 barrel of the anaesthetized rat in response to single whisker deflections made at <1 Hz. For the active whisker touch stimulus, in vivo activity was used from aggregated over principal barrels corresponding to the untrimmed whisker.

As an initial test of similarity with the corresponding in vivo dataset, trial-averaged PSTHs were first calculated for each parameter combination, both for all neurons, and for each of the layer-wise E and I populations. We then tested whether the latencies, 50% and 25%-decay points of the trial-averaged PSTHs were respectively no more than 10 ms, 10 ms, and 40 ms later than those of the corresponding in vivo populations, and that there was no secondary rise in the PSTHs after the initial decay (by testing that none of the PSTHs for individual populations were more than 35% above baseline after 75 ms). L1 I was excluded from these tests, as it showed little or no response to the stimulus and we had no in vivo reference. These tests allowed us to split the 4D parameter space into two regions, one where in silico responses are consistent with in vivo responses, and another where they are not (see 'Results').

## In silico experimental methods

Simulation of a morphologically detailed model first requires specifying the conditions (parameters and inputs) under which to perform the simulation. Given the correspondence between variables in the model and properties of biological systems, we are able to mimic to a certain extent the conditions and protocols of experimental studies. These are combinations of existing techniques or even techniques that cannot or have not yet been performed experimentally. On top of the compensation for missing external input and different $[Ca^{2+}]_o$, we implemented further mechanisms to simulate specific experimental conditions. On a technical level, these comprise somatic current or conductance injections, adjustment of synaptic connectivity parameters, and selective activation of thalamic fibers. These are static conditions, however, in that their time course must be determined before the simulation is run. Somatic injections and connectivity adjustments can be targeted with single neuron or single pair resolution, respectively. To each thalamic fiber or neuron in the model, we can assign an arbitrary spike train, triggering synaptic release from all of its synapses with anatomically determined

delays (from axonal conduction speed and path length). These mechanisms can target specific groups of neurons, for example, based on neuron properties, such as location, layer, m-type or e-type, or inhibitory subpopulation (determined based on me-type as in *Markram et al., 2015*; *Figure 2B*). We combined mechanisms to simulate various in vitro or in vivo experimental paradigms. Pathway lesions, for example, are simulated by selecting sets of pre- and postsynaptic neurons and removing all synaptic connections between them (*Figure 6—figure supplement 1A*). Simulated optogenetic stimulation, in addition to being targeted at specific groups of neurons, took into account the attenuation of light with depth for the wavelength being used. Neurons were then inhibited or excited according to the intensity of light reaching them (*Figure 6—figure supplement 1B*). Finally, sensory stimuli were simulated by generating instances of stimulus-specific stochastic spike trains activating the thalamic input fibers (*Figure 6—figure supplement 1C*).

## Optogenetic inhibition or activation

We model optogenetic inhibition or activation of a neuron population through a current injection at the soma of each cell in the population, with an intensity proportional to the cell's threshold current (see *Reva et al., 2023*; but with the technical caveat that we did not apply a hyperpolarizing holding current at the same time). To mimic the conditions of surface illumination, we considered the dependence of effective depolarization strength on cortical depth using a modified Beer–Lambert law approximation for the exponential attenuation of light intensity through scattering tissue (*Al-Juboori et al., 2013*; *Azimipour et al., 2014*, *Figure 6—figure supplement 1B*):

$$I(d) = I_0 \exp(-\mu_{eff}\, d) \tag{4}$$

where $I(d)$ describes the light intensity at depth $d$ (in mm), with a maximum light intensity $I_0$ (on the surface of the cortex) and an effective attenuation coefficient given by

$$\mu_{eff} = \sqrt{3\mu_a(\mu_a + \mu_s')} \tag{5}$$

Values for the absorption coefficient $\mu_a$ and the reduced scattering coefficient $\mu_s'$ were interpolated for the chosen wavelength from *Mesradi et al., 2013*. For implementation reasons, the targeted cells were grouped into depth bins and for each group the intensity at the center of the bin was used. Bins were equally distributed in terms of the injected current, and not in terms of depth. The value of $I_0$ was calibrated independently for each experimental paradigm.

## Modeling sensory inputs

Sensory inputs were simulated using a three-step procedure (*Figure 6—figure supplement 1C2*): First, a time series representing sensory stimulation is assigned to each selected thalamic fiber; second, the time series is optionally transformed with a fiber-dependent transfer function; third, a spike train is generated from the time series through a stochastic spiking process. Thalamic input fibers were associated with roughly columnar, overlapping volumes of the model that they formed synaptic connections in *Reimann et al., 2024*. Their centers were projected into the plane using a flat map of the model (*Bolaños-Puchet et al., 2024*), yielding $[x_\zeta, y_\zeta]$, the *flat locations* of each fiber $\zeta$. A sensory stimulus was defined based on fiber location as $\rho_\zeta(t) = \rho(x_\zeta, y_\zeta, t)$. The stimulus function could be partially stochastic and its value was evaluated for each time bin of the simulation (*Figure 6—figure supplement 1C1*). Next, a fiber could be associated with a transfer function $\upsilon_\zeta$ that transforms the results of $\rho$. If none is mentioned, identity was used as transfer function. Finally, a spiking process $\psi$ was used to instantiate a spike train for each fiber, based on its transformed time series. That is, the spike train associated with $\zeta$ was $\Gamma_\zeta = \psi(\upsilon_\zeta(\rho_\zeta))$. The processes used were stochastic, leading to different spike trains for fibers associated with the same time series. Different, specific $\rho$, $\upsilon$ and $\psi$ were used for different experimental paradigms that will be further described below.

## Recreating *Varani et al., 2022*

To study how input from L4 contributes to L2/3 subthreshold responses *Varani et al., 2022* used a 500-ms-long whisker hold paradigm, while patch-clamping PCs in L2/3 in anesthetized and awake mice. We encoded the whisker hold stimulus as a step function ($\rho_\zeta(t) = 1$, if 2000 ms $\leq$ t <2500 ms, 0 otherwise) in 10% of the VPM fibers within the seven-column subvolume. One of four transfer functions

were assigned to each fiber, based on the types of kinetic response properties of thalamic neurons identified in **Petersen et al., 2008**. The types were selective for whisker position ($v_{pos}$), velocity ($v_{vel}$), acceleration ($v_{acc}$), or direction ($v_{dir}$), and were implemented as

$$v_{pos}(t) = r_{max} \cdot \rho(t)$$

$$v_{vel}(t) = r_{max} \cdot (\rho(t+1) - \rho(t))$$

$$v_{acc}(t) = r_{max} \cdot (\rho(t+1) - 2\rho(t) + \rho(t-1))$$

$$v_{dir}(t) = r_{max} \cdot |\rho(t+1) - \rho(t)|_+$$

where $r_{max} = 150$ Hz denotes the firing rate of a thalamic fiber when its associated feature property is at the fiber's preferred value. Transfer functions were randomly assigned to fibers with the fractions identified in **Petersen et al., 2008** (**Figure 6A**, 11% coding for position and acceleration, 58% for velocity, and 20% for direction). The spiking process $\psi$ was an adapting Markov process (**Muller et al., 2007**) with an adaptation time constant of 100 ms. As the whisker movements were short lasting, rates were evaluated at submillisecond (0.1) resolution.

Similarly to before, only the seven-column subvolume (210k neurons) was used for the simulations, and the in vivo-like state was realized as $Ca^{2+} = 1.05$ mM, $R_{OU} = 0.4$, and $P_{FR} = 0.3$.

The optogenetic inhibition in this experiment targeted 95% (in line with **Varani et al., 2022**) of excitatory cells in layer 4. The authors found a few cells which also tested positive for Halo at the bottom of L3 as well, but as they did not quantify it, we decided not to target any lower L3 PCs in the in silico version of the experiment. Based on the 595 nm wavelength (yellow light), parameters were set to $\mu_a \approx 0.49 mm^{-1}$ and $\mu'_s \approx 4.12 mm^{-1}$. For the depth-based spatial binning of cells, five bins were used in L4. After scanning several values, $I_0$ was set to –200% as that reproduced the $\approx 10$ mV hyper-polarization of L4 PCs observed in vivo. In line with the in vivo experiment, the optogenetic stimulus ended in a (100 ms long) ramp to avoid rebound spikes (**Figure 6B**).

When going beyond reproducing the same experimental conditions and instead leveraging the in silico nature of our setup, synaptic pathways were lesioned by selecting the excitatory population in a given layer as the presynaptic population and the excitatory population in L2/3 as the postsynaptic population and not instantiating the connecting synapses during the simulation.

L2/3 PCs had to meet three criteria to be included in the subsequent analysis. Firstly, their activity was required to remain subthreshold during the 500-ms-long whisker hold stimulus and in 200-ms-long time windows before and after the stimulus, both in control and in silico optogenetic runs. Second, they had to be innervated by at least one (active) VPM fiber. Third, the derivative of their voltage trace had to cross the 1 mV/ms threshold in a 20 ms time window after stimulus onset in the control simula-tion. The last two were motivated by comparing subthreshold voltages to voltage traces from **Varani et al., 2022** that showed large, stimulus evoked EPSPs. Around 8% of L2/3 PCs in the central column met all the above criteria and their voltages were averaged to arrive to the traces shown in **Figure 6C and D**. Thus, unlike in the original analysis, cells rather than trials were averaged. The motivation for this approach is that while in vivo it is easier to repeat the same paradigm after establishing stable recording conditions in a given cell, in silico it is quicker to record from all cells in a single simulation, instead of repeating the stimulus several times.

### Recreating *Shapiro et al., 2022*

Since our model comprises the nbS1 and not the V1 brain area, we modeled visual drifting gratings in a more abstract way, without taking specifics of the visual system into account. In particular, we defined $\rho(x_\zeta, y_\zeta, t)$ as a spatio-temporal rate pattern corresponding to linear sinusoidal drifting grat-ings for a fraction $F_P$ of 937 VPM fibers projecting to a single simulated column (30k neurons, 520 $\mu m$ diameter). For all other VPM fibers, $\rho$ was zero. The gratings had a temporal frequency of $f_{temp} = 2$ Hz and a spatial frequency of 0.03 cycles/degree, which we translated to $f_{spat} = 0.001$ cycles/$\mu m$ by assuming a cortical magnification factor in rat of 30 $\mu m$/degree (**Gias et al., 2005**). In **Figure 6E**, the spatial grating patterns are illustrated at different points in time.

As in **Shapiro et al., 2022**, we used five contrast levels $C \in [0.06, 0.12, 0.24, 0.5, 1.0]$, which were defined as the Michelson contrast given the minimum and maximum luminance values $L_{min}$ and $L_{max}$:

$$C(L_{min}, L_{max}) = \frac{L_{min} - L_{max}}{L_{min} + L_{max}} \tag{6}$$

Since the physical quantity of luminance does not have a clear correspondence in our model, we used normalized luminance values $L_{min}$ and $L_{max}$ between 0.0 and 1.0 by computing the inverse Michelson contrast centered around a mean normalized luminance of 0.5. The resulting values of $L_{min}$ and $L_{max}$ where then scaled and shifted to the minimum and maximum rates of the sinusoidal modulation $R_{min}$ and $R_{max}$, respectively, such that the peak firing rate at contrast 1.0 was given by $R_{peak}$ and the mean of the modulation corresponding to the background firing rate at contrast 0.0 was given by $R_{bk}$. The resulting sinusoidal input rate signal can be written as

$$\rho_\zeta(t) = R(t, l_\zeta) = R_{min} + 0.5(R_{max} - R_{min})\left[1 + \sin(2\pi f_{temp}\, t - 2\pi f_{spat}\, l_\zeta)\right]\Big|_+ \tag{7}$$

where $t$ is the time in seconds, $l_\zeta$ the linear position of $\zeta$ within the grating, and $|_+$ denotes that firing rates are truncated to positive values only. *Figure 6F* illustrates a random series of rate signals corresponding to different contrasts. No transfer function was used, and spiking process $\psi$ was an adapting Markov process (*Muller et al., 2007*) with an adaptation time constant of 100 ms. Values for $F_P$, $R_{peak}$, and $R_{bk}$ were determined beforehand by a parameter optimization so that the resulting grating responses were qualitatively comparable with the ones reported by *Shapiro et al., 2022*; see the next subsection for details.

## Calibration of drifting grating stimulus

We ran a parameter scan to determine optimal values for $F_P$ (fraction of 937 VPM fibers projecting to the central column of our model to apply grating stimulus to), $R_{peak}$ (peak firing rate at contrast 1.0), and $R_{bk}$ (background firing rate at contrast 0.0). This calibration was done by running 45 simulations using all combinations of parameter values in the ranges $F_P \in [0.5, 0.75, 1.0]$, $R_{peak} \in [5.0, 10.0, 15.0, 20.0, 25.0]$ Hz, and $R_{bk} \in [0.05, 0.1, 0.2]$ Hz, and selecting the optimal combination amongst them. Each simulation lasted 40 seconds during which 20 contrast stimuli were presented for 1 second, followed by a 1 second (blank) inter-stimulus interval. Four contrast levels $C \in [0.06, 0.12, 0.5, 1.0]$ were presented five times each in random order, which was sufficient to fit sigmoidal tuning functions with four parameters (see *Equation 9*).

We observed that especially under strong stimulus conditions the response rates of PCs to the first and second cycles of the sinusoidal grating pattern (1-second stimulus at $f_{temp} = 2$ Hz) were quite different, with the second response largely attenuated due to synaptic depletion. Therefore, we extracted first and second peak firing rates $r_1$ and $r_2$, respectively, from the PSTHs computed with 1 ms resolution and 20 ms Gaussian smoothing (*Figure 6—figure supplement 3*). We defined a measure of the normalized peak difference as the Michelson contrast of the peak rates $r_1$ and $r_2$, given by

$$\hat{r}_{diff} = \frac{r_1 - r_2}{r_1 + r_2} \tag{8}$$

Additionally, we extracted average firing rates for each contrast level of the whole population of PCs within the full (1 second) as well as the first and second halves (0.5 second each) of the stimulation intervals. We then fitted sigmoidal tuning functions with parameters $c_{50}$, $m$, $n$, and $R_{max}$ (see *Equation 9*) to these average tuning responses (*Figure 6—figure supplement 5*). Finally, as summarized in *Figure 6—figure supplements 3 and 5*, we selected the best combination of parameters based on the following selection criteria:

- The peak firing rates in response to the grating stimulus should be sufficiently strong, covering a range of values including the one reported in *Shapiro et al., 2022*, *Figure 1D*. So we imposed the constraint that the maximum peak rates over the whole population of PCs of the first and second peaks $r_1$ and $r_2$ respectively should be at least $r_{th} = 30$ Hz at maximum contrast.
- The peak responses to the first and second cycle of the grating stimulus should not be too different. So we aimed for a low peak difference $\hat{r}_{diff}$ at maximum contrast.
- The overall tuning response of the PCs should be in a regime where the tuning curve would have a sigmoidal shape, meaning that it should be increasing but saturating with increasing contrasts. To fulfill this constraint, we aimed for
  - Low $c_{50}$, that is, the inflection point of the tuning curve would be at low contrasts.
  - High $n$, that is, a steep non-linear increase of the tuning curve until saturation.

In order to combine these criteria independent of the actual scales of $\hat{r}_{diff}$, $c_{50}$ and $n$, we computed their individual rank scores in decreasing ($\hat{r}_{diff}$, $c_{50}$: lower is better) or increasing ($n$: higher is better) order. We then selected the parameter combination with the highest product of rank scores, excluding the ones with peak firing rates $r_1$ and $r_2$ below $r_{th}$ and the ones with $c_{50}$ values close to 1.0 (border cases). We found an optimal parameter combination of $F_P = 1.0$, $R_{peak} = 10.0$ Hz, and $R_{bk} = 0.20$ Hz which we used throughout this in-silico experiment.

## Optogenetic stimulation

Optogenetic stimulation was targeted at either 1654 PV+ or 822 Sst+ interneurons in a single column. We used parameters $\mu_a \approx 0.46$ mm$^{-1}$ and $\mu_s' \approx 5.38$ mm$^{-1}$, based on the wavelength of 470 nm (blue light; see *Figure 6—figure supplement 1*). $I_0$ was increased from 0% to 300% in steps of 50%.

## Quantification of contrast tuning responses by sigmoidal functions

We quantified contrast tuning responses in the same way as described by *Shapiro et al., 2022* by least-squares fitting sigmoidal functions to the normalized tuning curves. Normalized tuning curves were obtained by computing the time-averaged firing rates of all 1-second stimulus intervals and dividing them by the mean baseline firing rate (i.e., w/o optogenetic stimulation) at the highest contrast level. The sigmoidal function was given by

$$R(c) = \frac{R_{max}\, c^n}{c^n + c_{50}^n} + m \tag{9}$$

where $R(c)$ describes the response amplitude at contrast $c$, $m$ is the baseline response at minimum contrast, $R_{max}$ is the maximum increase above baseline, $n$ defines the steepness of the curve, and $c_{50}$ is the contrast at half $R_{max}$. We used the coefficient of determination ($r^2$ score) as a measure of the goodness of fit.

## Detection of neurons with robust contrast tuning

We identified PV+, Sst+, and pyramidal neurons with robust contrast tuning behavior under all conditions. In the experimental study of *Shapiro et al., 2022*, tetrode recordings were used together with spike sorting. Correspondingly, we only considered neurons firing at rates above 0.5 Hz under all stimulus conditions, meaning they could potentially be detected by spike sorting (*Pedreira et al., 2012*). In addition, we considered neurons as being robustly tuned if they had strictly monotonically increasing tuning curves.

## Modeling the effects of optogenetic stimulation of interneurons

Direct photostimulation effects on interneurons were modeled by a divisive scaling model $R_{div}(c)$, a subtractive shifting model $R_{sub}(c)$, or a saturation additive model $R_{sat}(c)$ (*Shapiro et al., 2022*). For fitting these models, the parameters $c_{50}$, $m$, $n$, and $R_{max}$ of the underlying contrast tuning function $R(c)$ were kept constant at values obtained from baseline fits (i.e., w/o optogenetic stimulation). The divisive scaling model was defined as

$$R_{div}(c) = R(c)/g \tag{10}$$

with a scaling term $g$. The subtractive shifting model was given by

$$R_{sub}(c) = R(c) - h \Big|_+ \tag{11}$$

with a shifting term $h$ and rectification to rates equal or above zero. The saturation additive model was defined as

$$R_{sat}(c) = R(c) + \frac{S\, c^{-n}}{c^{-n} + c_{50}^{-n}} + A \tag{12}$$

with a saturation term $S$ and an additive term $A$. We used the $r^2$ score to measure the goodness of model fits.

Indirect effects on PCs receiving inhibitory input from optogenetically activated interneurons were modeled by a conductance-based model $R_{cond}(c)$, assuming a saturating additive model description of interneurons (*Shapiro et al., 2022*). Response rates under this conductance-based model were given by

$$R_{cond}(c) = \left[ \Delta V(c) - V_{th} \right]^3 \Big|_+ \tag{13}$$

with a spike threshold $V_{th} = 3.4$ mV and rectification to rates equal or above zero. The membrane potential $\Delta V(c)$ as a function of contrast was given by

$$\Delta V(c) = \frac{g_L R_L + g_E(c) R_E + g_I(c) R_I}{g_L + g_E(c) + g_I(c)} - V_r \tag{14}$$

with values for leak conductance $g_L = 6$ nS, leak reversal potential $R_L = -50$ mV, excitatory reversal potential $R_E = 0$ mV, inhibitory reversal potential $R_I = -65$ mV, and resting potential $V_r = -50$ mV as reported by *Shapiro et al., 2022*. The excitatory synaptic conductance was given by

$$g_E(c) = \frac{g_{E\_max} c^n}{c^n + c_{50}^n} + g_{E\_min} \tag{15}$$

with the excitatory conductances at low/high contrast given by $g_{E\_min}$ and $g_{E\_max}$, respectively. The inhibitory synaptic conductance was given by

$$g_I(c) = g_E(c) + \frac{S c^{-n}}{c^{-n} + c_{50}^{-n}} + \Delta g_{IE\_min} + A \tag{16}$$

with an inhibitory conductance offset $\Delta g_{IE\_min} = 2$ nS at low contrast relative to $g_{E\_min}$.

In a first step, parameters $S$ and $A$ were set to zero and the model parameters $c50$, $g_{E\_min}$, $g_{E\_max}$, and $n$ were fit to baseline responses of PCs. In a second step, those parameter values were kept constant and the model parameters $S$ and $A$ were fit to PC responses under photostimulation conditions. Again, we used the $r^2$ score to measure the goodness of model fits.

### Recreating *Prince et al., 2021*

We also compared the in vivo state of our model with the results of a recent in vitro study (*Prince et al., 2021*), which explored how neurons with different biophysical properties encode different types of signals. The in vitro study optically activated groups of 10 PCs in slices of L2/3 barrel cortex from mouse lines labeling fast-spiking (FS) and regular-spiking (RS) interneurons. Each PC was targeted individually and a binary signal was encoded either through changes in the rate of optical pulses applied to the 10 stimulus neurons, or through changes in the synchronicity of the pulses. The mutual information shared between the binary signal (when encoded either as changes in firing rate or synchronicity) and the firing rate of different inhibitory sub-types (recorded using whole-cell patch clamping in vitro) was analyzed.

While the in vitro stimuli used optical pulses targeted at single neurons with timings drawn from Poisson processes, it was not necessary to explicitly model such an optical stimulus in silico, as we can instead elicit spikes directly in the model. In the original study, the 10 stimulus neurons were uniformly separated in a grid-like pattern at horizontal and vertical distances of ~50 $\mu m$. The binary signal alternated between two states at random intervals of 2–7 seconds. For the rate coding paradigm, the timings of the optical pulses for the 10 stimulus neurons were drawn from 10 independent inhomogeneous Poisson processes, with the rates of optical activation varying between 5 Hz and 0.5 Hz for the up and down states, respectively (average firing rate: 2.7 Hz). For the synchronous case, the timings of the optical pulses for the 10 stimulus neurons were drawn from a single inhomogeneous Poisson process during the up state, or 10 independent inhomogeneous Poisson processes during the down state. For the synchronous case, the firing rates of the up and down state were both 2.7 Hz. We therefore tested stimulus encoding using a range of stimulus neuron counts: from 10 to 1000.

In our interpretation of the study, mutual information was measured between the binary signal and neural activity during a 0–5 ms and 5–50 ms window following a change in the signal. These represented 'early' and 'late' stimulus encoding windows following the change in the signal, respectively.

As the synchronous up state activated synchronous patterns at a rate of 2.7 Hz, we calculated that a synchronous pattern would only be activated during the early and late windows with probability 0.0135 ($= 2.7 * 5/1000$) and 0.1125 ($= 2.7 * 45/1000$), respectively. We therefore chose to use higher FR of 20 Hz for the synchronous up and down states, and to analyze mutual information between the binary signal and all 50 ms bins. To afford comparison with the rate code experiment type, we used 30 Hz and 10 Hz as the FRs of the up and down states respectively. We also compare stimulus coding during the first 50 ms following the stimulus change for both the synchronous and rate stimulus types.

## Analyzing mid-range connectivity and correlations

Neurons were split into hexagonal subvolumes of a specified diameter $d$ considering their locations in a flattened view (**Bolaños-Puchet et al., 2024**). Then, the number of mid-range connections within and between the hexagonal subvolumes were counted, yielding in $S$, the adjacency matrix of the resulting directed graph. Distances between hexagonal subvolumes were calculated by considering the centers of the hexagons in the flattened view. Furthermore, correlations of spiking activity of pairs of hexagonal subvolumes were also calculated. To that end, all spikes within a hexagonal subvolume were pooled and their number in 5 ms time bins; and lastly the Pearson correlations of the resulting time series were calculated. The correlation within a hexagonal subvolume was calculated similarly, but using the separate time series of the E and I populations. Together, this yielded $F$, the matrix of correlations of subvolumes. Based on $S$ and $F$, the expected correlations of mid-range inputs was calculated as follows: Let $i$ be a hexagonal subvolume and $S_i$ the column of $S$ associated with input counts into $i$. Then $P = S_i \cdot S_i^\top$ is in the matrix of counts of pairs of inputs for all pairs of hexagonal subvolumes. Combining $P$ and $F$ allows one to estimate the distribution of correlations of mid-range inputs into $i$, albeit at the population rather than single-cell level. For the correlation in **Figure 8A2, H, J1**, a hexagon size of $400 \mu m$ was used, while in **Figure 8I2** a smaller size of only $50 \mu m$ was used.

## Node participation

Given a connectivity graph $G$, a directed $n$-simplex in $G$ is a set of $n + 1$ nodes which are all all-to-all connected in $G$ in a feed-forward fashion, that is, such that any subset of these has a unique source and a unique sink (see **Reimann et al., 2017** for more details). The *n-node participation* of a node $v$ in $G$ is the number of directed $n$-simplices this node is part of. In particular, for $n = 1$, this is the total degree of the node in $G$. Thus, this can be thought of a generalization of the notion of degree that takes into account higher order interactions and has been shown to strongly correlate with other node centrality metrics **Sizemore et al., 2018**.

For any numeric property of neurons, for example, firing rate, we evaluate the effect of dimension on it by taking weighted averages across dimensions. That is for each dimension $k$, we take the weighted average of the property across neurons where the weights are given by node participation on dimension $k$. More precisely, let $N$ be the number of neurons and $\vec{V} \in \mathrm{R}^N$, be a vector of a property on all the neurons, for example, the vector of firing rates. Then in each dimension $k$ we compute

$$\mathbf{mean}_k = \frac{1}{N * \sum(\overrightarrow{Par_k})} \vec{V} \cdot \overrightarrow{Par_k},$$

where $\overrightarrow{Par_k}$ is the vector of node participation on dimension $k$ for all neurons and $\cdot$ is the dot product.

To measure the over and underexpression of the different m-types among those with the highest 5% of values of node participation, we used the hypergeometric distribution to determine the expected distribution of m-types in a random sample of the same size. More precisely, for each dimension $k$ and m-type $m$, let $N_{total}$ be the total number of neurons in the circuit, $N_m$ be the number of neurons of m-type $m$ in the circuit, $C_{top}$ be the number of neurons with the highest 5% values of node participation in dimension $k$, $C_m$ the number of neurons of mtype $m$ among these, and let $P = hypergeom(N_{total}, N_m, C_{top})$ be the hypergeometric distribution.

By definition, $P(x)$ describes the probability of sampling $x$ neurons of m-type $m$ in a random sample of size $C_{top}$. Therefore, using the cumulative distribution $F(x) = P(\text{Counts} \leq x)$, we can compute the p-values as follows:

$$\text{underexpression: } P(\text{Counts} \leq C_m) = F(C_m),$$

$$\text{overexpression: } P(\text{Counts} \geq C_m) = 1 - P(\text{Counts} < C_m - 1) = 1 - F(C_m - 1).$$

Small values indicate under and over representation, respectively.

## Quantification and statistical analysis

Details of all statistical analyses can be found in the figures and figure legends.

## Acknowledgements

The authors thank Fabien Delalondre, Adrien Devresse, Hugo Dictus, Juan B Hernando, Nicolas Cornu, Daniel Fernandez, Jeremy Fouriaux, and Ioannis Magnakaris for helpful discussions and contributions to the model; Eva Kenny for help with project management; Cyrille Favreau, Marwan Abdellah, and Nadir Roman for support with graphics and figure design and Karin Holm and Akiko Sato for support of manuscript development and helpful discussions. The authors would also like to thank Heiko Luhmann, Vicente Reyes Puerta, and Jhy-Jung Sun for supporting access to data used in the primary validation of the model. This study was supported by funding to the Blue Brain Project, a research center of the École polytechnique fédérale de Lausanne (EPFL), from the Swiss government's ETH Board of the Swiss Federal Institutes of Technology.

## Additional information

### Competing interests

Eilif B Muller: Reviewing editor, *eLife*. The other authors declare that no competing interests exist.

## Funding

| Funder | Grant reference number | Author |
|---|---|---|
| Board of the Swiss Federal Institutes of Technology | | James B Isbister |
| | | András Ecker |
| | | Christoph Pokorny |
| | | Sirio Bolaños-Puchet |
| | | Alexis Arnaudon |
| | | Omar Awile |
| | | Barros-Zulaica Natali |
| | | Jorge Blanco Alonso |
| | | Elvis Boci |
| | | Giuseppe Chindemi |
| | | Jean-Denis Courcol |
| | | Tanguy Damart |
| | | Thomas Delemontex |
| | | Alexander Dietz |
| | | Gianluca Ficarelli |
| | | Mike Gevaert |
| | | Joni Herttuainen |
| | | Genrich Ivaska |
| | | Weina Ji |
| | | Daniel Keller |
| | | James King |
| | | Pramod Kumbhar |
| | | Samuel Lapere |
| | | Polina Litvak |
| | | Darshan Mandge |
| | | Eilif B Muller |
| | | Fernando Pereira |
| | | Judit Planas |
| | | Rajnish Ranjan |
| | | Maria Reva |
| | | Armando Romani |
| | | Christian Rössert |
| | | Felix Schürmann |
| | | Vishal Sood |
| | | Aleksandra Teska |
| | | Anil Tuncel |
| | | Werner Van Geit |
| | | Matthias Wolf |
| | | Henry Markram |
| | | Srikanth Ramaswamy |
| | | Michael W Reimann |

The funders had no role in study design, data collection and interpretation, or the decision to submit the work for publication.

## Author contributions

James B Isbister, Michael W Reimann, Conceptualization, Resources, Data curation, Software, Formal analysis, Supervision, Validation, Investigation, Visualization, Methodology, Writing – original draft, Project administration, Writing – review and editing; András Ecker, Resources, Data curation, Software, Formal analysis, Validation, Investigation, Visualization, Methodology, Writing – original draft, Writing – review and editing; Christoph Pokorny, Conceptualization, Resources, Data curation, Software, Formal analysis, Validation, Investigation, Visualization, Methodology, Writing – original draft, Writing – review and editing; Sirio Bolaños-Puchet, Conceptualization, Resources, Data curation, Software, Formal analysis, Investigation, Visualization, Methodology, Writing – review and editing; Daniela Egas Santander, Conceptualization, Software, Formal analysis, Investigation, Visualization, Methodology, Writing – original draft, Writing – review and editing; Alexis Arnaudon, Resources, Data curation, Validation, Methodology; Omar Awile, Jorge Blanco Alonso, Thomas Delemontex, Alexander Dietz, Gianluca Ficarelli, Mike Gevaert, Joni Herttuainen, Genrich Ivaska, Weina Ji, Pramod Kumbhar, Fernando Pereira, Matthias Wolf, Software; Barros-Zulaica Natali, Tanguy Damart, Validation, Methodology; Elvis Boci, Visualization; Giuseppe Chindemi, Daniel Keller, Methodology; Jean-Denis Courcol, Resources, Software, Supervision, Project administration; James King, Vishal Sood, Data curation, Software; Samuel Lapere, Judit Planas, Christian Rössert, Software, Visualization; Polina Litvak, Validation, Investigation; Darshan Mandge, Maria Reva, Anil Tuncel, Software, Validation, Methodology; Eilif B Muller, Conceptualization, Data curation, Software, Methodology; Rajnish Ranjan, Data curation,

Supervision, Methodology; Armando Romani, Supervision; Felix Schürmann, Conceptualization, Resources, Data curation, Software, Funding acquisition, Project administration; Aleksandra Teska, Data curation, Software, Formal analysis, Validation, Investigation; Werner Van Geit, Data curation, Software, Validation, Methodology; Henry Markram, Conceptualization, Resources, Supervision, Funding acquisition, Methodology, Project administration; Srikanth Ramaswamy, Conceptualization, Supervision, Methodology

## Author ORCIDs
James B Isbister ⬩ https://orcid.org/0000-0002-1013-3013
András Ecker ⬩ https://orcid.org/0000-0001-9635-4169
Christoph Pokorny ⬩ https://orcid.org/0000-0002-9771-2180
Sirio Bolaños-Puchet ⬩ https://orcid.org/0000-0003-4049-6488
Daniela Egas Santander ⬩ https://orcid.org/0000-0001-9838-7992
Eilif B Muller ⬩ https://orcid.org/0000-0003-4309-8266
Judit Planas ⬩ https://orcid.org/0000-0002-8221-7988
Aleksandra Teska ⬩ https://orcid.org/0009-0005-6822-112X
Michael W Reimann ⬩ https://orcid.org/0000-0003-3455-2367

Reviewer #1 (Public review): https://doi.org/10.7554/eLife.99693.3.sa1
Author response https://doi.org/10.7554/eLife.99693.3.sa2

# Additional files

## Supplementary files
Supplementary file 1. Table of predictions made by the study.

Supplementary file 2. Synaptic parameters of excitatory pathways. Average class parameters are marked in bold and are used predictively (in lack of reference in vitro data) for the remaining pathways belonging to the same class. Physical dimensions are as follows: peak conductance $\hat{g}$: nS, depression and facilitation time constants $D$, $F$, and the EPSC $\tau_{decay}$: ms, the Hill coefficient of the nonlinear $[Ca^{2+}]_o$ dependent scaling of release probability $U_{Hill}$: mM, the release probability $U_{SE}$, the average number of vesicles in the release-ready pool $N_{RRP}$, and the NMDA/AMPA ratio $\hat{g}_{ratio}$ are dimensionless.

Supplementary file 3. Synaptic parameters of inhibitory pathways. Average class parameters are marked in bold and are used predictively (in lack of reference in vitro data) for the remaining pathways belonging to the same class. Physical dimensions are as follows: peak conductance $\hat{g}$: nS, depression and facilitation time constants $D$, $F$, and the IPSC $\tau_{decay}$: ms, the Hill coefficient of the nonlinear $[Ca^{2+}]_o$ dependent scaling of release probability $U_{Hill}$: mM, the release probability $U_{SE}$, the average number of vesicles in the release-ready pool $N_{RRP}$, and the $GABA_B/GABA_A$ ratio $\hat{g}_{ratio}$ are dimensionless.

Supplementary file 4. Synaptic parameters of thalamocortical synaptic pathways. Values taken from the internal connectivity (*Supplementary file 2*) are marked in bold. Physical dimensions are as follows: peak conductance $\hat{g}$: nS, depression and facilitation time constants $D$, $F$, and the EPSC $\tau_{decay}$: ms, the Hill coefficient of the nonlinear $[Ca^{2+}]_o$ dependent scaling of release probability $U_{Hill}$: mM, the release probability $U_{SE}$, the average number of vesicles in the release-ready pool $N_{RRP}$, and the NMDA/AMPA ratio $\hat{g}_{ratio}$ are dimensionless.

Supplementary file 5. Validation of PSP amplitudes. See *Figure 3B1*.

Supplementary file 6. Validation of first PSP amplitudes' CVs. See *Figure 3B2*.

Supplementary file 7. Validation of mPSC frequency. See *Figure 3E2*.

Supplementary file 8. Assumptions. Additionally, we assume that various input data generalizations between brain regions, organisms and animal ages do not affect validity. These are not listed again.

MDAR checklist

## Data availability
Simulatable models and simulation data are made available at the following links: - Anatomical and physiological model (nbS1): http://www.doi.org/10.7910/DVN/HISHXN - Anatomical and physiological model (seven column subvolume): http://www.doi.org/10.5281/zenodo.7930276 - SM-connectome

reproducing inhibitory trends from MICrONS with instructions of how to combine with the original seven column subvolume model: http://www.doi.org/10.5281/zenodo.10677883 - Simulation data: http://www.doi.org/10.5281/zenodo.17793038. The following data sourced used to construct the model are published in the following articles and made available at the following links: Anatomical model & Neuron morphology reconstructions (*Reimann et al., 2024*): http://www.doi.org/10.5281/zenodo.6906785 - Electrical neuron models & ion channel models (*Reva et al., 2023*): http://www.doi.org/10.5281/zenodo.7930276 - Volumetric atlases (*Bolaños-Puchet et al., 2024*): http://www.doi.org/10.5281/zenodo.8165004 - PSP amplitudes, CV of PSP amplitudes & mPSC frequencies, listed with their sources in the respective tables of *Supplementary files 5–7*.

The following datasets were generated:

| Author(s) | Year | Dataset title | Dataset URL | Database and Identifier |
|---|---|---|---|---|
| Reimann MW, Isbister JB, Ecker A, Pokorny C, Bolaños-Puchet S, Santander DE, Arnaudon A, Awile O, Barros-Zulaica N, Alonso JB, Boci E, Chindemi G, Courcol J-D, Damart T, Delemontex T, Dietz A, Ficarelli G, Gevaert M, Herttuainen J, Ivaska G, Ji W, Keller D, King J, Kumbhar P, Lapere S, Litvak P, Mandge D, Muller EB, Pereira F, Planas J, Ranjan R, Reva M, Romani A, Rössert C, Schürmann F, Sood V, Teska A, Tuncel A, Van Geit W, Wolf M, Markram H, Ramaswamy S | 2024 | BBP Somatosensory Cortex model - SONATA version | https://doi.org/10.7910/DVN/HISHXN | Harvard Dataverse, 10.7910/DVN/HISHXN |
| Isbister JB, Ecker A, Pokorny C, Bolaños-Puchet S, Santander DE, Arnaudon A, Awile O, Barros-Zulaica N, Alonso JB, Boci E, Chindemi G, Courcol J-D, Damart T, Delemontex T, Dietz A, Ficarelli G, Gevaert M, Herttuainen J, Ivaska G, Ji W, Keller D, King J, Kumbhar P, Lapere S, Litvak P, Mandge D, Muller EB, Pereira F, Planas J, Ranjan R, Reva M, Romani A, Rössert C, Schürmann F, Sood V, Teska A, Tuncel A, Van Geit W, Wolf M, Markram H, Ramaswamy S, Reimann MW | 2024 | A Model of Rodent Neocortical Micro- and Mesocircuitry | https://doi.org//10.5281/zenodo.7930276 | Zenodo, 10.5281/zenodo.7930276 |
| Pokorny C, Ecker A, Isbister JB, Reimann MW | 2024 | Rewired connectome of an SSCX model with inhibitory targeting based on trends found in MICrONS data | https://doi.org/10.5281/zenodo.10677883 | Zenodo, 10.5281/zenodo.10677883 |

*Continued on next page*

*Continued*

| Author(s) | Year | Dataset title | Dataset URL | Database and Identifier |
|---|---|---|---|---|
| Isbister J, Ecker A, Pokorny C, Egas Santander D, Bolaños Puchet S | 2025 | BBP Somatosensory Cortex Model Simulation Data | https://doi.org/10.5281/zenodo.17793038. | Zenodo, 10.5281/zenodo.17793038 |

The following previously published datasets were used:

| Author(s) | Year | Dataset title | Dataset URL | Database and Identifier |
|---|---|---|---|---|
| Reimann MW, Bolaños-Puchet S, Courcol J-D, Santander DE, Arnaudon A, Coste B, Delalondre F, Delemontex T, Devresse A, Dictus H, Dietz A, Ecker A, Favreau C, Ficarelli G, Gevaert M, Herttuainen J, Isbister JB, Kanari L, Keller D, King J, Kumbhar P, Lapere S, Lazovskis J, Lu H, Ninin N, Pereira F, Planas J, Pokorny C, Riquelme JL, Romani A, Shi Y, Smith JP, Sood V, Srivastava M, Van Geit W, Vanherpe L, Wolf M, Levi R, Hess K, Schürmann F, Muller EB, Markram H, Ramaswamy S | 2022 | A Model of Rat Non-barrel Somatosensory Cortex Anatomy | https://doi.org/10.5281/zenodo.6906784 | Zenodo, 10.5281/zenodo.6906784 |
| Bolaños-Puchet S, Teska A, Reimann MW | 2024 | Enhanced atlases and flatmaps of rodent neocortex | https://doi.org/10.5281/zenodo.8165004 | Zenodo, 10.5281/zenodo.8165004 |

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
