## [Editor Report · eLife Assessment]

This study reports a model of eight somatosensory areas of the rat cortex consisting of 4.2 million morphologically and electrically detailed neurons. The authors carry out simulation experiments aimed at understanding how multiscale organization of the cortical network shapes neural activity. While the reviewers found the results to be **solid**, they note that they could have likely been obtained using a much smaller portion of the model. Nonetheless, the release of the modeling platform represents a significant contribution to the field by providing a **valuable** resource for the scientific community.

---

## [Referee Report · Reviewer #1 (Public review)]

This paper presents a model of the whole somatosensory non-barrel cortex of the rat, with 4.2 million morphologically and electrically detailed neurons, with many aspects of the model constrained by a variety of data. The paper focuses on simulation experiments, testing a range of observations. These experiments are aimed at understanding how multiscale organization of the cortical network shapes neural activity.

Strengths

• The model is very large and detailed. With 4.2 million neurons and 13.2 billion synapses, as well as the level of biophysical realism employed, it is a highly comprehensive computational representation of the cortical network.

• Large scope of work - the authors cover a variety of properties of the network structure and activity in this paper, from dendritic and synaptic physiology to multi-area neural activity.

• Direct comparisons with experiments, shown throughout the paper, are laudable.

• The authors make a number of observations, like describing how high-dimensional connectivity motifs shape patterns of neural activity, which can be useful for thinking about the relations between the structure and the function of the cortical network.

• Sharing the simulation tools and a "large subvolume of the model" is appreciated.

Weaknesses

• A substantial part of this paper - the first few figures - focuses on single-cell and single-synapse properties, with high similarity to what was shown in Markram et al., 2015. Details may differ, but overall it is quite similar.

• Although the paper is about the model of the whole non-barrel somatosensory cortex, out of all figures, only one deals with simulations of the whole non-barrel somatosensory cortex. Most figures focus on simulations that involve one or a few "microcolumns". Again, it is rather similar to what was done in Markram et al., 2015 and constitutes relatively incremental progress.

• With a model like this, one has an opportunity to investigate computations and interactions across an extensive cortical network in an in vivo-like context. However, the simulations presented are not addressing realistic specific situations corresponding to animals performing a task or perceiving a relevant somatosensory stimulus. This makes the insights into roles of cell types or connectivity architecture less interesting, as they are presented for relatively abstract situations. It is hard to see their relationship to important questions that the community would be excited about - theoretical concepts like predictive coding, biophysical mechanisms like dendritic nonlinearities, or circuit properties like feedforward, lateral, and feedback processing across interacting cortical areas. In other words, what do we learn from this work conceptually, especially, about the whole non-barrel somatosensory cortex?

• Most of comparisons with in vivo-like activity are done using experimental data for whisker deflection (plus some from the visual stimulation in V1). But this model is for the non-barrel somatosensory cortex, so exactly the part of the cortex that has less to do with whiskers (or vision). Is it not possible to find any in vivo neural activity data from non-barrel cortex?

• The authors almost do not show raw spike rasters or firing rates. I am sure most readers would want to decide for themselves whether the model makes sense, and for that the first thing to do is to look at raster plots and distributions of firing rates. Instead, the authors show comparisons with in vivo data using highly processed, normalized metrics.

• While the authors claim that their model with one set of parameters reproduces many experimentally established metrics, that is not entirely what one finds. Instead, they provide different levels of overall stimulation to their model (adjusting the target "P_FR" parameter, with values from 0 to 1, and other parameters), and that influences results. If I get this right (the figures could really be improved with better organization and labeling), simulations with P_FR closer to 1 provide more realistic firing rate levels for a few different cases, however, P_FR of 0.3 and possibly above tends to cause highly synchronized activity - what the authors call bursting, but which also could be called epileptic-like activity in the network.

• The authors mention that the model is available online, but the "Resource availability" section does not describe that in substantial detail. As they mention in the Abstract, it is only a subvolume that is available. That might be fine, but more detail in appropriate parts of the paper would be useful.

Comments on revisions:

The authors addressed all my comments by revising and adding text as well as revising and adding some figures and videos. The limitations described in my previous review (above) mostly remain, but they are much better acknowledged and described now. These limitations can be addressed in the future work, whereas the current paper represents a step forward relative to the state of the art and provides a useful resource for the community.

Two minor points about the new additions to the paper:

(1) Something does not seem right in the sentence, "Unlike the Markram et al. (2015) model, the new model can also be exploited by the community and has already been used in a number of follow up papers studying (Ecker et al., 2024a,b; ...)". Should the authors remove "studying"?

(2) It is great that the authors added more plots and videos of the firing rates, but most of them show maximum-normalized rates, which sort of defeats the purpose. No scale on the y-axis is shown (it can be useful even for normalized data). And it is impossible to see anything for inhibitory populations.

These are minor points that may not need to be addressed. Overall, it is a nice study that is certainly useful for the field.

A great improvement is that the model is made fully available to the public.

---

## [Author Response]

The following is the authors’ response to the previous reviews.

**Public Reviews:**

**Reviewer #1 (Public review):**
This paper presents a model of the whole somatosensory non-barrel cortex of the rat, with 4.2 million morphologically and electrically detailed neurons, with many aspects of the model constrained by a variety of data. The paper focuses on simulation experiments, testing a range of observations. These experiments are aimed at understanding how the multiscale organization of the cortical network shapes neural activity.Strengths:(1) The model is very large and detailed. With 4.2 million neurons and 13.2 billion synapses, as well as the level of biophysical realism employed, it is a highly comprehensive computational representation of the cortical network.(2) Large scope of work - the authors cover a variety of properties of the network structure and activity in this paper, from dendritic and synaptic physiology to multi-area neural activity.(3) Direct comparisons with experiments, shown throughout the paper, are laudable.(4) The authors make a number of observations, like describing how high-dimensional connectivity motifs shape patterns of neural activity, which can be useful for thinking about the relations between the structure and the function of the cortical network.(5) Sharing the simulation tools and a "large subvolume of the model" is appreciated.

We thank the reviewer for these comments and are pleased they appreciated these aspects of the work.

Weaknesses:(1) A substantial part of this paper - the first few figures - focuses on single-cell and single-synapse properties, with high similarity to what was shown in Markram et al., 2015. Details may differ, but overall it is quite similar.

We thank the reviewer for this useful comment and agree that it is important to better highlight the incremental improvements to the model’s low-level physiology. The validity of any model can continuously be improved at all spatial scales and the validity of emergent network activity increases with improved validity at lower levels. For this reason, we felt it was valuable to improve the low-level physiology of the model.

Regarding neuron physiology, we have added the following in Section 2.1 on page 5:

“2.1 Improved modeling and validation of neuron physiology

Similarly to Markram et al. (2015), electrical properties of single neurons were modelled by optimizing ion channel densities in specific compartment-types (soma, axon initial segment (AIS), basal dendrite, and apical dendrite) (Figure 2B) using an evolutionary algorithm (IBEA; Van Geit et al., 2016) so that each neuron recreates electrical features of its corresponding electrical type (e-type) under multiple standardized protocols. Compared to Markram et al. (2015), electrical models were optimized and validated using (1) additional in vitro data, features and protocols, (2) ion channel and electrophysiological data corrected for the liquid junction potential, and (3) stochastic channels (StochKv3) now including inactivation profiles. The methodology and resulting electrical models are described in Reva et al. (2023) (see Methods), and generated quantitatively more accurate electrical activity, including improved attenuation of excitatory postsynaptic potentials (EPSPs) and back-propagating action potentials.”

And page 8:

“The new neuron models saw a 5-fold improvement in generalizability compared to Markram et al. (2015) (Reva et al., 2023).”

We have also made the descriptions of the improvements to synaptic physiology more explicit in Section 2.2 on page 9:

“2.2 Improved modeling and validation of synaptic physiology

The biological realism of synaptic physiology was improved relative to Markram et al. (2015) using additional data sources and by extending the stochastic version of the Tsodyks-Markram model (Tsodyks and Markram, 1997; Markram et al., 1998; Fuhrmann et al., 2002; Loebel et al., 2009) to feature multi-vesicular release, which in turn improved the accuracy of the coefficient of variations (CV; std/mean) of postsynaptic potentials (PSPs) as described in Barros-Zulaica et al. (2019) and Ecker et al. (2020). The model assumes a pool of available vesicles that is utilized by a presynaptic action potential, with a release probability dependent on the extracellular calcium concentration ([Ca2+]o; Ohana and Sakmann, 1998; Rozov et al., 2001; Borst, 2010). Additionally, single vesicles spontaneously release as an additional source of variability with a low frequency (with improved calibration relative to Markram et al. (2015)). The utilization of vesicles leads to a postsynaptic conductance with bi-exponential kinetics. Short-term plasticity (STP) dynamics in response to sustained presynaptic activation are either facilitating (E1/I1), depressing (E2/I2), or pseudo-linear (I3). E synaptic currents consist of both AMPA and NMDA components, whilst I currents consist of a single GABAA component, except for neurogliaform cells, whose synapses also feature a slow GABAB component. The NMDA component of E synaptic currents depends on the state of the Mg2+ block (Jahr and Stevens, 1990), with the improved fitting of parameters to cortical recordings from Vargas-Caballero and Robinson (2003) by Chindemi et al. (2022).”

(2) Although the paper is about the model of the whole non-barrel somatosensory cortex, out of all figures, only one deals with simulations of the whole non-barrel somatosensory cortex. Most figures focus on simulations that involve one or a few "microcolumns". Again, it is rather similar to what was done by Markram et al., 2015 and constitutes relatively incremental progress.

We thank the reviewer for this comment and have added the following text to the Discussion on page 33 to explain our rationale:

“In keeping with the philosophy of compartmentalization of parameters and continuous model refinement (see Introduction), it was essential to improve validity at the columnar scale (relative to Markram et al. (2015)) as part of demonstrating validity of the full nbS1. Indeed, improved parametrization and validation at smaller scales was essential to parameterizing background input which generated robust nbS1 activity within realistic [*Ca*^2+^]*o* and firing rate ranges. We view this as a major achievement, as it was unknown whether the model would achieve a stable and meaningful regime at the start of our investigation. Whilst we would have liked to go further, our primary goal was to publish a well characterized model as an open resource that others could use to undertake further in-depth studies. In this regard, we are pleased that the parametrization of the nbS1 model has already been used to study EEG signals (Tharayil et al., 2024), as well as propagation of activity between two subregions (Bolaños-Puchet and Reimann, 2024).”

We also make it clearer in the Introduction on page 4 that the improved validation of the emergent columnar regime was essential to stable activity at the larger scale:

“These initial validations demonstrated that the model was in a more accurate regime compared to Markram et al. (2015) – an essential step before testing more complex or larger-scale validations. For example, under the same parameterization we then observed selective propagation of stimulus-evoked activity to downstream areas, and…”

(3) With a model like this, one has an opportunity to investigate computations and interactions across an extensive cortical network in an in vivo-like context. However, the simulations presented are not addressing realistic specific situations corresponding to animals performing a task or perceiving a relevant somatosensory stimulus. This makes the insights into the roles of cell types or connectivity architecture less interesting, as they are presented for relatively abstract situations. It is hard to see their relationship to important questions that the community would be excited about - theoretical concepts like predictive coding, biophysical mechanisms like dendritic nonlinearities, or circuit properties like feedforward, lateral, and feedback processing across interacting cortical areas. In other words, what do we learn from this work conceptually, especially, about the whole non-barrel somatosensory cortex?

We thank the reviewer for this comment and agree that it would be very interesting to explore such topics. In the Introduction on page 4, we have updated the list of papers which have so far used the model for more in depth studies:

“…propagation of activity between cortical areas (Bolaños-Puchet and Reimann, 2024) the role of non-random connectivity motifs on network activity (Pokorny et al., 2024) and reliability (Egas Santander et al., 2024), the composition of high-level electrical signals such as the EEG (Tharayil et al., 2024), and how spike sorting biases population codes (Laquitaine et al., 2024).”

In the Discussion on page 33 we also add our additional thoughts on this topic:

“Whilst we would have liked to go further, our primary goal was to publish a well characterized model as an open resource that others could use to undertake further in-depth studies. In this regard, we are pleased that the parametrization of the nbS1 model has already been used to study EEG signals (Tharayil et al., 2024), as well as propagation of activity between two subregions (Bolaños-Puchet and Reimann, 2024). Investigation, improvement and validation must be continued at all spatial scales in follow up papers with detailed description, figures and analysis, which cannot be covered in this manuscript. Each new study increases the scope and validity of future investigations. In this way, this model and paper act as a stepping stone towards more complex questions of interest to the community such as perception, task performance, predictive coding and dendritic processing. This was similar for Markram et al. (2015) where the initial paper was followed by more detailed studies. Unlike the Markram et al. (2015) model, the new model can also be exploited by the community and has already been used in a number of follow up papers studying (Ecker et al., 2024a,b; Bolaños-Puchet and Reimann, 2024; Pokorny et al., 2024; Egas Santander et al., 2024; Tharayil et al., 2024; Laquitaine et al., 2024). We believe that the number of use cases for such a general model is vast, and is made larger by the increased size of the model.”

(4) Most comparisons with in vivo-like activity are done using experimental data for whisker deflection (plus some from the visual stimulation in V1). But this model is for the non-barrel somatosensory cortex, so exactly the part of the cortex that has less to do with whiskers (or vision). Is it not possible to find any in vivo neural activity data from the non-barrel cortex?

We agree with the reviewer that this is a weakness. We have expanded our discussion of the need to mix data sources to also consider our view for network level activity:

“This paper and its companion paper serve to present a methodology for modeling micro- and mesoscale anatomy and physiology, which can be applied for other cortical regions and species. With the rapid increase in openly available data, efforts are already in progress to build models of mouse brain regions with reduced reliance on data mixing thanks to much larger quantities of available atlas-based data. This also includes data for the validation of emergent network level activity. Here we chose to compare network-level activity to data mostly from the barrel cortex, as well as a single study from primary visual cortex. Whilst a lot of the data used to build the model was from the barrel cortex, the barrel cortex also represents a very well characterized model of cortical processing for simple and controlled sensory stimuli. The initial comparison of population-wise responses in response to accurate thalamic input for single whisker deflections was essential to demonstrating that the model was closer to in vivo, and we were unaware of similar data for nonbarrel somatosensory regions. Moreover, our optogenetic & lesion study demonstrated the capacity to compare and extend studies of canonical cortical processing in the whisker system.”

(5) The authors almost do not show raw spike rasters or firing rates. I am sure most readers would want to decide for themselves whether the model makes sense, and for that, the first thing to do is to look at raster plots and distributions of firing rates. Instead, the authors show comparisons with in vivo data using highly processed, normalized metrics.

We thank the reviewer for this comment and agree that better visualizations of the network activity under different conditions is essential for helping the reader assess the work. In addition to raster plots in Video 1, Video 3, Fig 6, Fig 5C, Fig S9a, S16a, we have additionally:

a) Changed the histograms of spontaneous activity in Fig 4G on page 13 to raster plots for the seven column subvolume for two contrasting meta-parameter regimes.

b) Added 4 new videos (Video 6a,b and 8a,b) showing all spontaneous and evoked meta-parameter combinations in hex0 and hex39 of the nbS1:

We have added improved plots showing the distributions of firing rates in the seven column subvolume on page 74:

With more detailed consideration in the Results on page 15:

“Long-tailed population firing rate distributions with means ∼ 1Hz

To study the firing rate distributions of different subpopulations and m-types, we ran 50s simulations for the meta-parameter combinations: [*Ca*^2+^]*o*: 1.05mM, *ROU*: 0.4,*PFR*: 0.3, 0.7 (Figure S4). Different subpopulations showed different sparsity levels (proportion of neurons spiking at least once) ranging from 6.6 to 42.5%. Wohrer et al. (2013) considered in detail the biases and challenges in obtaining ground truth firing rate distributions in vivo, and discuss the wide heterogeneity of reports in different modalities using different recording techniques. They conclude that most evidence points towards longtailed distributions with peaks just below 1Hz. We confirmed that spontaneous firing rate distributions were long-tailed (approximately lognormally distributed) with means on the order of 1Hz for most subpopulations. Importantly the layer-wise means were just below 1Hz in all layers for the *PFR* = 0.3 meta-parameter combination. Moreover, our recent work applying spike sorting to extracellular activity using this meta-parameter combination found spike sorted firing rate distributions to be lognormally distributed and very similar to in vivo distributions obtained using the same probe geometry and spike sorter (Laquitaine et al., 2024).

(6) While the authors claim that their model with one set of parameters reproduces many experimentally established metrics, that is not entirely what one finds. Instead, they provide different levels of overall stimulation to their model (adjusting the target "P_FR" parameter, with values from 0 to 1, and other parameters), and that influences results. If I get this right (the figures could really be improved with better organization and labeling), simulations with*PFR* closer to 1 provide more realistic firing rate levels for a few different cases, however, *PFR* of 0.3 and possibly above tends to cause highly synchronized activity - what the authors call bursting, but which also could be called epileptic-like activity in the network.

We thank the reviewer for this comment. We can now see that the motivation for *PFR* parameter was introduced very briefly in the results and that the results of the calibration and analysis of the spontaneous activity regime are not interpreted in relation to this parameter.

To address this, we have given more detail where it is first introduced in the Results on page 12:

“to account for uncertainty in the firing rate bias during spontaneous activity from extracellular spike sorted recordings…”

We then reconsider that it represents an unknown bias when interpreting the calibration and spontaneous activity results on page 15:

“We reemphasize that the [*Ca*^2+^]*o*, *ROU* and *PFR* meta-parameters account for uncertainty of in vivo extracellular calcium concentration, the nature of inputs from other brain regions and the bias of extracellularly recorded firing rates. Whilst estimates for [*Ca*^2+^]*o* are between 1.0 - 1.1mM (Jones and Keep, 1988; Massimini and Amzica, 2001; Amzica et al., 2002; Gonzalez et al., 2022) and estimates for PFR are in the range of 0.1 - 0.3 (Olshausen and Field, 2006), combinations of these parameters supporting in vivo-like stimulus responses in later sections will offer a prediction for the true values of these parameters. Both these later results and our recent analysis of spike sorting bias using this model (Laquitaine et al., 2024) predict a spike sorting bias corresponding to *PFR* ∼ 0.3, confirming the prediction of Olshausen and Field (2006).”

And in relation to the stimulus evoked responses on page 17:

“Specifically, simulations with PFR from 0.1 to 0.5 robustly support realistic stimulus responses, with the middle of this range (0.3) corresponding with estimates of in vivo recording bias; both the previous estimates of Olshausen and Field (2006) and from a spike sorting study using this model (Laquitaine et al., 2024).”

Following these considerations, the remainder of the experiments using the seven column subvolume only use a single meta-parameter on page 19.

For the full nbS1 we further discuss the importance of a P_FR value between 0.1 and 0.3 in the Results on page 26:

“Stable spontaneous activity only emerges in nbS1 at predicted in vivo firing rates

After calibrating the model of extrinsic synaptic input for the seven column subvolume, we tested to what degree the calibration generalizes to the entire nbS1. Notably, this included the addition of mid-range connectivity (Reimann et al., 2024). The total number of local and mid-range synapses in the model was 9138 billion and 4075 billion, i.e., on average full model simulations increased the number of intrinsic synapses onto a neuron by 45%. Particularly, we ran simulations for *PFR</i ∈ [0.1, 0.15, ..., 0.3]* using the OU parameters calibrated for the seven column subvolume for [Ca2+]o = 1.05mM and ROU = 0.4. Each of these full nbS1 simulations produced stable non-bursting activity (Figure 8A), except for the simulation for PFR</i = 0.3, which produced network-wide bursting activity (Video 6). Activity levels in the simulations of spontaneous activity were heterogeneous (Figure 8B, Video 7). In some areas, firing rates were equal to the target PFR, whilst in others they increased above the target (Figure 8C). In the more active regions, mean firing rates (averaged over layers) were on the order of 30-35% of the in vivo references for the maximum non-bursting PFR simulation (target PFR : 0.25). This range of firing rates again fits with the estimate of firing rate bias from our paper studying spike sorting bias (Laquitaine et al., 2024) and the meta-parameter range supporting realistic stimulus responses in the seven column subvolume. This also predicts that the nbS1 cannot sustain higher firing rates without entering a bursting regime.

Finally, we also added to our discussion of biases in extracellular firing rates in the Discussion on page 32:

“This is also inline with our recent work using the model, which estimated a spike sorting bias corresponding to PFR = 0.3 using virtual extracellular electrodes (Laquitaine et al., 2024).”

We also thank the reviewer for pointing out that we did not define the term “bursting” in the main text. We have added the following definition and discussion in the Results on page 15:

“Note that the most correlated meta-parameter combination [Ca^2+^]_o_: 1.1mM, *ROU*: 0.2, *PFR*: 1.0 produced network-wide “bursting” activity, which we define as highly synchronous all or nothing events (Video 1). Such activity, which may be characteristic of epileptic activity, can be studied with the model but is not the focus of this study.”

(7) The authors mention that the model is available online, but the "Resource availability" section does not describe that in substantial detail. As they mention in the Abstract, it is only a subvolume that is available. That might be fine, but more detail in appropriate parts of the paper would be useful.

Firstly, we are pleased to say that the full nbS1 model is now available to download, in addition to the seven hexagon subvolume. In the manuscript, we have:

a) Added to the Introduction at the bottom of page 4:

“To provide a framework for further studies and integration of experimental data, the full model is made available with simulation tools, as well as a smaller subvolume with the optional new connectome capturing inhibitory targeting rules from electron microscopy”.

b) Updated the open source panel of Figure 1:

Secondly, we thank the reviewer for noticing that the description of the available model is not well described in the “Resource availability” statement and have addressed this by:

a) Adding the following to the “Resource availability” statement on page 36:

“Both the full nbS1 model and smaller seven hexagon subvolume are available on Harvard Dataverse and Zenodo respectively in SONATA format (Dai et al., 2020) with simulation code. DOIs are listed under the heading ``Final simulatable models'' in the Key resources table. An additional link is provided to the SM-Connectome with instructions on how to use it with the seven hexagon subvolume model.”

b) Creating a new subheading in the “Key resources table” titled: “Final simulatable models” to make it clearer which links refer to the final models.

**Reviewer #2 (Public review):**
Summary:This paper is a companion to Reimann et al. (2022), presenting a large-scale, data-driven, biophysically detailed model of the non-barrel primary somatosensory cortex (nbS1). To achieve this unprecedented scale of a bottom-up model, approximately 140 times larger than the previous model (Markram et al., 2015), they developed new methods to account for inputs from missing brain areas, among other improvements. Isbister et al. focus on detailing these methodological advancements and describing the model's ability to reproduce in vivo-like spontaneous, stimulus-evoked, and optogenetically modified activity.Strengths:The model generated a series of predictions that are currently impossible in vivo, as summarized in Table S1. Additionally, the tools used in this study are made available online, fostering community-based exploration. Together with the companion paper, this study makes significant contributions by detailing the model's constraints, validations, and potential caveats, which are likely to serve as a basis for advancing further research in this area.

We thank the reviewer for these comments, and are pleased they appreciate these aspects of the work.

Weaknesses:That said, I have several suggestions to improve clarity and strengthen the validation of the model's in vivo relevance.Major:(1) For the stimulus-response simulations, the authors should also reference, analyze, and compare data from O'Connor et al. (2010; https://pubmed.ncbi.nlm.nih.gov/20869600/) and Yu et al .(2016; https://pubmed.ncbi.nlm.nih.gov/27749825/) in addition to Yu et al. 2019, which is the only data source the authors consider for an awake response. The authors mentioned bias in spike rate measurements, but O'Connor et al. used cell-attached recordings, which do not suffer from activity-based selection bias (in addition, they also performed Ca2+ imaging of L2/3). This was done in the exact same task as Yu et al., 2019, and they recorded from over 100 neurons across layers. Combining this data with Yu et al., 2019 would provide a comprehensive view of activity across layers and inhibitory cell types. Additionally, Yu et al. (2016) recorded VPM neurons in the same task, alongside whole-cell recordings in L4, showing that L4 PV neurons filter movement-related signals encoded in thalamocortical inputs during active touch. This dataset is more suitable for extracting VPM activity, as it was collected under the same behavior and from the same species (Unlike Diamond et al., 1992, which used anesthetized rats). Furthermore, this filtering is an interesting computation performed by the network the authors modeled. The validation would be significantly strengthened and more biologically interesting if the authors could also reproduce the filtering properties, membrane potential dynamics, and variability in the encoding of touch across neurons, not just the latency (which is likely largely determined by the distance and number of synapses).

We thank the reviewer for pointing out these very useful studies. We have taken on board this suggestion for a future model of the mouse barrel cortex.

(2) The authors mention that in the model, the response of the main activated downstream area was confined to L6. Is this consistent with in vivo observations? Additionally, is there any in vivo characterization of the distance dependence of spiking correlation to validate Figure 8I?

We are not aware of data confirming the propagation of activity to downstream areas being confined to layer 6 but have considered the connectivity further between these two regions on page 27, as well as studying this further in follow up work:

“Stable propagation of evoked activity through mid-range connectivity only emerges in nbS1 at predicted in vivo firing rates

We repeated the previous single whisker deflection evoked activity experiment in the full model, providing a synchronous thalamic input into the forelimb sub-region (S1FL; Figure 8E; Video 8 & 9). Responses in S1FL were remarkably similar to the ones in the seven column subvolume, including the delays and decays of activity (Figure 8F). However, in addition to a localized primary response in S1FL within 350μm of the stimulus, we found several secondary responses at distal locations (Figure 8E; Video 9), which was suggestive of selective propagation of the stimulus-evoked signal to downstream areas efferently connected by mid-range connectivity. The response of the main activated downstream area (visible in Figure 8E) was confined to L6 (Figure 8G). In a follow up study using the model to explore the propagation of activity between cortical regions (Bolaños-Puchet and Reimann, 2024), it is described how the model contains both a feedforward projection pattern, which projects to principally to synapses in L1 & L23, and a feedback type pattern, which principally projects to synapses in L1 & L6. On visualizing the innervation profile from the stimulated hexagon to the downstream hexagon we can see that we have stimulated a feedback pathway (Figure S16)”

With referenced Figure S16 on page 85:

We did find in vivo evidence of similar layer-wise and distance dependence of correlations in the somatosensory cortex discussed on page 27 of the Results:

“The distance dependence of correlations followed a similar profile to that observed in a dataset characterizing spontaneous activity in the somatosensory cortex (Reyes-Puerta et al., 2015a) (compare red line in Figure 8I with Figure S16). In the in vivo dataset spiking correlation was also low but highest in lower layers, with short “up-states” in spiking activity constrained to L5 & 6 (see Figure 1E,F in (Reyes-Puerta et al., 2015a)). In the model, they are constrained to L6.”

With Figure S16a on page 85 showing the distance dependence of correlations in the anaesthetized barrel cortex during spontaneous activity (digitization from the reference paper):

(3) Across the figures, activity is averaged across neurons within layers and E or I cell types, with a limited description of single-cell type and single-cell responses. Were there any predictions regarding the responses of particular cell types that significantly differ from others in the same layer? Such predictions could be valuable for future investigations and could showcase the advantages of a data-driven, biophysically detailed model.

We thank the review for this comment. In addition to new analyses at higher granularity addressed in other comments, we have added the following comparison of stimulus-evoked membrane potential dynamics in different subpopulations for the original connectome and SM-connectome in Figure 7 on page 24.

This gave interesting results discussed in a new subsection on page 26:

“EM targeting trends hyperpolarize Sst+ and HT3aR+ late response, and disinhibit L5/6 E

Studying somatic membrane potentials for different subpopulations in response to whisker deflections shows that PV+, L23E and L4E subpopulations are largely unaffected in the SM-connectome (Figure 7E). Interestingly, Sst+ and 5HT3aR+ subpopulations show a strong hyperpolarization in the late response that isn’t present in the original connectome. Interestingly, this corresponds with a stronger late response in L5/6 E populations, which could be caused by disinhibition due to the Sst+ and 5HT3aR+ hyperpolarization. This could be explored further in follow up studies using our connectome manipulator tool (Pokorny et al., 2024).”

(4) 2.4: Are there caveats to assuming the OU process as a model for missing inputs? Inputs to the cortex are usually correlated and low-dimensional (i.e., communication subspace between cortical regions), but the OU process assumes independent conductance injection. Can (weakly) correlated inputs give rise to different activity regimes in the model? Can you add a discussion on this?

We agree with the reviewer that there are caveats to assuming an OU process for the model of missing inputs and have added the following to the Discussion on page 31:

“The calibration framework could optimize per population parameters for other compensation methods, whilst still offering an interpretable spectrum of firing rate regimes at different levels of P_FR_. For example, more realistic compensation schemes could be explored which introduce (a) correlations between the inputs received by different neurons and (b) compensation distributed across dendrites, as well as at the soma. We predict that such changes would make spontaneous activity more correlated at the lower spontaneous firing rates which supported in vivo like responses (P_FR_ : 0.1 − 0.5), which would in turn make stimulus-responses more noise correlated.”

(5) 2.6: The network structure is well characterized in the companion paper, where the authors report that correlations in higher dimensions were driven by a small number of neurons with high participation ratios. It would be interesting to identify which cell types exhibit high node participation in high-dimensional simplices and examine the spiking activity of cells within these motifs. This could generate testable predictions and inform theoretical cell-type-specific point neuron models for excitatory/inhibitory balanced networks and cortical processing.

We thank the reviewer for this suggestion. We have added two supplementary figures to address this suggestion, which are discussed in the Results on Page 16:

“Additionally, we studied the structural effect on the firing rate (here measured as the inverse of the inter-spike interval, ISI, which can be thought of as a proxy of non-zero firing rate). We found that for the connected circuit, the firing rate increases with simplex dimension; in contrast with the disconnected circuit, where this relationship remains flat (see Figure S6 red vs. blue curves and Methods).

This also demonstrates high variability between neurons, in line with biology, both structurally (Towlson et al., 2013; Nigam et al., 2016) and functionally (Wohrer et al., 2013; Buzs´aki and Mizuseki, 2014). We next identified the cell types that are overexpressed in the group of neurons that have the 5% highest values of node participation across dimensions (Figure S7). This could inform theoretical point neuron models with cell-type specificity, for example. We found that while in dimension one (i.e., node degree) this consists mostly of inhibitory cells, in higher dimensions the cell types concentrate in layers 4, 5 and 6, especially for TPC neurons. This is in line with our structural layer-wise findings in Figure 8B in Reimann et al. (2024).”

Which reference new Figures S6 and S7:

With the methodology for S6 described on page 49 of the Methods:

“For any numeric property of neurons, e.g., firing rate, we evaluate the effect of dimension on it by taking weighted averages across dimensions. That is for each dimension k, we take the weighted average of the property across neurons where the weights are given by node participation on dimension k. More precisely, let N be the number of neurons and −→V ∈ RN, be a vector of a property on all the neurons e.g., the vector of firing rates. Then in each dimension k we compute\begin{document}$$\displaystyle  \operatorname{mean}_{k}=\frac{1}{N * \sum\left(\overrightarrow{P a r_{k}}\right)} \vec{V} \cdot \overrightarrow{P a r_{k}}$$\end{document}

Where \begin{document}$\overrightarrow{P_{a r k}}$\end{document} is the vector of node participation on dimension k for all neurons and · is the dot product.

To measure the over and underexpression of the different m-types among those with the highest 5% of values of node participation, we used the hypergeometric distribution to determine the expected distribution of m-types in a random sample of the same size. More precisely, for each dimension k and m-type m, let N_total_ be the total number of neurons in the circuit, Nm be the number of neurons of m-type m in the circuit, Ctop be the number of neurons with the highest 5% values of node participation in dimension k, Cm the number of neurons of mtype m among these, and let *P = hypergeom(Ntotal</sub<,Nm,Ctop)* be the hypergeometric distribution.

By definition, *P(x)* describes the probability of sampling *x* neurons of m-type *m* in a random sample of size C_top_. Therefore, using the cumulative distribution F(x) = *P*(Counts ≤ x), we can compute the p-values as follows:\begin{document}$$\displaystyle \left( C_{m}-1\right)=1-F\left(C_{m}-1\right)$$\end{document}

Small values indicate under and over representation respectively….”

Minor:(1) Since the previous model was published in 2015, the neuroscience field has seen significant advancements in single-cell and single-nucleus sequencing, leading to the clustering of transcriptomic cell types in the entire mouse brain. For instance, the Allen Institute has identified ~10 distinct glutamatergic cell types in layer 5, which exceeds the number incorporated into the current model. Could you discuss (1) the relationship between the modeled me-types and these transcriptomic cell types, and (2) how future models will evolve to integrate this new information? If there are gaps in knowledge in order to incorporate some transcriptome cell types into your model, it would be helpful to highlight them so that efforts can be directed toward addressing these areas.

We thank the reviewer for this suggestion, particularly the idea to describe what types of data would be valuable towards improving the model in future. We have added the following to the Discussion on page 33:

“In our previous work (Roussel et al., 2023) we linked mouse inhibitory me-models to transcriptomic types (t-types) in a whole mouse cortex transcriptomic dataset (Gouwens et al., 2019). This can provide a direct correspondence in future large-scale mouse models. As we model only a single electrical type for pyramidal cells there is no one-to-one correspondence between our me-models and the 10 different pyramidal cell types identified there. We are not currently aware of any method which can recreate the electrical features of different types of pyramidal cells using only generic ion channel models. To achieve the firing pattern behavior of more specific electrical types, usually ion channel kinetics are tweaked, and this would violate the compartmentalization of parameters. In future we hope to build morpho-electric-transcriptomic type (met-type) models by selecting gene-specific ion channel models (Ranjan et al., 2019, 2024) based on the met-type’s gene expression. Data specific to different neuron sections (i.e. soma, AIS, apical/basel dendrites) of different met-types, such as gene expression, distribution of ion channels, and voltage recordings under standard single cell protocols would be particularly useful.”

(2) For the optogenetic manipulation, it would be interesting if the model could reproduce the paradoxical effects (for example, Mahrach et al. reported paradoxical effects caused by PV manipulation in S1; https://pubmed.ncbi.nlm.nih.gov/31951197/). This seems a more relevant and non-trivial network phenomenon than the V1 manipulation the authors attempted to replicate.

We thank the reviewer for this valuable idea. Indeed, our model is able to reproduce paradoxical effects under certain conditions. We added the following new supplementary Figure S12 demonstrating this finding (black arrows).

Which we discuss in the Results on page 22:

“However, at high contrasts, we observed a paradoxical effect of the optogenetic stimulation on L6 PV+ neurons, reducing their activity with increasing stimulation strength (Figure S12B; cf. Mahrach et al. (2020)). This effect did not occur under grey screen conditions (i.e., at contrast 0.0) with a constant background firing rate of 0.2 Hz or 5 Hz respectively (not shown). The individual…”

and added to the Discussion on page 32:

“Also, we predicted a paradoxical effect of optogenetic stimulation on L6 PV+ interneurons, namely a decrease in firing with increased stimulus strength. This is reminiscent of the paradoxical responses found by Mahrach et al. (2020) in the mouse anterior lateral motor cortex (in L5, but not in L2/3) and barrel cortex (no layer distinction) respectively. While Mahrach et al. (2020) conducted their recordings in awake mice not engaged in any behavior, we observed this effect only when drifting grating patterns with high contrast were presented. Nevertheless, consistent with their findings, we found the effect only in deep but not in superficial layers, and only for PV+ interneurons but not for PCs. Our model could therefore be used to improve the understanding of this paradoxical effect in follow up studies. These examples demonstrate that the approach of modeling entire brain regions can be used to further probe the topics of the original articles and cortical processing.”

**Recommendations for the authors:**

**Reviewer #1 (Recommendations for the authors):**
My specific comments are in the Public Review. The summarizing point is that this is a sprawling paper, and it is easy for readers to get confused. Focusing on specific connections between known functional properties and findings in this model, especially for the full-scale model, will be helpful.

We thank the reviewer for this comment and for their related recommendation (4) below, and have added subheadings through-out the results.

**Reviewer #2 (Recommendations for the authors):**
(1) P4. What are the 10 free parameters?

We thank the reviewer for pointing out that it would be useful to summarize the 10 parameters at this stage of the text, and have adjusted the sentence to:

“As a result, the emerging in-vivo like activity is the consequence of only 10 free parameters representing the strength of extrinsic input from other brain regions into 9 layer-specific excitatory and inhibitory populations, and a parameter controlling the noise structure of this extrinsic input.”

(2) Table 1 and S1 are extremely useful. Could you provide a table summarizing the major assumptions or gaps in the model, their potential influence on the results, and possible ways to collect data that could support or challenge these assumptions? Currently, this information is scattered throughout the manuscript.

We thank the reviewer for this very useful suggestion and have added a Table S8 on page 68:

(3) Figure 4F is important, but the legend is unclear. What is the unit on the x-axis? The values seem too large to represent per-neuron measurements.

Thank you to the reviewer for raising this. Indeed the values are estimated mean numbers of missing number synapses per neuron by population. Such numbers are difficult to estimate but we have further discussed our rationale, justification and consideration of whether these numbers are accurate in the Results, as follows:

“Heterogeneity in synaptic density within and across neuron classes and sections makes estimating the number of missing synapses challenging (DeFelipe and Fariñas, 1992). Changing the assumed synaptic density value of 1.1 synapses/μm would only change the slope of the relationship, however. Estimates of mean number of existing and missing synapses per population were within reasonable ranges; even the larger estimate for L5 E (due to higher dendritic length; Figure S3) was within biological estimates of 13,000 ± 3,500 total afferent synapses (DeFelipe and Fariñas, 1992).”

This text references the new supplementary Figure S3:

Moreover, these numbers represent the number of synapses, rather than the number of connections. The number of connections is usually used for quantifications such as indegree, and are usually much lower.

We have also updated the caption and axis labels of the original figure:

(4) Including additional subsections or improving the indexing in the Results section could be beneficial. In its current format, it's difficult to distinguish where the model description ends and where the validation begins. Some readers may want to focus more on the validation than other parts, so clearer segmentation would improve readability.

We have addressed this comment with the opening comment in the authors “Recommendations for authors”.

(5) P4. 2nd paragraph. Original vs rewired connectome. The term "rewired connectome" may give the impression that it refers to an artificial manipulation rather than a modification based on the latest data. It might be helpful to use a different term (e.g., SM-connectome as described later in the paper?).

We have adjusted the text in the introduction:

“Additionally, we generated a new connectome which captured recently characterized spatially-specific targeting rules for different inhibitory neuron types (Schneider-Mizell et al., 2023) in the MICrONS electron microscopy dataset (MICrONS-Consortium et al., 2021), such as increased perisomatic targeting by PV+ neurons, and increased targeting of inhibitory populations by VIP+ neurons. Comparing activity to the original connectome gave predictions about the role of these additional targeting rules.”

(6) Figures 7 B, C, D: what is v1/v2? Original vs SM-Connectome?

We thank the reviewer for noticing this and have corrected the figure to use “Orig” and “SM” consistent with the rest of the figure.

(7) Page 23, 2.10: what is phi?

We thank the reviewer for noticing this inconsistency with the earlier text, and have updated the text to read: “Particularly, we ran simulations for PF R ∈ [0.1, 0.15, ..., 0.3] using the OU para-maters calibrated for the seven column subvolume for [Ca^2+^] = 1.05 mM and R_OU_ = 0.4.”